# SINGLE ANSWER IS NOT ENOUGH: ON GENERATING RANKED LISTS WITH MEDICAL REASONING MODELS

## ABSTRACT

This paper presents a systematic study on enabling *medical* reasoning models (MRMs)–which achieve SOTA performance on multiple-choice benchmarks–to remain robust when producing alternative *answer formats*. Answer formats define the structure of a final answer in a generated response, such as an option, free text, or a ranked list. Although clinical decision-making typically involves weighing multiple plausible possibilities, current MRMs are trained to produce only one answer, and their robustness beyond that format is not well studied. We focus on the *ranked-list* format as an alternative that better reflects clinical uncertainty. To address this gap, we evaluate *prompting* and *fine-tuning* for enabling MRMs to generate ranked lists across common medical benchmarks. While prompting provides a lightweight solution, MRMs vary widely in their ability to follow such instructions. We therefore explore supervised fine-tuning (SFT) and reinforcement fine-tuning (RFT) as stronger adaptation methods. SFT trains models to imitate ranked outputs, whereas RFT optimizes behavior through reward functions; we introduce new rewards tailored to ranked-list generation and analyze their effects through ablations. Our results show that although some SFT models handle certain formats well, RFT yields more consistent robustness across multiple answer formats. A case study on a modified MedQA benchmark with multiple valid answers further reveals that MRMs can recognize clinically sound alternatives even when misaligned with a benchmark's preferred ground truth. To the best of our knowledge, this is the first systematic investigation of adapting MRMs to alternative answer formats such as ranked lists. We hope this study lays the foundation for developing more flexible and clinically aligned MRMs.

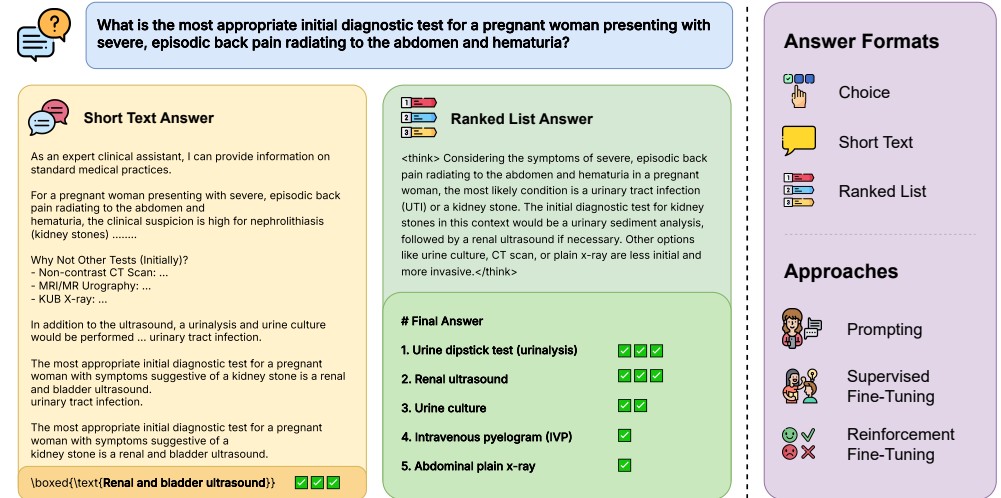

Figure 1: **Left** Comparison of model responses between a correct *short answer* and a *list answer* containing possible correct answers, which depends on the clinical context. **Right** We systematically study medical LLMs across three answer formats—*choice*, *short text*, and *list answers*—using three approaches: *prompting*, *supervised fine-tuning*, and *reinforcement fine-tuning*

# 1  INTRODUCTION

Recent advances in reasoning models have shown remarkable progress in verifiable domains such as mathematics and programming (Comanici et al., 2025; Yang et al., 2025; Guo et al., 2025). The medical domain, however, presents a distinct challenge. Clinical decision-making rarely involves selecting a single best treatment; instead, it requires managing uncertainty by weighing multiple plausible options (Varkey, 2020). Current medical reasoning models (MRMs), trained primarily on multiple-choice question (MCQ) datasets, inherit a single-best-answer bias and are optimized to produce only one answer (Chen et al., 2024; Huang et al., 2025a; Liu et al., 2025a).

In practice, single-answer outputs can mislead clinicians, who may over-rely on automated, potentially incorrect recommendations. By contrast, presenting a ranked list of options better reflects diagnostic uncertainty and supports safer decision-making (Goddard et al., 2012; Meng et al., 2024; Tao et al., 2020). For example, Jabbour et al. (2023) show that clinicians are more likely to adopt an incorrect diagnosis of pneumonia–even when it contradicts their initial judgment–simply because it appears as the model's proposed answer. This risk naturally extends to MRMs that produce only single-answer responses.

We argue that a *ranked list of plausible options* is a more informative and clinically aligned answer format. Ranked lists mirror clinical practice, where practitioners generate a differential diagnosis (DDx) (Lamba et al., 2021) before forming management plans, and they encourage collaboration between models and human experts by prompting reevaluation of the presented options. Figure 1 illustrates the contrast between single-answer and ranked-list responses.

Despite these motivations, the **robustness**[1] of MRMs across **answer formats** remains understudied– specifically, *how well a model trained in one format can follow instructions to produce another.* For example, it is unclear how reliably an MCQ-trained MRM can generate a ranked list when prompted. Our study focuses on ranked lists with a single ground-truth answer, evaluating whether the correct answer appears at the top of the list. This setup reflects limitations of existing medical QA benchmarks, as discussed in Section 5, and differs from differential diagnosis tasks in which multiple plausible answers matter (Lim et al., 2025). Nevertheless, our goal extends beyond differential diagnosis: we seek to understand answer-format robustness more broadly across general medical QA and related settings.

This motivates the central question of our study: **How can MRMs be made to generate ranked lists from open-ended problems?** To answer this, we investigate two approaches: **prompting**, which guides existing models toward the desired format, and **fine-tuning**, which trains models to produce responses in that format.

Prompting offers a lightweight way to steer model behavior without additional training; however, its effectiveness for MRMs is less understood than in general medical LLMs (Sahoo et al., 2025; Schulhoff et al., 2025). Compounding this challenge, recent work suggests that reasoning models may exhibit degraded instruction-following ability (Li et al., 2025a; Fu et al., 2025; Jang et al., 2025), directly limiting their ability to adopt new answer formats. We therefore examine how well prompting alone can steer MCQ-trained MRMs to generalize to other formats.

Fine-tuning, in contrast, modifies model weights and provides a stronger mechanism for steering behavior. We study two established methods: supervised fine-tuning (SFT) and reinforcement fine-tuning (RFT). SFT trains models on annotated prompt–response pairs, which in reasoning models can be derived through data synthesis (Chen et al., 2024; Qin et al., 2024) or knowledge distillation (Huang et al., 2024; 2025b). RFT instead optimizes reward signals (Zhang et al., 2025a; Guo et al., 2025), allowing models to discover behaviors that better align with the target answer format. We evaluate how MRMs trained via SFT or RFT generalize across answer formats and conduct ablation studies to analyze how different RFT design choices affect performance and stability.

Our findings show that SFT-trained MRMs can inadvertently entangle answer format with knowledge recall, reducing performance when evaluated on formats not seen during training. SFT models trained specifically on ranked lists generalize poorly to other formats. In contrast, RFT models

---

[1]In this paper, we use the terms *robust* and *generalizes* interchangeably to describe a model's ability to perform well across different answer formats.

trained on the MCQ format exhibit strong robustness across answer formats, though they still underperform compared with RFT models trained directly on the ranked-list format.

Our contributions are as follows:

- We present the first systematic study of steering MRMs via prompting and fine-tuning for ranked-list generation. We introduce an evaluation framework covering three answer formats–MCQ, QA, and List–and a pipeline for converting MCQ benchmarks into open-ended formats. We find that many models may know the correct answers but may often fail to select the benchmark-preferred one.
- We conduct a comprehensive study of SFT and RFT under specific answer formats for MRMs. We show that SFT trained on MCQs generalizes reasonably well across formats, whereas RFT models generalize both to unseen examples and across answer formats.
- We develop and propose reward functions for RFT targeting ranked-list generation, each offering different trade-offs in model behavior.
- We perform extensive ablation studies of RFT for MRMs, demonstrating that sequencing multiple RFT stages as a curriculum over answer formats (MCQ $\rightarrow$ List) improves stability and reduces collapse compared with training directly on the list format.

## 2 EXPERIMENTAL SETUP

### 2.1 ANSWER FORMATS

We define an *answer format* as the structure in which the model must present the final answer in its generated response. In this study, we focus on three formats: **MCQ**, **QA**, and **List**. The **MCQ** format evaluates models in the conventional multiple-choice setting, where the output is a single selected option. The **QA** format serves as an open-ended baseline, requiring the model to produce a short free-text answer rather than choosing from predefined options. The **List** format generalizes QA by allowing multiple answers within a single response, capturing scenarios involving alternatives or multiple plausible options. QA is a special case of the List format, corresponding to a single-element list. In the List setting, we expect the model to output an ordered list of short textual items, each representing a potential answer. Because it better reflects clinical uncertainty and our target task setting, the List format is the primary focus of our experiments. Examples of QA and List outputs are provided in Figure 1.

### 2.2 EVALUATION

**Datasets** For MCQ answer format, we evaluate approaches on standard medical MCQ benchmarks: (1) **MedQA** (Jin et al., 2021), (2) **MedMCQA** (Pal et al., 2022), (3) **MedXpertQA** (Zuo et al., 2025) (text), and (4) **MMLU Pro** (Wang et al., 2024) (health). For QA and List answer formats, we convert the MCQ into *open-ended QA benchmarks* with an LLM-based pipeline similarly to Myrzakhan et al. (2024). The LLM generates an open-ended version of an MCQ question with the same ground-truth answer if it deems the question convertible. See Appendices B and D.

**Metrics** We evaluate model performance using *accuracy* (Acc) and *mean reciprocal rank* (MRR) (Radev et al., 2002). Accuracy measures whether the model is able to output the correct ground truth (choice or exact short answer). This applies to MCQ questions ($\text{Acc}_{\text{MCQ}}$), QA ($\text{Acc}_{\text{QA}}$), and ranked-list ($\text{Acc}_{\text{List}}$) answer formats. For the list answer format, MRR ($\text{MRR}_{\text{List}}$) further evaluates *the position of the correct item* in the list, assigning higher scores when the correct item appears earlier. MRR provides a proxy for how effectively a model ranks correct items within its list outputs.

Since models may produce correct answers that do not exactly match the ground-truth string for non-MCQ formats, we also utilize LLM-based evaluation variants (Arora et al., 2025): $\text{Acc}_{\text{QA}}^{\text{LLM}}$, $\text{Acc}_{\text{List}}^{\text{LLM}}$, and $\text{MRR}_{\text{List}}^{\text{LLM}}$. All metrics are normalized to $[0, 1]$ and reported as percentages. Additional details are in Appendix E.

**Ranked List Analyses** We utilize additional metrics to assess differences in generated lists: (1) *average list length*–the mean number of items in valid, non-empty, lists (**VLL**), and (2) *average correct-answer position* (**CP**). These metrics show how models express uncertainty: CP reflects

confidence calibration, while VLL signals potential gaming through over-generation or hallucination through excessively long outputs. they characterize model behavior under different settings. We also examine response length and training dynamics in Appendices I and J.

## 2.3 MODEL SELECTION

We include state-of-the-art (SOTA) MRMs trained with different approaches in this study: **HuatuoGPT-o1**, SFT on synthetic data from an agentic pipeline (Chen et al., 2024); **m1**, SFT on distilled data from a teacher model (Huang et al., 2025a); and **AlphaMed**, RFT using MCQ-based verifiable rewards (Liu et al., 2025a). They share the same initial model, **Qwen2.5-7B-Instruct** (Qwen et al., 2025), which we select as the *primary model* in this study and as our main baseline for fine-tuning experiments. We also explore the effects of initial models for RFT in Section 4.3.2.

We broaden the comparison on SOTA proprietary and open-weight models. For proprietary models, we include the **Gemini 2.5** family (Flash Lite, Flash, Pro) (Comanici et al., 2025) to assess intra-family performance, and **GPT-4.1-mini** (OpenAI, 2025) for comparison with Gemini 2.5 Flash. For open-weight models, we evaluate **MedGemma** (4B and 27B) (Sellergren et al., 2025), with **Gemma 3-4B-Instruct** (Team et al., 2025) (the general counterpart), and **OpenThinker3-7B** (Guha et al., 2025), a reasoning model built on Qwen2.5-7B-Instruct. Within the Qwen family, we also consider the **3B**, **14B** sizes, and newer **Qwen3-4B** (Yang et al., 2025). Appendix C provides more details.

## 3 PROMPTING

We design six prompt templates derived from three answer formats–**MCQ**, **QA**, and **List**–and two variants: zero-shot and chain-of-thought (**CoT**). These are used for both evaluation and as prior prompts for RFT, where prior prompt (Taveekitworachai et al., 2025) guides model behavior *during RFT* and can differ from prompt used *during inference*. The CoT variant follows Xie et al. (2025), with added answer format instructions. Zero-shot variants omit reasoning instructions (`<think>` tags) while retaining the core components. Based on this setup, we expect models to perform best in the MCQ format, worse in the QA format, and improved in the ranked-list format compared to QA, since lists increase the likelihood of producing a correct answer in a single inference. Table 1 presents the average performance across benchmarks for each model and answer format, including the CoT variant (non-aggregated results in Appendix N.1). Additional discussion in Appendix I.1.

**MCQ**   To set a baseline, we first evaluate MCQ performance. As expected, proprietary models outperform open-weight ones, even without domain-specific training. Most medical models are competitive to the proprietary models, with the exception of HuatuoGPT-o1. HuatuoGPT-o1, SFT on data synthesized via an agentic pipeline, improves over its base model (Qwen2.5 7B Instruct) but lags behind other MRMs, often ignoring the required format and producing unparsable outputs (see Figure 28.) In contrast, m1–SFT on teacher-distilled data–performs competitively, though its reasoning style mirrors the teacher's verbosity (see Figure 29). This contrast suggests that designed synthetic data may introduce greater distributional shift than naturally generated data (Li et al., 2025b).

**MCQ vs. QA**   Contrary to the expectation, QA accuracy often exceeded MCQ accuracy (in 7 of 15 cases where $\text{Acc}_{\text{QA}}^{\text{LLM}}$ exceeded $\text{Acc}_{\text{MCQ}}$ by more than 3 pp., including open-weight models such as Qwen, Gemma 3 4B, MedGemma 4B, and OpenThinker3), suggesting these models can retain knowledge independent of format. In contrast, SOTA proprietary models (Gemini 2.5 Pro, GPT-4.1-mini) and MRMs (HuatuoGPT-o1, AlphaMed) declined, indicating reliance on answer-format cues. Overall, smaller open-weight models appear more robust across formats, while stronger proprietary and medical models depend more on format cues for knowledge recall (Singh et al., 2025).

**QA vs. List**   Except for Qwen2.5 7B Instruct and OpenThinker3 7B, all models achieve higher $\text{Acc}_{\text{List}}^{\text{LLM}}$ than $\text{Acc}_{\text{QA}}^{\text{LLM}}$, likely indicating they often **know the correct answer but fail to select the benchmark's choice**. Although additional studies are required to confirm these causal effects. The gap between $\text{Acc}_{\text{List}}^{\text{LLM}}$ and $\text{MRR}_{\text{List}}^{\text{LLM}}$, along with low $\text{MRR}_{\text{List}}^{\text{LLM}}$ and CP, supports this finding (see Section 5). Still, most models rank the correct answer near the top, showing they can generally

Table 1: Performance of proprietary, open-weight, and medical LLMs under zero-shot and CoT prompting across MCQ, QA, and List benchmarks. Positive and negative CoT effects are colored; **bold** highlights the best overall score or largest change per metric. For ranked list answers, **CP** is the average rank of the correct item and **VLL** the average length of valid, non-empty, lists.

| | MCQ | QA | | List | | | | | |
|---|---|---|---|---|---|---|---|---|---|
| | $Acc_{MCQ}$ | $Acc_{QA}$ | $Acc_{QA}^{LLM}$ | $Acc_{List}$ | $Acc_{List}^{LLM}$ | $MRR_{List}$ | $MRR_{List}^{LLM}$ | CP | VLL |
| *Proprietary Models* | | | | | | | | | |
| Gemini 2.5 Flash Lite | 48.47 | 12.35 | 48.69 | 14.84 | 53.82 | 13.34 | 46.36 | 1.39 | 2.86 |
| +CoT | -22.88 | -1.01 | -6.27 | -6.74 | -29.44 | -5.62 | -23.42 | 1.17 | 1.92 |
| Gemini 2.5 Flash | 55.19 | 11.95 | 46.10 | 17.49 | 62.66 | 15.51 | 53.52 | 1.41 | 3.00 |
| +CoT | -19.75 | -0.51 | -1.69 | -9.99 | -35.74 | -8.59 | -29.40 | 1.30 | 2.66 |
| Gemini 2.5 Pro | 58.68 | 10.89 | 49.20 | 18.44 | 68.46 | 16.44 | 58.85 | 1.40 | 3.41 |
| +CoT | -0.62 | -0.58 | -1.12 | -1.94 | -3.18 | -1.96 | -3.09 | 1.41 | 3.46 |
| GPT-4.1 Mini | 54.72 | 9.30 | 47.02 | 12.78 | 61.71 | 11.42 | 53.82 | 1.36 | 3.26 |
| +CoT | -7.01 | +1.38 | +1.98 | +0.07 | +3.06 | -0.04 | +2.69 | 1.36 | 3.72 |
| *Open-weight General Models* | | | | | | | | | |
| Qwen2.5 3B Instruct | 29.62 | 5.68 | 35.66 | 9.18 | 38.70 | 7.35 | 29.55 | 1.69 | 3.09 |
| +CoT | +3.06 | +0.21 | +4.53 | -1.65 | -9.25 | -0.25 | -3.13 | 1.27 | 1.80 |
| Qwen2.5 7B Instruct *(our initial model)* | 13.43 | 7.90 | 43.19 | 10.38 | 39.96 | 8.81 | 33.07 | 1.45 | 2.39 |
| +CoT | +17.81 | +0.56 | +0.16 | +0.66 | +8.57 | +0.06 | +5.44 | 1.91 | 185.72 |
| Qwen2.5 14B Instruct | 35.88 | 10.41 | 45.33 | 13.09 | 52.84 | 11.20 | 43.44 | 1.48 | 3.05 |
| +CoT | +0.45 | -0.30 | +2.47 | -0.45 | -4.71 | +0.07 | -0.82 | 1.30 | 2.36 |
| Qwen3 4B Instruct 2507 | 43.82 | 10.47 | 47.22 | 11.39 | 53.01 | 8.96 | 40.49 | 1.70 | 3.96 |
| +CoT | -2.70 | -0.45 | -2.54 | +1.20 | -2.43 | +1.91 | +1.56 | 1.50 | 3.56 |
| Gemma 3 4B IT | 30.43 | 6.22 | 36.62 | 11.77 | 47.77 | 8.72 | 34.84 | 1.83 | 4.68 |
| +CoT | -2.29 | +0.53 | +3.54 | -2.48 | -10.34 | -1.53 | -5.75 | 1.67 | 4.13 |
| OpenThinker3 7B | 27.57 | 5.27 | 31.03 | 5.44 | 28.56 | 5.00 | 24.51 | 1.43 | 3.06 |
| +CoT | **-23.82** | -0.41 | -0.33 | -5.32 | -27.72 | -4.89 | -23.77 | 1.29 | 4.76 |
| *Open-weight Medical Models* | | | | | | | | | |
| MedGemma 4B IT | 37.09 | 7.85 | 43.19 | 13.34 | 53.34 | 9.58 | 38.65 | 2.15 | 95.06 |
| +CoT | -6.57 | +0.42 | -0.35 | +0.22 | -3.50 | +0.25 | -2.98 | 3.14 | 482.76 |
| MedGemma 27B IT | 48.97 | **12.78** | 47.64 | 14.45 | 50.74 | 12.78 | 43.33 | 1.46 | 3.26 |
| +CoT | -16.24 | -2.18 | -6.68 | +0.73 | +2.42 | +0.74 | +2.57 | 1.43 | 3.88 |
| HuatuoGPT o1 7B | 17.75 | 1.10 | 3.83 | 10.13 | 35.68 | 7.87 | 27.58 | 1.70 | 4.39 |
| +CoT | -7.54 | -0.53 | -1.63 | -9.84 | -35.13 | -7.61 | -27.10 | 1.46 | 2.45 |
| m1 7B 23K | 39.26 | 7.65 | 38.97 | 11.04 | 50.02 | 8.26 | 36.62 | 2.01 | 13.39 |
| +CoT | -7.88 | +0.06 | -1.71 | -5.34 | -14.00 | -3.64 | -7.28 | 1.64 | 19.03 |
| AlphaMed 7B Instruct RL | 40.51 | 0.82 | 9.46 | 4.48 | 19.25 | 3.27 | 14.26 | 1.83 | 2.59 |
| +CoT | -3.04 | **+4.88** | **+15.00** | -0.13 | +1.56 | +0.20 | +2.14 | 1.89 | 55.74 |
| *Our Knowledge-Distilled MRMs (based on Qwen2.5 7B Instruct)* | | | | | | | | | |
| SFT-MCQ | **39.60** | 11.08 | **48.04** | 15.48 | 57.65 | 11.17 | 39.71 | 2.83 | 141.72 |
| +CoT | +1.94 | +0.64 | -0.48 | -3.98 | -21.80 | +0.17 | -5.04 | 1.09 | 1.46 |
| SFT-QA | 37.67 | **11.25** | 46.57 | 12.94 | 51.18 | 10.23 | 38.92 | 1.81 | 13.85 |
| +CoT | +0.13 | +0.01 | +0.47 | -5.06 | -26.39 | -2.57 | -15.24 | 1.15 | 1.78 |
| SFT-List | 10.68 | 0.24 | 1.15 | 15.26 | 48.91 | **13.49** | **41.85** | 1.41 | 2.52 |
| +CoT | +1.94 | +0.18 | +0.65 | -0.30 | -0.01 | -0.21 | -0.31 | 1.42 | 2.57 |
| *Our RFT MRMs (based on Qwen2.5 7B Instruct)* | | | | | | | | | |
| RFT-MCQ | **39.34** | 9.96 | **46.33** | 9.89 | 40.06 | 8.16 | 33.00 | 1.45 | 2.29 |
| RFT-QA | 36.80 | 1.04 | 25.22 | 0.62 | 3.59 | 0.48 | 2.82 | 1.67 | 3.01 |
| RFT-List-Acc | 22.40 | 4.28 | 19.01 | **22.11** | 56.61 | **16.17** | 40.26 | 2.07 | 5.94 |
| RFT-List-MRR | 18.23 | 4.43 | 21.90 | 20.96 | **61.60** | 15.83 | 44.89 | 2.11 | 16.97 |
| RFT-List-Judge-MRR | 20.49 | 6.49 | 30.36 | 14.86 | 60.90 | 12.16 | **48.68** | 1.64 | 4.46 |

order plausible options. Lists are usually short (2–5 items), suggesting models generate mostly relevant candidates, though MedGemma 4B is an exception with VLL = 95.06.

Medical fine-tuning improves MedGemma 4B and m1 over their initial models but yields smaller gains for HuatuoGPT-o1 and AlphaMed. HuatuoGPT-o1 underperforms its base model, likely due to overfitting to response format, as previously discussed. In contrast to HuatuoGPT-o1, m1 performs well across answer formats (revisit in Section 4.1), despite also using SFT–the key difference lies in the data source, as previously discussed. However, AlphaMed, an MRM trained with RFT for MCQ, fails to generalize to other answer formats; we revisit this finding again in Section 4.2.

**Reassessing CoT with modern models** We reassess CoT as a precursor technique to modern reasoning models and find limited benefits. Only Qwen2.5 7B Instruct shows consistent gains, and AlphaMed improves slightly on QA. In most other cases–especially MCQ tasks and the Gemini 2.5 family–CoT provides little or even harms performance. Although CoT generally reduces accuracy, it reliably alters behavior: nearly all models produce longer outputs (except Gemma 3 4B on QA/list

tasks), yet this verbosity shows a weak negative correlation with accuracy (see Appendix I.1). CoT can sometimes improve ranking, as CP decreases or remains stable. This suggests that when models already know the correct answer, CoT makes them more confident in ranking it higher. The only exceptions are Qwen2.5 7B Instruct and MedGemma 4B, which increase CP while generating long lists with repetitive items (185+ and 480+ items; see Section 4.3.2 and appendix L).

# 4 FINE-TUNING

For the fine-tuning datasets, we adopt the training set from AlphaMed (Liu et al., 2025a), which includes MedQA and portions of the MedMCQA training splits. This makes both MedQA and MedMCQA in-domain benchmarks for evaluation. The MCQ version is used to train MCQ answer formats, while the open-ended version (see Section 2.2) is used to train QA and list answer formats, each paired with its respective prior prompt. All experiments are done on 4×H100 GPUs.

## 4.1 SUPERVISED FINE-TUNING (SFT)

**Experimental Setup** To obtain a training set, we distill from **Qwen3-30B-A3B-Thinking-2507** (Yang et al., 2025) (a reasoning variant) on the AlphaMed datasets, prepended with each prompt template and filtered by rejection sampling (`max_retries`=20; only correct responses retained). Additional details, including a number of records, are provided in Appendix G. We then fine-tune **Qwen2.5-7B-Instruct** on the corresponding datasets to obtain three models (SFT-MCQ, SFT-QA, and SFT-List), one for each answer format considered. Fine-tuning is performed with `LLaMA-Factory` (Zheng et al., 2024) using the hyperparameters provided in Appendix H.1.

**Results and Discussions** Table 1 shows that SFT on distilled datasets, regardless of answer format, generally outperforms initial models. Examples of generated responses from each model is available in Appendix O.1. CoT, however, offers limited benefit to SFT models, similar to what is observed with m1, regardless of format. CoT slightly improves MCQ performance and CP but often shortens lists, unlike m1, except in SFT-List. We speculate that, having been trained on a ranked-list format, CoT leads SFT-List to "overthink," adding more items to the list to be safe.

Given that SFT-MCQ and m1 are trained with a similar approach, differing only in training sets and hyperparameters, it is unsurprising that SFT-MCQ performs well across formats, similar to m1. These models achieve the best results not only on MCQ but also on QA ($\text{Acc}_{\text{QA}}^{\text{LLM}}$) and list formats ($\text{Acc}_{\text{List}}^{\text{LLM}}$). However, SFT-MCQ produces unusually long lists (VLL = 141.72, CP = 2.83, see Figure 33), and unlike m1, its lists shorten considerably when CoT is applied. The model initially produces diverse items but soon repeats content, possibly because list-style outputs are underrepresented in its training data, lowering the likelihood of generating a stop token.

While SFT-QA generalizes across formats, though not best on QA, SFT-List achieves the highest $\text{MRR}_{\text{List}}^{\text{LLM}}$ but performs poorly on $\text{Acc}_{\text{List}}^{\text{LLM}}$, MCQ, and QA, showing weak robustness across formats. SFT-List persistently outputs lists–even when instructed with other formats–yielding unextractable answers. We conjecture that list formats are underrepresented in LLM training, leading to over-association between list format and knowledge expression.

## 4.2 REINFORCEMENT FINE-TUNING (RFT)

RFT (Zhang et al., 2025a) is based on RL using verifiable rewards (Lambert et al., 2025), which can be directly checked against ground truth, often without a learned reward model. Unlike SFT, where a model learns fixed input–output pairs, RFT incentivizes responses that maximize reward, balancing exploration and exploitation. For example, AlphaMed uses RFT to encourage extended reasoning traces that improve medical MCQ performance without explicit response examples. In this study, we extend RFT beyond MCQ to the ranked-list format. As reward functions for this format are not well studied, we design one for our setting and assess its effectiveness. Section 4.3 details ablations on the effects of initial models, reward functions, and hyperparameters.

### 4.2.1 EXPERIMENTAL SETUP

We RFT **Qwen2.5 7B Instruct** using the AlphaMed dataset for the MCQ answer format and QA-AlphaMed for the QA and ranked-list formats, resulting in five models for the main experiments, as the ranked-list format includes three types of reward functions (RFT-MCQ, RFT-QA, RFT-List-Acc, RFT-List-MRR, and RFT-List-Judge-MRR). Training is performed with `verl` (Sheng et al., 2025), with hyperparameters and additional details provided in Appendix H.2.

**Reward Function**  The reward function is a crucial component of the RFT setup. It provides the signal that incentivizes desired model behaviors during training, e.g., assigning high scores to correct answers. We adopt a design similar to Guo et al. (2025), consisting of two components: (i) a **correctness reward**, which measures whether the output matches the ground truth, and (ii) an optional **format reward**, which checks proper output formatting when using CoT prior prompts. All rewards are normalized to $[0, 1]$, with each component weighted equally.

CORRECTNESS REWARD  The correctness reward is task-dependent. Let $\hat{y}$ denote the model prediction, $y^*$ the ground truth, $\mathbb{1}[\cdot]$ the indicator function, and $N(\cdot)$ a normalization function. For multiple-choice and QA tasks, we define $R_{\text{MCQ}} = \mathbb{1}[\hat{y} = y^*]$ and $R_{\text{QA}} = \mathbb{1}[N(y^*) \text{ is a substring of } N(\hat{y})]$. For list-style answers, $y^*$ is compared against all items $\hat{Y} = \{\hat{y}_1, \ldots, \hat{y}_n\}$, with $R_{\text{List}} = \max_{i=1,\ldots,n} R_{\text{QA}}(y^*, \hat{y}_i)$. This treats the list as an *unordered set*: the model receives full reward if the ground truth appears anywhere in the list, regardless of position.

RANK-AWARE CORRECTNESS FOR LISTS  We design a new reward inspired by MRR that incentivizes *higher placement* of correct answers. Let $r$ denote the position of the first correct item (1-indexed). We define $R_{\text{MRR}} = R_{\text{List}}(y^*, \hat{y}_i) \cdot 1/r$. However, models may exploit this by generating excessively long lists to maximize coverage. A length-penalized variant is discussed in Appendix L.

LLM JUDGE REWARD  Inspired by recent studies that use LLM judges as rewards in RFT (Guo et al., 2025; Su et al., 2025; Zhou et al., 2025), we extend MRR with an LLM-based judge (Judge-MRR), where semantic equivalence between each $\hat{y}_i$ and $y^*$ is determined by an LLM rather than by normalized exact match. This provides a more accurate reward signal in cases with varied wording. We ablate different models and prompts for Judge-MRR in Section 4.3.1.

FORMAT REWARD  When using a CoT variant of the prior prompts, we add a format reward (Guo et al., 2025) that checks whether the output contains exactly one pair of `<think>` and `</think>` tags, with `<think>` appearing at the very start. The reward is $R_{\text{format}} = 1$ if the conditions are satisfied, and $0$ otherwise. We compare setups with and without this format reward in Section 4.3.1.

### 4.2.2 RESULTS AND DISCUSSIONS

**General Performance Trend**  From Table 1, RFT-MCQ matches SFT-MCQ and m1, confirming that medical training improves performance over initial models. Like SFT-MCQ, RFT-MCQ performs best on QA benchmarks, while RFT-QA improves but still lags behind. Since MCQ training benefits both MCQ and QA formats, and list training benefits list tasks, we further examine the effects of mixed-format training in Appendix K. Most RFT models generate short lists, except RFT-List-MRR, which–unlike the repetitive outputs of the SFT models–produces long but diverse lists.

RFT-MCQ and AlphaMed, trained on the same dataset, show similar overall trends but diverge on QA and list tasks. This may be due to differences in training setup: AlphaMed was trained with roughly twice as many steps, while RFT-MCQ also incorporates a format component in the reward function. We analyze the effects of training duration and reward design in Section 4.3.1. We provide examples of generated responses from the RFT models in Appendix O.2.

**RFT-QA Exhibits List-like Behavior**  We find many cases where RFT-QA generates multiple answers inside \boxed{} during QA evaluation (see Figure 35). This list-like behavior suggests that under our settings, the model finds it natural to output multiple answers to increase its chance of receiving a reward, given that the reward function is based on substring-normalized matching. This constitutes reward exploitation, i.e., reward hacking. However, while such responses achieve higher rewards during training, they lead to poor evaluation scores on LLM-based metrics, as judges

are instructed to penalize this behavior. We also observe that this style carries over to list-format evaluation, where individual items often contain multiple answers, resulting in significantly lower scores. This emphasizes the importance of reward design, which we discuss further in Section 4.3.1.

**Reward Function Effects With RFT-List**  RFT-List models excel on list-based metrics: RFT-List-MRR achieves the highest $\mathrm{Acc}_{\mathrm{List}}^{\mathrm{LLM}}$, while RFT-List-Judge-MRR leads on $\mathrm{MRR}_{\mathrm{List}}^{\mathrm{LLM}}$. All outperform SFT-List and, unlike it, also perform robustly on non-list formats. Examples of generated responses from RFT-List-Judge-MRR under MCQ, QA, and List evaluations are shown in Figures 38, 40 and 41, respectively. This indicates that RFT is not only more robust across answer formats than SFT, but also extends prior findings that RFT generalizes better to *unseen examples* (Chu et al., 2025) to the setting of *unseen answer formats*.

We conjecture that the same mechanism underlying RFT's advantage on unseen examples also applies here: RFT tends to prune irrelevant or inaccurate knowledge from its reasoning paths, reducing entanglement between knowledge recall and answer format. In contrast, SFT attempts to imitate the full reasoning trajectory, which can inadvertently bind answer formats to the specific reasoning or knowledge patterns observed during training (Chu et al., 2025).

In addition, reward choice shapes final performance. Among the three list rewards, $\mathrm{Acc}_{\mathrm{List}}$ boosts exact matches, while $\mathrm{MRR}_{\mathrm{List}}^{\mathrm{LLM}}$ encourages longer lists, increasing the chance of including the correct answer and yielding the highest $\mathrm{Acc}_{\mathrm{List}}^{\mathrm{LLM}}$. The $\mathrm{MRR}_{\mathrm{List}}^{\mathrm{LLM}}$ reward, as expected, produces the best $\mathrm{MRR}_{\mathrm{List}}^{\mathrm{LLM}}$. Since none of the current rewards explicitly control list length, $\mathrm{MRR}_{\mathrm{List}}$ rewards result in the longest outputs. Appendix L discusses a length-penalized variant of the reward functions.

### 4.3 Ablation Studies for RFT

In this section, we present ablation studies on factors influencing medical reasoning models trained with RFT. Section 4.3.1 examines the format component of the reward function, extended training, and the Judge-MRR setup. Finally, Section 4.3.2 studies the effect of the initial model choice.

#### 4.3.1 Factors Affecting RFT

We conduct scaled-down ablation studies to better understand effects of key factors in RFT training, including the **format component of the reward function**, **extended training duration**, the **role of prior prompts**, and the **choice of judge model**. The full results are available at Appendix I.5.1.

**Excluding the format reward or extending training has limited effect on the model**  Removing the format reward does not substantially affect final model performance or list behaviors. Similarly, extending training from two to four epochs does not substantially improve performance. For example, in RFT-MCQ, $\mathrm{Acc}_{\mathrm{MCQ}}$ remains around 39% with or without the format reward (39.34% → 39.56%), and four epochs very slightly increase it to 39.97%.

**Effects of prior prompts**  Prior prompts can influence the initial optimization space during RFT and interact with reward function components. Removing or modifying prior prompts shows mixed effects, and our ablation scale remains insufficient for conclusive findings (see Appendix I.5.1).

**Different judge models**  Changing the judge model substantially impacts performance. Replacing GPT-4.1-mini with Gemini 2.5 Flash improves $\mathrm{Acc}_{\mathrm{QA}}^{\mathrm{LLM}}$ from 30.36% to 43.16% and boosts MCQ accuracy (20.49% → 33.11%), while maintaining comparable list accuracy (60.90% vs. 59.34%). In contrast, simplifying the judge prompt severely degrades ranked-list evaluation, with $\mathrm{MRR}_{\mathrm{List}}^{\mathrm{LLM}}$ falling from 48.68% to 26.19%. This degradation arises because the model exploits weaknesses in the simplified judge by producing vague or grouped answers, which yield higher training rewards (see Figure 18a) but fail to generalize at evaluation time. These findings highlight that both the choice of judge and the design of the judge prompt are critical to final performance.

#### 4.3.2 Effects of Initial Models Used in RFT

To examine how findings generalize across base models, we extend our RFT setup to **smaller models** (*Qwen2.5 3B Instruct*), **more recent model families** (*Qwen3 4B Instruct*), and **continual RFT**

**from existing reasoning models**. We consider three scenarios: continual RFT from (1) **a general reasoning model**, *OpenThinker3* (domain adaptation); (2) **an MRM trained with SFT-MCQ**, *m1*; and (3) **an MRM trained with RFT-MCQ**, *AlphaMed*. Appendix I.5.2 provides full results.

**Model family and scale**  Qwen3 4B after RFT becomes competitive with Gemini 2.5 Pro (RFT-List-Acc's 53.01% $\rightarrow$ 71.60% vs. 68.46% $Acc_{List}^{LLM}$, and RFT-List-MRR's 47.22% $\rightarrow$ 48.54% vs. 49.20% $Acc_{List}^{LLM}$). Despite being similar in size to Qwen2.5 3B, Qwen3 4B is consistently stronger. Qwen2.5 3B struggles with RFT-QA setup (27.60%) but benefits from RFT-List setups (35.66% $\rightarrow$ 40.20% $Acc_{QA}^{LLM}$, 38.70% $\rightarrow$ 59.82% $Acc_{List}^{LLM}$). We conjecture that RFT-List setups provide denser signals that transfer to QA, as the model can attempt multiple answers in a single inference call, allowing it to incorporate more from the training data.

**Continual RFT benefits reasoning models**  For OpenThinker3, MCQ and QA performance converge to a similar range across RFT setups, but the RFT-List setup yields a substantial boost: $Acc_{MCQ}$ rises from 27.57% to 33.74–34.60%, and $Acc_{QA}^{LLM}$ from 31.03% to 39.78–41.42%. In contrast, for list evaluations, non-list RFT models reach only 33.28–39.02% $Acc_{List}^{LLM}$, whereas list-based RFT jumps to 56.98–59.44%, with $MRR_{List}^{LLM}$ improving from 24.51% to 35.03%.

A similar pattern holds for m1; all RFT setups bring MCQ and QA into a comparable range. However, only RFT-List improve list-format accuracy, while RFT-MCQ and RFT-QA reduce the performance. These results suggest that SFT $\rightarrow$ RFT is most beneficial when the setup is RFT-List.

AlphaMed further illustrates the benefits of sequencing: after initial RFT-MCQ, subsequent RFT-QA lifts $Acc_{QA}^{LLM}$ from 9.46% $\rightarrow$ 38.35% and RFT-List-Acc training improves $Acc_{List}^{LLM}$ from 19.25% $\rightarrow$ 57.29%, while retaining MCQ ability. However, RFT-MCQ$\rightarrow$RFT-MCQ degrades $Acc_{List}^{LLM}$, reinforcing that MCQ is easier but less transferable than QA/List answer formats.

**List rewards may incentivize a large list**  RFT-List setups often produce excessively long lists; for instance, Qwen2.5 3B, Qwen3 4B, OpenThinker3, and m1 average over 700 items under RFT-List-Acc/MRR training. This may stem from repetition at the tail end and a reduced probability of generating a stop token. AlphaMed does not show this behavior, suggesting that initial MCQ training stabilizes later QA/List training. This supports the view that curriculum learning in RFT is beneficial not only for sequencing data difficulty (Stojanovski et al., 2025; Xie et al., 2025), but also for sequencing answer formats—from simpler MCQ to more complex QA or list outputs. See Appendix K for further discussion and Appendix L for length-penalized rewards.

## 4.4 DISCUSSIONS

The reward functions introduced in Section 4.2.1 improve MRM accuracy across multiple answer formats and represent a step toward enhancing ranked-list generation. While these rewards implicitly convey a notion of model confidence–e.g., correct items should rank higher–they do not explicitly quantify the differences in confidence magnitudes. Knowing how strongly the model's confidence in rank 1 over rank 2 is valuable in some application, but supporting such calibrated confidence would require an additional training objective. Our approach is orthogonal to confidence calibration in RFT settings. For example, Damani et al. (2025) introduce calibration-oriented rewards, which are complementary to our reward functions.

Another consideration is that our experiments focus on academic benchmark settings, where performance (e.g., accuracy) is the primary metric. Additional objectives and safeguards would be required to translate our work to real-world clinical use. For instance, a fine-tuned model optimizing for benchmark accuracy may overlook practical constraints–such as avoiding potentially harmful suggestions or ensuring the inclusion of rare but clinically important conditions. These challenges reflect limitations in existing datasets, which often provide only a single ground-truth answer rather than multiple valid options. Future work could address this by constructing more realistic benchmarks or introducing reward functions that explicitly account for safety and clinical relevance.

Overall, our study aims to provide a foundation and a new perspective on answer-format–aware training for MRMs. Rather than replacing clinical experts, we view this direction as supporting more effective collaboration between models and practitioners. We hope this work encourages further

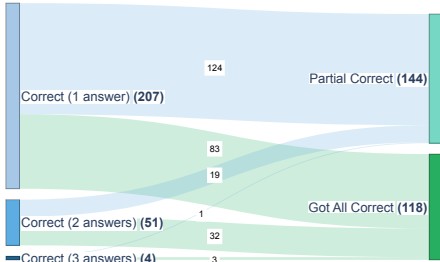

(a) Records shifting between *correct* and *incorrect* from single- to multi-answer evaluation.

(b) Correct responses split into those covering all vs. partial valid answers under multi-answer evaluation.

Figure 2: Re-evaluation of MedQA with multiple valid answers.

research into principled, clinically grounded approaches for training MRMs to generate richer and safer answer formats.

## 5 CASE STUDY: MEDQA WITH MULTIPLE VALID ANSWERS

As uncertainty–such as incomplete patient information–is common in real-world settings, multiple answers may be valid. A ranked-list format can broaden clinical perspectives and mitigate cognitive blind spots, inviting diverse views to guide patient care. However, these constraints are not fully accounted for in current medical benchmarks, which mostly rely on single-answer evaluation.

To demonstrate this, we adapt MedQA by adding metadata with multiple expert-annotated valid answers from Saab et al. (2024), creating a modified version where each record includes several valid answers. We then take ranked lists generated by **RFT-List-Judge-MRR** (our best model on $\text{MRR}_{\text{List}}^{\text{LLM}}$), from its original QA-MedQA evaluation, and re-evaluate them on the modified benchmark using a normalized exact-match comparison.

Results from Figure 2a show that, out of 1,149 records, 43 were classified as *incorrect* under single-answer evaluation but actually contained *valid answers* according to the modified benchmark. This suggests that the model may already possess sufficient knowledge to generate valid responses; however, it fails to select the answer that matches the benchmark's preferred label, highlighting inherent biases in benchmark development.

Furthermore, Figure 2b shows that 133 of the original 233 correct answers–including 55 records with more than one valid answer–contained all expert-validated answers within the generated lists. This indicates that models can capture nuanced sets of valid options when generating lists. Nevertheless, the current model is still limited: not all generated list answers include all valid cases. This limitation is unsurprising given that most training datasets and benchmarks nowadays are single-best-answer formats. Addressing this gap through improved datasets that capture the nuances of medical applications represents a promising future direction beyond the scope of this work.

## 6 CONCLUSIONS

We study how prompting and fine-tuning steer MRMs toward generating ranked-list answers. RFT generalizes better than SFT across diverse answer formats, and ranking-oriented rewards (e.g., MRR, Judge-MRR) further improve performance under ranked-list evaluations. Our study provides the first systematic comparison of prompting vs. fine-tuning for ranked-list generation in medicine, introduces reward functions inspired by MRR, and includes comprehensive ablations analyzing how initial models, reward design, and hyperparameters shape RFT performance. We hope this work motivates development of models and benchmarks that better reflect real-world clinical practice, moving beyond the limitations of single-answer evaluation.

## LIMITATIONS

As one of the first studies to investigate ranked-list answer generation for medical reasoning models, our work opens several promising directions for future research. In this paper, we focus exclusively on ranked lists, but alternative non-list formats could be explored in subsequent studies. Although our evaluation is limited to the medical domain, the underlying approach may generalize to other fields. Further progress will also require larger-scale experimentation. We also restrict our study to a monolingual and text-only setup. Extending the approach to multimodal and multilingual settings is an exciting avenue for future research. Given our limited compute budget, these results are informative in our setting but may not fully generalize to larger-scale RFT training, e.g., larger models, longer training, and larger datasets; however, we expect several findings to hold.

While we report behavioral metrics, we do not conduct a detailed analysis of reasoning traces, as our emphasis is on the final answer format. Changing this format may itself influence reasoning traces, which we leave for future investigation. Similarly, more extensive exploration of data, algorithms, and hyperparameters–beyond what was feasible under our computational budget–remains an important direction.

Another limitation is the lack of medical benchmarks with multiple correct or ranked answers. To approximate this setting, we adapted multiple-choice QA datasets into a ranked-answer format. Developing benchmarks that better capture the diversity of real-world clinical reasoning is beyond the scope of this work but represents a valuable opportunity for the community. In addition, we focus on a single-turn setting, consistent with traditional accuracy-oriented benchmarks. Evaluating and developing models that can respond in multiple formats within multi-turn interactions–such as sequential diagnosis, where ranked lists may be updated as new information becomes available, or dynamic settings like simulated patient encounters–is an interesting direction for future work.

Finally, ranked lists themselves have limitations. While they indicate which answers a model considers most likely, they do not reflect differences in probability magnitude. For instance, the top-ranked item may have a 50% likelihood, while the rest trail far behind. Capturing such disparities is an open challenge. We speculate that orthogonal approaches, such as those explored by Stangel et al. (2025); Damani et al. (2025), could complement ranked-list methods.

## ETHICS STATEMENT

This work evaluates methods for generating ranked medical answers using reasoning models, tested on academic benchmarks that may not fully reflect real-world clinical environments–an acknowledged limitation that our methods aim to partly address. LLMs remain prone to hallucinations, and their outputs must be interpreted with caution; they are not substitutes for professional judgments.

## REPRODUCIBILITY STATEMENT

We are committed to ensuring the reproducibility of our work. Detailed descriptions of models, datasets, training procedures, and evaluation protocols are provided in Section 2. Additional implementation details and training hyperparameters are available throughout the Appendix. In addition, datasets and source code are publicly available at the links in Appendix B and Appendix H.2.

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

# Appendix

## Table of Contents

## LLM Usage Statement

LLMs were used only for supportive tasks such as proofreading, grammar refinement, and polishing text. The research ideas, initial drafts, and core content were developed entirely by the authors; LLMs did not generate the main body of the manuscript. We also used LLMs to assist with code snippets for evaluation and analysis, but all codes and logic were reviewed by the authors.

## A  Related Work

### A.1  Medical Reasoning Model

The popularity of reasoning models in verifiable domains (e.g. mathematics and coding) (Zhang et al., 2025a) has inspired several studies in the medical domain to develop a medical reasoning model with similar approaches. Early work includes HuatuoGPT-o1 (Chen et al., 2024), which propose a data synthesis pipeline for supervised fine-tuning (SFT) and then perform reinforcement fine-tuning (RFT). Next, m1 (Huang et al., 2025a) investigates knowledge distillation from DeepSeek R1, an approach similarly to s1 for verifiable domains (Muennighoff et al., 2025). Later, AlphaMed (Liu et al., 2025a) proposed an approach utilizing RFT only, similarly to DeepSeek-R1-Zero. In contrast to these approaches, the objective of this paper aimed at investigating settings where reasoning models must answer as a list. Recently, Med-U1 (Zhang et al., 2025b) was released as an attempt to generalize RFT beyond MCQ settings. Med-U1 moves in a similar direction to ours, though with a different focus. Specifically, Med-U1 explores three answer formats–MCQ, numeric, and short answer–using different types of rewards for each. By contrast, we examine MCQ, short answer, and ranked-list answer formats, and we further investigate a variety of reward functions, including LM-based reward functions.

### A.2  Prompting

A notable example of prompting approaches is chain-of-thought prompting (Wei et al., 2022), which serves as a precursor to reasoning models. While there is a large body of work on prompting with LLMs, prompting in reasoning models has been less explored, and we lack a systematic understanding of how well prompting approaches that are effective in LLMs transfer to reasoning models. Furthermore, recent studies have shown that reasoning model has reduced instruction following capabilities (Li et al., 2025a; Fu et al., 2025; Jang et al., 2025), which directly associated with ability to prompt an LM. This study aims to systematically investigate whether reasoning models and modern LLMs can be prompted to generate answers in different formats.

### A.3  Fine-Tuning To Obtain A Reasoning Model

There are two main fine-tuning approaches in turning an LLM into a reasoning model: 1) SFT on responses with reasoning traces and 2) RFT with a verfiable reward function. SFT requires responses with reasoning traces to fine-tune a model to imitate the style of a reasoning model's answers. Often, gathering these training sets are often done by generating responses from existing reasoning models, i.e., knowledge distillation. In the field of medical reasoning models, m1 utilized this approach.

In contrast, RFT does not teach reasoning to the LM directly, but rather incentivizes the models to generate intermediate tokens, often called thoughts, that maximize the defined rewards during training. Reward design is one of the core factors of RFT for incentivization. In MCQ settings, the reward design is a simple verifiable reward on the correct choices. The core component of reward is often the accuracy reward, i.e., if the model generates a correct choice matched with the ground truth, the model scored the reward, and vice versa. Similar approach was done in Liu et al. (2025a) and Zhang et al. (2025b). To extend beyond MCQ, Zhang et al. (2025b) introduced a different reward for different scenarios: allowable range for numerical value, exact match and rouge-l for open-ended QA. For the ranked list format, our work borrows the fundamental idea from the information

retrieval field–mean reciprocal rank (MRR)–which award higher score if the correct answer is higher in rank. We use MRR as reward to incentivize an LM to generate a ranked list answer using the RFT approach. We also study incorporating additional terms to penalize long list to investigate effects of reward on reasoning model performance and behaviors. In addition, we also experiment with using LM judge as a reward.

### A.4 MEDICAL EVALUATION

A growing body of work has explored how to evaluate LLMs in medical settings, often focusing on dataset construction, multimodal understanding, or complex interactive environments. Zhou et al. (2024) examines how to create grounded benchmark datasets leveraging existing medical knowledge bases. Unlike this line of work, we do not introduce a new dataset; rather, we study robustness to answer formats—especially ranked lists—and focus specifically on MRMs, not general-purpose medical LLMs. Lin et al. (2025) evaluates models on image-based tasks such as X-ray interpretation, ultrasound reading, and image reconstruction. These tasks require vision-language capabilities, which differ fundamentally from our focus on text-to-text MRMs. As a result, multimodal comprehension lies outside the scope of our work.

Another line of research studies *differential diagnosis* and *hierarchical evaluation*. H-DDx (Lim et al., 2025) introduces a pipeline that maps free-text diagnoses to ICD-10 codes and evaluates them using hierarchical metrics. Although we reference differential diagnosis as one potential use case for ranked lists, our goal differs: we investigate answer-format robustness rather than proposing new diagnostic frameworks or hierarchical metrics. Accordingly, our evaluation uses standard rank-aware metrics like MRR, which align with single-answer benchmarks.

A recent study has also explored more realistic or interactive evaluation settings. AI Hospital (Fan et al., 2025) evaluates LLMs in multi-agent, multi-turn clinical environments to simulate complex medical interactions. Similarly, Nori et al. (2025) studies iterative medical reasoning, where a model gathers information and revises its diagnosis across multiple turns. In contrast, our study is carried out in a single-turn, accuracy-focused setting designed to isolate the effects of answer format and training method on MRMs.

Finally, works such as Griot et al. (2025); Arora et al. (2025) evaluate higher-level clinical capabilities including metacognition and rubric-based reasoning. While we also use LLM judges in our evaluation, we follow a reference-based comparison between generated and ground-truth answers rather than rubric-based scoring. Overall, our work complements these efforts by offering the first systematic examination of how MRMs trained on one answer format generalize when instructed to produce another, with particular emphasis on the ranked-list format.

## B DATASET OVERVIEW

Table 2 summarizes the datasets used in this study, including both training data and evaluation benchmarks. We report the dataset splits, the number of instances, the task format (MCQ or QA), and the associated license terms. All datasets are publicly available under licenses that permit their use for research purposes.

The QA versions of the datasets are obtained through the conversion pipeline described in Appendix D. This process ensures consistency between the original MCQ-style questions and their QA counterparts.

The MCQ variants are used to measure baseline performance in the standard multiple-choice format, which remains a common evaluation protocol for medical reasoning models. The QA variants are used to evaluate models in an open-ended setting where answers are produced as ranked lists. In addition, we also employ QA benchmarks to evaluate free-form answers (without predefined options), which serve as a more challenging baseline for assessing model generalization.

AlphaMed and QA-AlphaMed are used in RFT experiments, while SFT-*-AlphaMed datasets are used in SFT experiments. Mixed-AlphaMed is used for the training experiments described in Appendix K. Each record in Mixed-AlphaMed is prepended with an appropriate prior prompt, depending on the record type and the experiment. Additional details on how these datasets are used in train-

| Dataset | Split | Count | Task | License |
|---|---|---|---|---|
| **Training datasets** | | | | |
| AlphaMed (Liu et al., 2025a) | Train | 19,778 | MCQ | MIT |
| QA-AlphaMed | Train | 14,382 | QA | Apache 2.0 |
| Mixed-AlphaMed | Train | 34,160 | Mixed | Apache 2.0 |
| SFT-MCQ-AlphaMed | Train | 16,591 | MCQ | Apache 2.0 |
| SFT-QA-AlphaMed | Train | 9,416 | QA | Apache 2.0 |
| SFT-List-AlphaMed | Train | 9,705 | QA | Apache 2.0 |
| **Evaluation benchmarks** | | | | |
| MedQA[2] (Jin et al., 2021) | Test | 1,273 | MCQ | MIT |
| QA-MedQA | Test | 1,233 | QA | Apache 2.0 |
| MedMCQA[3] (Pal et al., 2022) | Test | 6,150 | MCQ | Apache 2.0 |
| QA-MedMCQA | Test | 2,180 | QA | Apache 2.0 |
| MedXpertQA[4] (Zuo et al., 2025) | Test-Text | 2,450 | MCQ | MIT |
| QA-MedXpertQA | Test-Text | 2,086 | QA | Apache 2.0 |
| MMLU Pro[5] (Wang et al., 2024) | Test-Health | 818 | MCQ | MIT |
| QA-MMLU Pro | Test-Health | 736 | QA | Apache 2.0 |

Table 2: Overview of the training dataset and evaluation benchmarks. All datasets are publicly available under licenses that permit their use for this type of research.

ing and evaluation are provided in Section 2. The datasets are available at anonymous1entity/med-datasets and anonymous1entity/med-sft-datasets.

## C  MODEL OVERVIEW

Table 3 summarizes the proprietary and open-weight models evaluated in this study. The proprietary models (Gemini 2.5 family and GPT-4.1 Mini) do not disclose parameter counts, while the open-weight models span several major families, including Qwen, Gemma, MedGemma, OpenThinker, HuatuoGPT, m1, and AlphaMed. Importantly, all medical reasoning models sized at 7B parameters–namely HuatuoGPT o1 7B, m1 7B 23K, and AlphaMed 7B Instruct RL–are derived from Qwen2.5 7B Instruct, with additional post-training targeted to medical domains. In contrast, MedGemma 4B originates from Gemma 3 4B It.

## D  PIPELINE FOR CONVERTING MCQS TO QA FORMAT

This section describes the pipeline used to convert a dataset in MCQ format into QA format. Inspired by Myrzakhan et al. (2024), we design a prompt to determine whether a given question–choices–ground-truth triplet can be converted from MCQ to QA. First, we ask an LLM to reason thoroughly before giving a verdict on whether the question can be converted. If the LLM deems the conversion possible, it generates a QA-style question that yields the same ground-truth answer.

All of this happens in a single LLM call, since modern models show significant improvements in performance, which reduces the need for a separate two-stage process. This simplification also reduces the cost of conversion. We also ask the model to output confidence scores for further use in the filtration process. We note that the number of resulting QA questions differs from the original number of MCQs, as some questions may not be entirely suitable for conversion and are therefore excluded. We use `gpt-4.1-mini-2025-04-14` as the generation model with default sampling parameters, except for the temperature, which we set to 0.1. The prompt used for conversion appears in Figure 3. This pipeline applies to both the training dataset (AlphaMed) and the benchmarks used in this study.

Your task is to review a multiple–choice question, its answer choices, and the ground truth. Determine if, after possible revision (including adding clarifying information), the question can be answered correctly **without** the answer choices––as a standalone, open–ended question.

* For incomplete–sentence questions (e.g., "During swallowing, ..."), use your knowledge to complete the sentence accurately.
* For identification questions (e.g., "Which of the following structures is part of the small intestine?"), consider if the question can be revised so an informed respondent could answer it without choices.
* The revised question MUST be specific enough so that the answer can be determined without ambiguity, and it MUST BE the ground truth.

If, even after revision, the question cannot be answered confidently without the choices, return **"NO"**. If unsure, default to **"NO"**. Only return **"YES"** if you are confident the revised question can be answered independently.

**Instructions:**

1. Reason through your decision inside '<think>' and '</think>' tags.
2. Output your verdict––**only** "YES" or "NO"––inside '<verdict>' and '</verdict>' tags. Nothing else should appear within '<verdict>'.
3. If "YES", provide your revised version of the question inside '<revised_question></revised_question>'.
4. Finally, rate your confidence that this revised question can be answered in close–ended QA format (1 = lowest, 5 = highest) inside '<confidence></confidence>'.

–––

**Question**
{question}

**Choices**
{choices}

**Ground truth**
{ground_truth}

Figure 3: Prompt used for converting MCQ questions into their equivalent QA format.

| Model | Access | Size | Category |
|---|---|---|---|
| Gemini 2.5 Flash Lite[6] (Comanici et al., 2025) | P | N/A | GRM |
| Gemini 2.5 Flash[7] (Comanici et al., 2025) | P | N/A | GRM |
| Gemini 2.5 Pro[8] (Comanici et al., 2025) | P | N/A | GRM |
| GPT-4.1 Mini[9] (OpenAI, 2025) | P | N/A | GLM |
| Qwen2.5 3B Instruct[10] (Qwen et al., 2025) | OW | 3B | GLM |
| Qwen2.5 7B Instruct[11] (Qwen et al., 2025) | OW | 7B | GLM |
| Qwen2.5 14B Instruct[12] (Qwen et al., 2025) | OW | 14B | GLM |
| Qwen3 4B Instruct 2507[13] (Yang et al., 2025) | OW | 4B | GLM |
| Gemma 3 4B IT[14] (Team et al., 2025) | OW | 4B | GLM |
| MedGemma 4B IT[15] (Sellergren et al., 2025) | OW | 4B | MLM |
| MedGemma 27B IT[16] (Sellergren et al., 2025) | OW | 27B | MLM |
| OpenThinker 3 7B[17] (Guha et al., 2025) | OW | 7B | GRM |
| HuatuoGPT o1 7B[18] (Chen et al., 2024) | OW | 7B | MRM |
| m1 7B 23K[19] (Huang et al., 2025a) | OW | 7B | MRM |
| AlphaMed 7B Instruct RL[20] (Liu et al., 2025a) | OW | 7B | MRM |

Table 3: Overview of **proprietary (P)** and **open-weight (OW)** models used in this study, categorized as **general reasoning models (GRM)**, **medical reasoning models (MRM)**, **general-purpose LLMs (GLM)**, or **medical-specialized LLMs (MLM)**.

# E  EVALUATION METRICS

## E.1  ADDITIONAL DETAILS ON METRICS

For the MCQ ($\text{Acc}_{\text{MCQ}}$) setting, accuracy is computed by exact match between the predicted choice and the ground truth. For the QA ($\text{Acc}_{\text{QA}}$) setting, we use normalized (lowercasing) exact match between the extracted answer and the ground truth to obtain accuracy. For the ranked-list setting, we report both accuracy ($\text{Acc}_{\text{List}}$)–whether the ground-truth answer appears anywhere in the list– and MRR ($\text{MRR}_{\text{List}}$), which additionally accounts for the position of the correct answer, assigning higher scores when it appears earlier in the list.

Since models may produce correct answers that do not exactly match the ground-truth string, we also utilize LLM-based evaluation variants for non-MCQ formats: LLM-Acc for QA ($\text{Acc}_{\text{QA}}^{\text{LLM}}$) and ranked lists ($\text{Acc}_{\text{List}}^{\text{LLM}}$) and LLM-MRR ($\text{MRR}_{\text{List}}^{\text{LLM}}$) for ranked lists. These provide more flexible judgments of correctness and complement the exact-match metrics. Reporting both exact-match and LLM-based metrics provides a more nuanced perspective on training effects and generalization.

## E.2  LLM-BASED METRICS

There are three LLM-based metrics utilized in this study: LLM-Acc for QA answers, LLM-Acc for list answers, and LLM-MRR for list answers. These LLM-based metrics can capture semantically correct answers that differ in surface form, our hypothesis is that effective training should already improve performance under strict exact-match evaluation, even without relying on the more forgiving LLM-based measures. All LLM-based metrics use `gpt-4.1-mini-2025-04-14` as the judge. The prompt used for LLM-Acc on QA answers is shown in Figure 4, while the prompt for LLM-MRR on list answers is shown in Figure 5.

These prompts incorporate validation rules to guard against attempts to game the judge by producing nonsensical answers. We also instruct the model to respond in a predefined format to facilitate answer extraction. All evaluations are performed with the default sampling temperature, except when explicitly set to 0.0.

We note that LLM-Acc is derived from the results of the LLM-MRR judge. In particular, an LLM receives an LLM-Acc score of 1.0 for a given question if the correct answer appears in the output,

You are evaluating whether predicted answers match the ground truth answer semantically, even if they are not exactly the same text.

Ground Truth Answer: "{ground_truth}"
Predicted Answers: {predicted_text}

IMPORTANT VALIDATION RULES:
1. Each predicted answer must contain EXACTLY ONE focused response
2. If any answer contains multiple distinct answers, options, or attempts to cover multiple possibilities (e.g., using "or", "and", commas to separate different answers, bullet points, or lists), that answer is INVALID
3. Too long answers will be considered invalid gaming attempts
4. Only evaluate answers that contain single, focused responses
5. Answers containing repeated words, phrases, or synonyms (e.g., "pneumonia pneumonia", "MI heart attack myocardial infarction", "diabetes DM diabetes mellitus") are INVALID
6. More than 2 occurrences of the same root word or concept in a single answer is INVALID
7. Overly broad terms that could match multiple conditions (e.g., "infection", "cardiac event", "abnormal values", "inflammatory condition") are INVALID unless they precisely match the ground truth
8. Use of special characters, symbols, or separators to bypass comma rules (2.) (e.g., "|", "/", ":", ";", unicode spaces) is INVALID
9. Incomplete answers containing only part of medical terms or abbreviations without full context are INVALID
10. Answers containing non−medical terms, gibberish, or obvious filler content are INVALID
11. If multiple answers contain very similar but slightly different versions of the same answer, only the first valid answer should be considered
12. Medically impossible or contradictory answers indicate INVALID
13. The same medical concept appearing in multiple answers with slight variations is INVALID

Task: Determine if ANY of the predicted answers is semantically equivalent to the ground truth answer. Consider:
− Different phrasings that mean the same thing
− Medical terminology and abbreviations
− Minor spelling variations
− Synonyms and equivalent expressions
− Clinical accuracy and meaning
− Answers must be clinically specific enough to distinguish from other conditions
− Abbreviations must be standard medical abbreviations, not made−up shortcuts
− Technical terms must be used correctly in proper medical context
− Synonyms must be legitimate medical synonyms, not loose associations

RESPONSE FORMAT:
Provide your reasoning and analysis before giving the final verdict. Keep it short and concise.
Verdict: [Provide ONLY "MATCH" if any predicted answer is semantically equivalent to the ground truth, or "NO_MATCH" if none are equivalent or if gaming behavior is detected]

Figure 4: Prompt used for evaluating LLM-Acc on QA answers.

You are evaluating which predicted answer (if any) matches the ground truth answer semantically.

Ground Truth Answer: "{ground_truth}"
Predicted Answers (in rank order):
{predicted_list}

IMPORTANT VALIDATION RULES:
1. Each rank position must contain EXACTLY ONE focused answer
2. If any rank contains multiple distinct answers, options, or attempts to cover multiple possibilities (e.g., using "or", "and", commas to separate different answers, bullet points, or lists), that rank is INVALID
3. Too long answers will be considered invalid gaming attempts
4. Only evaluate ranks that contain single, focused responses
5. Ranks containing repeated words, phrases, or synonyms (e.g., "pneumonia pneumonia", "MI heart attack myocardial infarction", "diabetes DM diabetes mellitus") are INVALID
6. More than 2 occurrences of the same root word or concept in a single rank is INVALID
7. Overly broad terms that could match multiple conditions (e.g., "infection", "cardiac event", "abnormal values", "inflammatory condition") are INVALID unless they precisely match the ground truth
8. Use of special characters, symbols, or separators to bypass comma rules (2.) (e.g., "|", "/", ":", ";", unicode spaces) is INVALID
9. Incomplete answers containing only part of medical terms or abbreviations without full context are INVALID
10. Answers containing non−medical terms, gibberish, or obvious filler content are INVALID
11. If lower ranks (2, 3, 4, etc.) contain obviously incorrect answers that appear designed to make rank 1 look better by comparison, treat as INVALID
12. If multiple ranks contain very similar but slightly different versions of the same answer, return the first valid rank
13. Medically impossible or contradictory answers across ranks indicate INVALID
14. The same medical concept appearing in multiple ranks with slight variations is INVALID

Task: Determine which predicted answer number (1, 2, 3, etc.) is semantically equivalent to the ground truth answer. Consider:
− Different phrasings that mean the same thing
− Medical terminology and abbreviations
− Minor spelling variations
− Synonyms and equivalent expressions
− Clinical accuracy and meaning
− Answers must be clinically specific enough to distinguish from other conditions
− Abbreviations must be standard medical abbreviations, not made−up shortcuts
− Technical terms must be used correctly in proper medical context
− Synonyms must be legitimate medical synonyms, not loose associations

RESPONSE FORMAT:
Provide your reasoning and analysis before giving the final verdict. Keep it short and concise.
Verdict: [Provide ONLY the number of the first valid matching answer (e.g., "1", "2", "3", etc.), or "NO_MATCH" if none are equivalent or if gaming behavior is detected]

Figure 5: Prompt used for evaluating LLM-MRR on list answers.

You are evaluating which predicted answer (if any) matches the ground truth answer semantically.

Ground Truth Answer: "{ground_truth}"

Predicted Answers (in rank order):
{predicted_list}

Task: Determine which predicted answer number (1, 2, 3, etc.) is semantically equivalent to the ground truth answer. Consider:
– Different phrasings that mean the same thing
– Medical terminology and abbreviations
– Minor spelling variations
– Synonyms and equivalent expressions
– Clinical accuracy and meaning

Respond with ONLY the number of the first matching answer (e.g., "1", "2", "3", etc.), or "NO_MATCH" if none are equivalent.

Figure 6: A simpler judge prompt used for an ablation study in Section 4.3.1.

You are a helpful and harmless expert clinical assistant. The assistant provides the user with the accurate answer. When you finally reach a conclusion, clearly state the final answer in \boxed{}. You always begins your answer with the choice, e.g., A., B., C., D., E. in the \boxed{}. Now the user asks you to solve a problem.

{query}
{choices}

Figure 7: Zero-shot prompt template for multiple-choice questions (MCQ).

regardless of its rank. This contrasts with LLM-MRR, where the rank assigned by the judge is also taken into account when computing the reward.

The prompt for LLM-MRR is also used as a judge prompt during RFT in Section 4.2 as well. We also have a simpler version of this judge prompt, which is Figure 6, used for an ablation study in Section 4.3.1.

## F    PROMPT TEMPLATES

We design a total of six prompt templates: MCQ, MCQ-CoT, QA, QA-CoT, List, and List-CoT. Our templates are adapted from the prior-prompt approach introduced by Xie et al. (2025), with modifications to better suit the medical domain. Specifically, we adjust the role prompt and, in the MCQ-CoT variant, explicitly require the model to begin its final answer with the selected choice marker (e.g., A, B, C). The other answer formats reuse the same general template with instructions tailored to QA or list-style outputs. For the list format, we additionally provide a one-shot example to illustrate the expected output structure.

For zero-shot variants, we remove the reasoning instruction and omit the `<think>` and `</think>` tags, leaving only the answer-format instruction and general components. All prompting experiments are run with consistent decoding parameters across models to ensure comparability: `temperature=0.0`, `top_p=1.0`, and `top_k=-1`. We set `max_tokens=8192` for most models, but increase this to `16384` for Gemini 2.5 models to accommodate their typically more verbose reasoning chains. The complete set of prompt templates used in our experiments is shown in Figures 7 to 12.

ou are a helpful and harmless expert clinical assistant. The assistant first thinks about the reasoning process in the mind and then provides the user with the accurate answer. The reasoning process is enclosed within <think></think> tags followed by an answer, i.e., <think> reasoning process here </think> answer here. After thinking, when you finally reach a conclusion, clearly state the final answer in \boxed{}. You always begins your answer with the choice, e.g., A., B., C., D., E. in the \boxed{}. Now the user asks you to solve a problem.

{query}
{choices}

Figure 8: CoT prompt template for multiple-choice questions (MCQ-CoT).

You are a helpful and harmless expert clinical assistant. The assistant provides the user with the accurate answer. Now the user asks you to solve a problem. When you finally reach a conclusion, clearly state the final answer in \boxed{}.

{query}

Figure 9: Zero-shot prompt template for open-ended QA.

You are a helpful and harmless expert clinical assistant. The assistant first thinks about the reasoning process in the mind and then provides the user with the accurate answer. The reasoning process is enclosed within <think></think> tags followed by an answer, i.e., <think> reasoning process here </think> answer here. After thinking, when you finally reach a conclusion, clearly state the final answer in \boxed{}. Now the user asks you to solve a problem.

{query}

Figure 10: CoT prompt template for open-ended QA (QA-CoT).

You are a helpful and harmless expert clinical assistant. The assistant provides the user with an accurate answer. When you finally reach a conclusion, clearly list all possible answers in order from most likely to least likely. Start with "# Final Answer" followed by numbered lines using the format '
n. answer' for each answer. Each item MUST contain only the answer without any explanation or reasoning.

Example:
# Final Answer
1. xxx
2. xxx

Now the user asks you to solve a problem.

{query}

Figure 11: Zero-shot prompt template for list-style answers.

You are a helpful and harmless expert clinical assistant. The assistant first thinks about the reasoning process and then provides the user with an accurate answer. The reasoning process is enclosed within <think></think> tags followed by an answer, i.e., <think>reasoning process here</think> answer here. After thinking, when you finally reach a conclusion, clearly list all possible answers in order from most likely to least likely. Start with "# Final Answer" followed by numbered lines using the format 'n. answer' for each answer. Each item MUST contain only the answer without any explanation or reasoning.

Example:
<think>...</think>

# Final Answer
1. xxx
2. xxx

Now the user asks you to solve a problem.

{query}

Figure 12: CoT prompt template for list-style answers (List-CoT).

You are a medical validation expert. Your task is to validate whether a medical response contains the correct answer.
Given:
– Correct Answer: <CORRECT_ANSWER>
– Generated Response: <RESPONSE>

Please determine if the generated response contains or aligns with the correct answer. Consider:
1. For MCQ questions with option letters (A, B, C, D, E): Check if the response contains the correct option letter, and optional answer
2. For other questions: Check if the response mentions the correct answer explicitly or implicitly

Respond with only "VALID" if the response contains the correct answer, or "INVALID" if it does not.

Figure 13: Validation prompt used for multiple-choice (MCQ) and multiple-answer (MQA) formats.

## G  SFT TRAINING DATASET PREPARATION

We construct the SFT training dataset through knowledge distillation from Qwen3-30B-A3B-Thinking-2507-FP8[21] (Yang et al., 2025). Specifically, we provide questions from AlphaMed and AlphaMedQA, coupled with the CoT variants of each prompt template corresponding to the answer format under consideration.

To ensure correctness, we apply rejection sampling using an LLM judge, gpt-4o-mini-2024-07-18. For MCQ and MQA responses, we use the validation prompt shown in Figure 13, while list answers are validated with the prompt in Figure 14. We use a sampling temperature of 0.7, a maximum token length of 8192, and allow up to 20 retries for incorrect responses. Responses that remain incorrect after rejection sampling are discarded. The filtered records are retained and used to train the distilled SFT models. Additional details on training are provided in Appendix H.1.

---

[21]https://huggingface.co/Qwen/Qwen3-30B-A3B-Thinking-2507-FP8

You are a medical validation expert. Your task is to validate whether a medical response with a list format contains the correct answer.
Given:
– Correct Answer: <CORRECT_ANSWER>
– Generated Response: <RESPONSE>

The generated response should contain a numbered list of possible answers. Please determine if the correct answer appears anywhere in this list. Consider:
1. The correct answer may appear as an exact match in one of the list items
2. The correct answer may appear with slight variations or paraphrasing
3. Look for the answer in the "# Final Answer" section with numbered items

Respond with only "VALID" if the correct answer appears in the generated list, or "INVALID" if it does not.

Figure 14: Validation prompt used for list-style answers.

## H  TRAINING HYPERPARAMETERS

In this section, we describe the training hyperparameters used in our experiments. All training experiments used about 815 GPU hours on a 4xH100 node.

### H.1  SFT

We use LLaMA-Factory[22] (Zheng et al., 2024) v0.9.3, which is released under the Apache 2.0 license. For SFT, we perform full fine-tuning with DeepSpeed ZeRO Stage 3 (Rajbhandari et al., 2020). Training is conducted with a per-device batch size of 2 and a gradient accumulation step of 4, resulting in an effective batch size of 8. We use a learning rate of $1 \times 10^{-5}$ for 2 epochs with a cosine learning rate scheduler. Training is performed with `bf16` precision, and FlashAttention-2 is enabled to improve efficiency. We set the warmup ratio to 0.05. The training datasets are the SFT-*-AlphaMed variants listed in Table 2. Each variant is used to train one model, resulting in three models in total, corresponding to the three answer formats investigated in this study.

### H.2  RFT

We use verl[23] (Sheng et al., 2025) v0.5.0, released under the Apache 2.0 license. Training is performed with full fine-tuning (no offloading) using FSDP2 (Zhao et al., 2023) as the backend and group relative policy optimization (GRPO) (Shao et al., 2024) without a KL regularization term (Liu et al., 2025b; Xie et al., 2025).

We train with a batch size of 256, divided into 64 mini-batches. The maximum prompt length is 2048 tokens, and the maximum response length is 4096 tokens, constrained by available compute. The learning rate is set to $1 \times 10^{-6}$, with padding removed and gradient checkpointing enabled. Torch compile is also enabled for efficiency.

For rollouts, we use vLLM[24] (Kwon et al., 2023). Log-probability computation is performed with a micro-batch size of 8 per GPU. For each prompt, we sample 8 responses with the default verl parameters. Dynamic batching is enabled for greater efficiency, targeting a maximum of 24,576 tokens per GPU for the actor, reference, and rollout models. We train for 2 epochs with no warmup. The initial model, training set, and reward function are selected according to the configuration of each experiment. The reward function code is available in the repository.

---

[22]https://github.com/hiyouga/LLaMA-Factory
[23]https://github.com/volcengine/verl
[24]https://github.com/vllm-project/vllm

Table 4: Average response length (mean ± standard deviation) for MCQ, QA, and list-based answer formats across benchmarks. Note that **LL** denotes a list length, i.e., the mean number of items across lists, including empty lists (items = 0.)

| | MCQ | QA | List | CP | LL | VLL |
|---|---|---|---|---|---|---|
| *Proprietary Models* | | | | | | |
| Gemini 2.5 Flash Lite | $1585 \pm 3720$ | $511 \pm 997$ | $93 \pm 735$ | 1.39 | 2.77 | 2.86 |
| +CoT | $4360.07 \pm 5704$ | $2850 \pm 4535$ | $7398 \pm 8135$ | 1.17 | 0.87 | 1.92 |
| Gemini 2.5 Flash | $473 \pm 251$ | $273 \pm 220$ | $32 \pm 33$ | 1.41 | 2.98 | 3.00 |
| +CoT | $1759 \pm 1367$ | $1576 \pm 1036$ | $1021 \pm 857$ | 1.30 | 1.16 | 2.66 |
| Gemini 2.5 Pro | $425 \pm 188$ | $459 \pm 289$ | $38 \pm 55$ | 1.40 | 3.38 | 3.41 |
| +CoT | $1326 \pm 326$ | $1527 \pm 568$ | $1019 \pm 282$ | 1.41 | 3.29 | 3.46 |
| GPT-4.1 Mini | $353 \pm 156$ | $285 \pm 158$ | $137 \pm 86$ | 1.36 | 3.20 | 3.26 |
| +CoT | $357 \pm 122$ | $292 \pm 110$ | $252 \pm 88$ | 1.36 | 3.71 | 3.72 |
| *Open-weight Models* | | | | | | |
| Qwen2.5 3B Instruct | $271 \pm 221$ | $214 \pm 254$ | $36 \pm 32$ | 1.69 | 3.09 | 3.09 |
| +CoT | $361 \pm 305$ | $315 \pm 385$ | $167 \pm 234$ | 1.27 | 1.78 | 1.80 |
| Qwen2.5 7B Instruct *(our initial model)* | $72 \pm 68$ | $146 \pm 138$ | $27 \pm 18$ | 1.45 | 2.39 | 2.39 |
| +CoT | $2393 \pm 3355$ | $196 \pm 125$ | $4434 \pm 3991$ | 1.91 | 184.19 | 185.72 |
| Qwen2.5 14B Instruct | $129 \pm 72$ | $140 \pm 124$ | $35 \pm 21$ | 1.48 | 3.04 | 3.05 |
| +CoT | $217 \pm 123$ | $213 \pm 237$ | $163 \pm 196$ | 1.30 | 2.36 | 2.36 |
| Qwen3 4B Instruct 2507 | $757 \pm 867$ | $450 \pm 660$ | $69 \pm 214$ | 1.70 | 3.96 | 3.96 |
| +CoT | $894 \pm 929$ | $538 \pm 646$ | $458 \pm 808$ | 1.50 | 3.32 | 3.56 |
| Gemma 3 4B IT | $400 \pm 112$ | $432 \pm 200$ | $61 \pm 135$ | 1.83 | 4.59 | 4.68 |
| +CoT | $382 \pm 669$ | $304 \pm 163$ | $251 \pm 527$ | 1.67 | 3.43 | 4.13 |
| MedGemma 4B IT | $297 \pm 809$ | $152 \pm 447$ | $951 \pm 2554$ | 2.15 | 95.01 | 95.06 |
| +CoT | $7958 \pm 1141$ | $493 \pm 1445$ | $8180 \pm 309$ | 3.14 | 419.18 | 482.76 |
| MedGemma 27B IT | $1081 \pm 847$ | $653 \pm 636$ | $52 \pm 214$ | 1.46 | 3.20 | 3.26 |
| +CoT | $1424 \pm 878$ | $966 \pm 949$ | $1016 \pm 1050$ | 1.43 | 3.81 | 3.88 |
| OpenThinker3 7B | $4789 \pm 2695$ | $4348 \pm 2844$ | $4744 \pm 2982$ | 1.43 | 2.06 | 3.06 |
| +CoT | $7340 \pm 2139$ | $4450 \pm 2862$ | $7630 \pm 1729$ | 1.29 | 0.23 | 4.76 |
| HuatuoGPT o1 7B | $472 \pm 169$ | $492 \pm 260$ | $47 \pm 224$ | 1.70 | 2.80 | 4.39 |
| +CoT | $488 \pm 210$ | $501 \pm 202$ | $375 \pm 272$ | 1.46 | 0.03 | 2.45 |
| m1 7B 23K | $1578 \pm 2203$ | $1542 \pm 1967$ | $1657 \pm 2930$ | 2.01 | 13.39 | 13.39 |
| +CoT | $8185 \pm 202$ | $2328 \pm 2423$ | $8149 \pm 542$ | 1.64 | 14.62 | 19.03 |
| AlphaMed 7B Instruct RL | $311 \pm 187$ | $216 \pm 275$ | $19 \pm 63$ | 1.83 | 1.93 | 2.59 |
| +CoT | $416 \pm 480$ | $266 \pm 607$ | $1052 \pm 2503$ | 1.89 | 47.86 | 55.74 |
| *Our Knowledge-Distilled Medical Reasoning Models (based on Qwen2.5 7B Instruct)* | | | | | | |
| SFT-MCQ | $2749 \pm 1626$ | $1419 \pm 1204$ | $2438 \pm 3310$ | 2.83 | 141.11 | 141.72 |
| +CoT | $2643 \pm 1577$ | $1390 \pm 1203$ | $1671 \pm 1277$ | 1.09 | 1.44 | 1.46 |
| SFT-QA | $2365 \pm 1646$ | $1235 \pm 1045$ | $1758 \pm 2355$ | 1.81 | 13.50 | 13.85 |
| +CoT | $2425 \pm 1605$ | $1388 \pm 1136$ | $15312 \pm 1172$ | 1.15 | 1.04 | 1.78 |
| SFT-List | $2856 \pm 1614$ | $1287 \pm 1184$ | $1281 \pm 992$ | 1.41 | 2.50 | 2.52 |
| +CoT | $2776 \pm 1591$ | $1333 \pm 1153$ | $1425 \pm 1112$ | 1.42 | 2.55 | 2.57 |
| *Our RFT Medical Reasoning Models (based on Qwen2.5 7B Instruct)* | | | | | | |
| RFT-MCQ | $204 \pm 106$ | $167 \pm 65$ | $33 \pm 123$ | 1.45 | 2.29 | 2.29 |
| RFT-QA | $296 \pm 302$ | $291 \pm 247$ | $46 \pm 91$ | 1.67 | 0.29 | 3.01 |
| RFT-List-Acc | $208 \pm 231$ | $195 \pm 433$ | $165 \pm 177$ | 2.07 | 5.94 | 5.94 |
| RFT-List-MRR | $174 \pm 110$ | $157 \pm 207$ | $319 \pm 1154$ | 2.11 | 16.97 | 16.97 |
| RFT-List-Judge-MRR | $168 \pm 376$ | $193 \pm 622$ | $133 \pm 214$ | 1.64 | 4.45 | 4.46 |

# I  ADDITIONAL DISCUSSIONS

In this section, we provide an additional discusson on results presented in the main paper.

## I.1  PROMPTING RESULTS

**MCQ vs. QA**   For HuatuoGPT-o1 and AlphaMed, the drop is expected, as both were trained specifically on MCQ. Interestingly, this is not the case for m1, which was also trained on MCQ data for medical benchmarks. Although HuatuoGPT-o1 and m1 used comparable training data sizes ($\approx$20K samples), their training paradigms differ. Both HuatuoGPT-o1 and m1 were trained with SFT, whereas AlphaMed was trained with RFT. The key distinction between HuatuoGPT-o1 and m1 is that the former relied on synthetic reasoning trajectories, while m1 utilized distilled trajectories from a large reasoning model.

**Non-MCQ formats are unfamiliar to the models**   We conduct a Wilcoxon signed-rank test[25] comparing MCQ with other formats across benchmarks, prompting strategies, and models. The results show that changing the answer format from *MCQ* to *QA* ($p = 4.26 \times 10^{-4}$, $r = 0.317$) significantly alters model performance, with a small-to-moderate effect size. The effect is even

---

[25]We found evidence against normality for both pairs using the Shapiro–Wilk test ($p = 0.000614$ for *MCQ* vs. *QA* and $p = 0.0016$ for *MCQ* vs. *List*). Consequently, we employed the Wilcoxon signed-rank test.

stronger for *MCQ* to *ranked-list* ($p = 9.38 \times 10^{-16}$, $r = 0.667$), indicating a large effect. We attribute this to the fact that the majority of current medical benchmarks are available in MCQ format (e.g., (Jin et al., 2021; Pal et al., 2022; Zuo et al., 2025; Wang et al., 2024)), and models trained to excel in this setting may have learned to associate the answer format with knowledge (Li et al., 2024; Singh et al., 2025). Consequently, changing the answer format leads to substantial performance differences, particularly for the ranked-list format, which is less common and thus less familiar to models.

**Do longer responses lead to better performance?** We observe from Table 4 that the majority of reasoning models produce longer responses than standard LLMs. While this trend holds for most reasoning models (for example, OpenThinker3 and m1 generate long responses of around 4K and 1K tokens, respectively, regardless of answer format), AlphaMed is an exception. AlphaMed is the only open-weight medical reasoning model trained with RFT rather than SFT. We further discuss the impact of RFT on response length in Section 4.2.

Statistical testing using Pearson's correlation between model score and mean response length across all prompting variants, benchmarks, metrics, answer formats, and models included in this experiment revealed a small but statistically significant negative correlation ($r = -0.144$, $p = 7.4 \times 10^{-6}$, $|r| = 0.144$). These results indicate that performance is negatively, but only weakly, associated with response length. In other words, producing longer responses weakly and negatively affects performance.

Therefore, the premise that reasoning models always produce longer answers (Guo et al., 2025; Muennighoff et al., 2025) and that longer responses signal greater performance (Guo et al., 2025) does not hold in our setting. In fact, prior work on efficient reasoning (Sui et al., 2025) suggests that various training techniques can encourage concise reasoning chains while maintaining high performance. This further supports the conclusion that response length is a poor indicator of final model performance.

## I.2 FINE-TUNING RESULTS

## I.3 SFT RESULTS

As shown in Table 4, models trained with SFT in any format consistently produce longer responses (around 1K–3K tokens). These findings align with what was observed with m1, another knowledge-distilled medical reasoning model from prior work, discussed in Section 3.

## I.4 RFT RESULTS

**RFT does not always incentivize long responses** Models trained with RFT are surprisingly concise, often producing shorter answers than the initial model prompted with CoT, similar to AlphaMed and HuatouGPT o1. We conjecture that RFT primarily incentivizes models to make the most effective use of their intermediate generated tokens to maximize performance, rather than encouraging longer responses. However, longer responses may still correlate with higher accuracy in certain scenarios.

## I.5 ABLATION STUDIES FOR RFT

### I.5.1 RFT FACTORS

Tables 5 and 6 present the performance and list metrics for the experiments in Section 4.3.1. In the no-prior-prompt setting, models tend to generate longer responses (e.g., RFT-MCQ averages 204 → 474 tokens), whereas responses are slightly shorter under the no-CoT-prompt setting (e.g., 204 → 129 tokens). By contrast, both judge models yield similar list behaviors and training dynamics, such as average response lengths around 168–239 tokens for MCQ and 133–141 tokens for list outputs.

Effects of Prior Prompts To account for changes in prior prompts, we adjust our setup when removing prior prompts or parts of them. Specifically, in the no-prompt and no-CoT settings, we exclude the format component from the reward function, since the absence of explicit thinking tags would otherwise drive it toward zero.

Table 5: Performance results of the ablation study on factors affecting RFT. The focus is on the reward component in the reward function, extended training duration, and the effects of prior prompts across models. Rw.Fn. denotes Reward Function.

| | Prior Prompt | Rw.Fn. | MCQ $\text{Acc}_{\text{MCQ}}$ | QA $\text{Acc}_{\text{QA}}$ | $\text{Acc}_{\text{QA}}^{\text{LLM}}$ | List $\text{Acc}_{\text{List}}$ | $\text{Acc}_{\text{List}}^{\text{LLM}}$ | $\text{MRR}_{\text{List}}$ | $\text{MRR}_{\text{List}}^{\text{LLM}}$ |
|---|---|---|---|---|---|---|---|---|---|
| **RFT-MCQ** | MCQ-CoT | $\text{Acc}_{\text{MCQ}}$ | 39.34 | 9.96 | **46.33** | 9.89 | 40.06 | 8.16 | 33.00 |
| No format reward | MCQ-CoT | $\text{Acc}_{\text{MCQ-NF}}$ | 39.56 | 9.48 | 46.24 | 10.01 | 37.91 | 9.19 | 33.47 |
| 4 Epochs | MCQ-CoT | $\text{Acc}_{\text{MCQ}}$ | **39.97** | 9.78 | 45.07 | 9.91 | 35.46 | 9.35 | 32.20 |
| No prompt | ✗ | $\text{Acc}_{\text{MCQ-NF}}$ | 38.95 | 9.91 | 46.12 | 10.32 | 41.94 | 9.25 | 36.49 |
| MCQ prompt | MCQ | $\text{Acc}_{\text{MCQ-NF}}$ | 39.80 | 9.21 | 44.59 | 10.04 | 38.78 | 8.91 | 32.73 |
| **RFT-QA** | QA-CoT | $\text{Acc}_{\text{QA}}$ | 36.80 | 1.04 | 25.22 | 0.62 | 3.59 | 0.48 | 2.82 |
| No prompt | ✗ | $\text{Acc}_{\text{QA-NF}}$ | 27.76 | 4.08 | 29.17 | 9.75 | 41.92 | 8.32 | 34.75 |
| QA prompt | QA | $\text{Acc}_{\text{QA-NF}}$ | 28.23 | 0.95 | 24.78 | 11.42 | 49.93 | 9.06 | 38.35 |
| **RFT-List-Acc** | List-CoT | $\text{Acc}_{\text{List}}$ | 22.40 | 4.28 | 19.01 | 22.11 | 56.61 | **16.17** | 40.26 |
| List prompt | List | $\text{Acc}_{\text{List-NF}}$ | 12.97 | 9.43 | 44.46 | **24.16** | 67.08 | 13.83 | 37.13 |
| **RFT-List-MRR** | List-CoT | $\text{MRR}_{\text{List}}$ | 18.23 | 4.43 | 21.90 | 20.96 | 61.60 | 15.83 | 44.89 |
| List prompt | List | $\text{MRR}_{\text{List-NF}}$ | 10.41 | **10.37** | 46.21 | 22.06 | 63.00 | 15.28 | 41.24 |
| **RFT-List-Judge-MRR** | List-CoT | $\text{MRR}_{\text{List}}^{\text{LLM}}$ | 20.49 | 6.49 | 30.36 | 14.86 | 60.90 | 12.16 | **48.68** |
| Gemini judge | List-CoT | $\text{MRR}_{\text{List}}^{\text{LLM-Gemini}}$ | 33.11 | 9.40 | 43.16 | 13.07 | 59.34 | 10.95 | 48.00 |
| Simple judge prompt | List-CoT | $\text{MRR}_{\text{List}}^{\text{LLM-Simple}}$ | 19.02 | 4.37 | 27.27 | 4.75 | 31.84 | 3.86 | 26.19 |

Table 6: Average response length (mean $\pm$ standard deviation) for MCQ, QA, and list-based answer formats across benchmarks and metrics related to the ranked list answer format from the generated evaluation responses for the RFT factors ablation study.

| | Prior Prompt | Rw.Fn. | MCQ | QA | List | CP | LL | VLL |
|---|---|---|---|---|---|---|---|---|
| **RFT-MCQ** | MCQ-CoT | $\text{Acc}_{\text{MCQ}}$ | $204 \pm 106$ | $167 \pm 65$ | $33 \pm 123$ | 1.45 | 2.29 | 2.29 |
| No format reward | MCQ-CoT | $\text{Acc}_{\text{MCQ-NF}}$ | $178 \pm 138$ | $159 \pm 57$ | $129 \pm 114$ | 1.31 | 1.96 | 1.97 |
| 4 Epochs | MCQ-CoT | $\text{Acc}_{\text{MCQ}}$ | $336 \pm 107$ | $211 \pm 138$ | $178 \pm 141$ | 1.24 | 1.74 | 1.74 |
| No prompt | ✗ | $\text{Acc}_{\text{MCQ-NF}}$ | $474 \pm 547$ | $322 \pm 388$ | $147 \pm 196$ | 1.34 | 2.23 | 2.23 |
| MCQ prompt | MCQ | $\text{Acc}_{\text{MCQ-NF}}$ | $129 \pm 112$ | $121 \pm 79$ | $24 \pm 16$ | 1.41 | 2.14 | 2.15 |
| **RFT-QA** | QA-CoT | $\text{Acc}_{\text{QA}}$ | $296 \pm 302$ | $291 \pm 247$ | $46 \pm 91$ | 1.67 | 0.29 | 3.01 |
| No prompt | ✗ | $\text{Acc}_{\text{QA-NF}}$ | $259 \pm 378$ | $414 \pm 1015$ | $156 \pm 288$ | 1.50 | 2.80 | 2.97 |
| QA prompt | QA | $\text{Acc}_{\text{QA-NF}}$ | $130 \pm 183$ | $158 \pm 207$ | $43 \pm 22$ | 1.68 | 3.48 | 3.48 |
| **RFT-List-Acc** | List-CoT | $\text{Acc}_{\text{List}}$ | $208 \pm 231$ | $195 \pm 433$ | $165 \pm 177$ | 2.07 | 5.94 | 5.94 |
| List prompt | List | $\text{Acc}_{\text{List-NF}}$ | $85 \pm 114$ | $144 \pm 194$ | $7864 \pm 1551$ | 5.57 | 615.17 | 615.17 |
| **RFT-List-MRR** | List-CoT | $\text{MRR}_{\text{List}}$ | $174 \pm 110$ | $157 \pm 207$ | $319 \pm 1154$ | 2.11 | 16.97 | 16.97 |
| List prompt | List | $\text{MRR}_{\text{List-NF}}$ | $28 \pm 75$ | $100 \pm 150$ | $79 \pm 235$ | 2.59 | 9.51 | 9.51 |
| **RFT-List-Judge-MRR** | List-CoT | $\text{MRR}_{\text{List}}^{\text{LLM}}$ | $168 \pm 376$ | $193 \pm 622$ | $133 \pm 214$ | 1.64 | 4.45 | 4.46 |
| Gemini judge | List-CoT | $\text{MRR}_{\text{List}}^{\text{LLM-Gemini}}$ | $239 \pm 131$ | $200 \pm 142$ | $141 \pm 103$ | 1.58 | 4.25 | 4.25 |
| Simple judge prompt | List-CoT | $\text{MRR}_{\text{List}}^{\text{LLM-Simple}}$ | $192 \pm 194$ | $163 \pm 286$ | $140 \pm 43$ | 1.51 | 3.53 | 3.53 |

Table 7: Performance results of the ablation study on different initial models.

| | MCQ | QA | | List | | | |
|---|---|---|---|---|---|---|---|
| | $\mathrm{Acc}_{MCQ}$ | $\mathrm{Acc}_{QA}$ | $\mathrm{Acc}_{QA}^{LLM}$ | $\mathrm{Acc}_{List}$ | $\mathrm{Acc}_{List}^{LLM}$ | $\mathrm{MRR}_{List}$ | $\mathrm{MRR}_{List}^{LLM}$ |
| **RFT-MCQ** | | | | | | | |
| Qwen2.5 7B Instruct | 39.34 | 9.96 | 46.33 | 9.89 | 40.06 | 8.16 | 33.00 |
| Qwen2.5 3B Instruct | 31.28 | 6.82 | 36.45 | 7.74 | 32.20 | 6.96 | 28.01 |
| Qwen3 4B Instruct | **45.22** | 11.00 | 46.05 | 13.66 | 54.82 | 12.01 | 46.18 |
| OpenThinker3 7B | 33.74 | 5.19 | 40.85 | 6.88 | 33.28 | 6.27 | 29.11 |
| m1 7B 23K | 44.98 | 8.49 | 41.67 | 12.40 | 46.24 | 11.20 | 40.49 |
| AlphaMed 7B | 42.03 | 3.50 | 18.64 | 1.52 | 3.57 | 1.51 | 3.44 |
| **RFT-QA** | | | | | | | |
| Qwen2.5 7B Instruct | 36.80 | 1.04 | 25.22 | 0.62 | 3.59 | 0.48 | 2.82 |
| Qwen2.5 3B Instruct | 32.64 | 0.84 | 27.60 | 9.84 | 44.78 | 7.34 | 33.94 |
| Qwen3 4B Instruct | 45.16 | 8.56 | 44.97 | 12.34 | 55.46 | 10.53 | **46.57** |
| OpenThinker3 7B | 34.01 | 4.67 | 39.78 | 4.31 | 39.02 | 3.90 | 33.49 |
| m1 7B 23K | 43.06 | 6.88 | 45.54 | 9.53 | 45.35 | 8.52 | 40.04 |
| AlphaMed 7B | 41.74 | 3.94 | 38.35 | 10.14 | 37.89 | 9.63 | 34.95 |
| **RFT-List-Acc** | | | | | | | |
| Qwen2.5 7B Instruct | 22.40 | 4.28 | 19.01 | 22.11 | 56.61 | **16.17** | 40.26 |
| Qwen2.5 3B Instruct | 32.16 | 7.25 | 39.29 | 20.66 | 59.82 | 12.08 | 34.68 |
| Qwen3 4B Instruct | 43.72 | 11.69 | 48.45 | **27.74** | **71.60** | 15.94 | 40.60 |
| OpenThinker3 7B | 34.23 | 5.75 | 41.42 | 20.05 | 56.98 | 11.03 | 31.93 |
| m1 7B 23K | 44.04 | 8.96 | 43.92 | 26.48 | 66.34 | 15.52 | 37.11 |
| AlphaMed 7B | 38.35 | 5.82 | 25.38 | 17.86 | 57.29 | 14.32 | 44.16 |
| **RFT-List-MRR** | | | | | | | |
| Qwen2.5 7B Instruct | 18.23 | 4.43 | 21.90 | 20.96 | 61.60 | 15.83 | 44.89 |
| Qwen2.5 3B Instruct | 32.01 | 7.93 | 40.20 | 20.52 | 58.42 | 12.53 | 34.91 |
| Qwen3 4B Instruct | 44.32 | **11.93** | **48.54** | 17.60 | 58.60 | 12.32 | 40.27 |
| OpenThinker3 7B | 34.60 | 5.57 | 40.77 | 18.23 | 59.44 | 11.25 | 35.03 |
| m1 7B 23K | 43.54 | 9.85 | 45.36 | 23.57 | 67.29 | 15.61 | 42.10 |
| AlphaMed 7B | 36.44 | 7.15 | 30.38 | 18.47 | 55.22 | 15.05 | 43.91 |

**No prior prompt**   The no-prior-prompt setting is only applicable to MCQ and QA, since a ranked-list format require a one-shot example. For RFT-MCQ setup, $\mathrm{Acc}_{MCQ}$ is essentially stable (39.34% $\rightarrow$ 38.95%). In contrast, for RFT-QA, removing the prior prompt improves $\mathrm{Acc}_{QA}^{LLM}$ (25.22% $\rightarrow$ 29.17%), while also substantially boosting list performance (3.59% $\rightarrow$ 41.92% $\mathrm{Acc}_{List}^{LLM}$).

**No CoT prompt**   When removing the CoT instruction, effects are mixed. For RFT-MCQ, $\mathrm{Acc}_{MCQ}$ increased very slightly from 39.34% $\rightarrow$ 39.80%. For RFT-QA, QA performance is similar (25.22% $\rightarrow$ 24.78%), but list accuracy improves sharply (3.59% $\rightarrow$ 49.93% $\mathrm{Acc}_{List}^{LLM}$). For RFT-List-Acc, list accuracy rises from 56.61% to 67.08% and QA accuracy improves (19.01% $\rightarrow$ 44.46%), though MCQ performance decreases (22.40% $\rightarrow$ 12.97%). Similarly, for RFT-List-MRR, $\mathrm{Acc}_{QA}^{LLM}$ improves from 21.90% to 46.21% and list accuracy grows slightly (61.60% $\rightarrow$ 63.00%), while MCQ drops (18.23% $\rightarrow$ 10.41%). We conjecture that without the thinking template, the model achieves higher accuracy due to the optimization objective in both QA and list answer formats evaluation (since QA is a special case of list), but at the cost of robustness in other formats.

Another notable side effect is that under the List prompt, the average list length increases dramatically (5.94 $\rightarrow$ 615.17), as the model tends to repeat sets of results. We observe similar behaviors in other models trained with the same reward functions (Section 4.3.2). While removing prior prompt or CoT suggests improved performance, a key trade-off is the loss of the `<think></think>` structure, which is important for certain test-time scaling techniques such as budget forcing (Muennighoff et al., 2025) or thinking interventions (Wu et al., 2025).

### I.5.2   INITIAL MODELS

Tables 7 and 8 report performance and list metrics for the experiments in Section 4.3.2. We observe that model family influences response length after RFT, broadly mirroring zero-shot response-length trends. For example, SFT-trained reasoning models retain high average token counts after RFT. For other families, however, response length does not necessarily correlate with performance (as previously discussed) and varies without a consistent trend.

Table 8: Average response length (mean ± standard deviation) for MCQ, QA, and list-based answer formats across benchmarks and metrics related to the ranked list answer format from the generated evaluation responses for the initial model ablation study.

| | MCQ | QA | List | CP | LL | VLL |
|---|---|---|---|---|---|---|
| **RFT-MCQ** | | | | | | |
| Qwen2.5 7B Instruct | 204 ± 106 | 167 ± 65 | 33 ± 123 | 1.45 | 2.29 | 2.29 |
| Qwen2.5 3B Instruct | 197 ± 213 | 190 ± 159 | 179 ± 94 | 1.35 | 1.94 | 1.94 |
| Qwen3 4B Instruct | 845 ± 613 | 554 ± 509 | 453 ± 599 | 1.44 | 3.40 | 3.46 |
| OpenThinker3 7B | 1314 ± 666 | 1376 ± 1098 | 1042 ± 953 | 1.38 | 2.58 | 2.62 |
| m1 7B 23K | 1395 ± 748 | 1105 ± 703 | 1091 ± 901 | 1.34 | 2.49 | 2.50 |
| AlphaMed 7B | 342 ± 286 | 309 ± 663 | 278 ± 608 | 1.08 | 0.60 | 1.45 |
| **RFT-QA** | | | | | | |
| Qwen2.5 7B Instruct | 296 ± 302 | 291 ± 247 | 46 ± 91 | 1.67 | 0.29 | 3.01 |
| Qwen2.5 3B Instruct | 264 ± 306 | 301 ± 298 | 327 ± 640 | 1.78 | 4.16 | 4.19 |
| Qwen3 4B Instruct | 946 ± 874 | 673 ± 736 | 524 ± 536 | 1.48 | 3.64 | 3.64 |
| OpenThinker3 7B | 1270 ± 583 | 1088 ± 562 | 791 ± 467 | 1.41 | 2.96 | 2.96 |
| m1 7B 23K | 1183 ± 794 | 800 ± 453 | 749 ± 936 | 1.32 | 3.20 | 3.20 |
| AlphaMed 7B | 256 ± 142 | 246 ± 337 | 220 ± 237 | 1.19 | 1.55 | 1.57 |
| **RFT-List-Acc** | | | | | | |
| Qwen2.5 7B Instruct | 208 ± 231 | 195 ± 433 | 165 ± 177 | 2.07 | 5.94 | 5.94 |
| Qwen2.5 3B Instruct | 315 ± 188 | 217 ± 195 | 7881 ± 1522 | 4.47 | 807.73 | 808.08 |
| Qwen3 4B Instruct | 817 ± 690 | 510 ± 524 | 7929 ± 1346 | 8.65 | 800.19 | 801.19 |
| OpenThinker3 7B | 1519 ± 889 | 1221 ± 804 | 8087 ± 826 | 5.95 | 828.88 | 831.05 |
| m1 7B 23K | 1478 ± 1208 | 1148 ± 1211 | 8158 ± 425 | 7.40 | 772.45 | 773.23 |
| AlphaMed 7B | 256 ± 114 | 202 ± 323 | 3419 ± 3782 | 1.75 | 4.82 | 4.83 |
| **RFT-List-MRR** | | | | | | |
| Qwen2.5 7B Instruct | 174 ± 110 | 157 ± 207 | 319 ± 1154 | 2.11 | 16.97 | 16.97 |
| Qwen2.5 3B Instruct | 334 ± 236 | 240 ± 198 | 7182 ± 2601 | 4.04 | 733.87 | 734.94 |
| Qwen3 4B Instruct | 752 ± 787 | 537 ± 787 | 52 ± 94 | 2.17 | 5.79 | 5.79 |
| OpenThinker3 7B | 1476 ± 883 | 1195 ± 817 | 8048 ± 984 | 5.87 | 811.56 | 814.41 |
| m1 7B 23K | 1543 ± 1039 | 1249 ± 1204 | 7679 ± 1895 | 5.55 | 717.71 | 719.24 |
| AlphaMed 7B | 277 ± 118 | 198 ± 285 | 218 ± 95 | 1.66 | 5.48 | 5.48 |

## J TRAINING DYNAMICS

We examine training dynamics using two metrics: (1) **reward progression** and (2) **response length trends**. These metrics allow us to holistically observe how changes in factors such as reward function, model family, or answer format affect the training process. The list results are presented and discussed alongside quantitative performance metrics for the main experiments, where applicable.

### J.1 MAIN EXPERIMENT

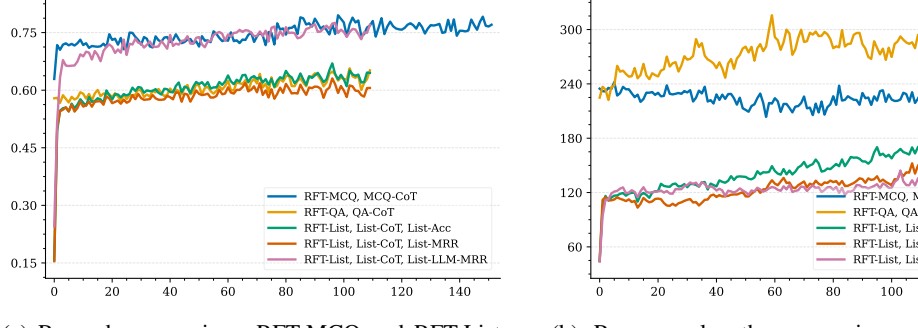

(a) Reward progression. RFT-MCQ and RFT-List-LLM-MRR methods achieve high rewards.

(b) Response length progression. Different approaches yield varying response length behaviors.

Figure 15: Training dynamics across different answer formats and reward functions.

This sections provides a discussion on training dynamics of the models from Section 4.2. Training dynamics is provided in Figure 15. All RFT models share similar training dynamics with some differences. Among the RFT-List models, different reward types also lead to slightly different training dynamics, with the exception of RFT-List-Judge-MRR. For reward progression, most models exhibit a common trend: an initial low reward followed by a sharp increase, reflecting behavior aimed

at optimizing the format reward. The reward then continues to increase gradually over the course of training. We also observe that RFT-MCQ and RFT-List-Judge-MRR achieve higher rewards than the other models, suggesting that these models are able to score correct answers more consistently under their respective reward types. QA is more challenging due to its reliance on exact match rewards, similar to the list format. However, the final reward obtained during training does not reliably predict final performance (Pearson $r = -0.267$, $p = 0.0671$, across all our RFT models in all experiments).

For response length progression, most models follow a pattern similar to reward progression: starting with short responses that gradually increase in length. An exception is RFT-MCQ, which consistently produces longer responses than the other models, and RFT-QA, which generates responses that are longer than those from most other models.

## J.2 FACTORS AFFECTING RFT

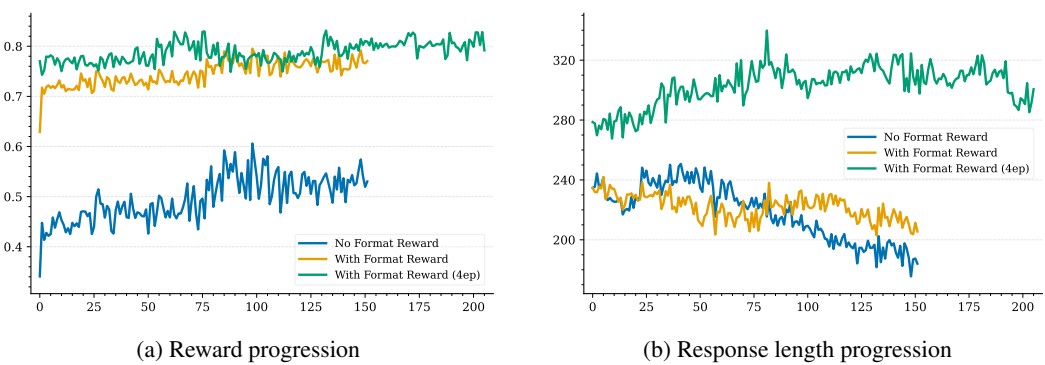

(a) Reward progression                    (b) Response length progression

Figure 16: Training dynamics comparison between using and not using format reward, as well as the effect of extending training from 2 to 4 epochs.

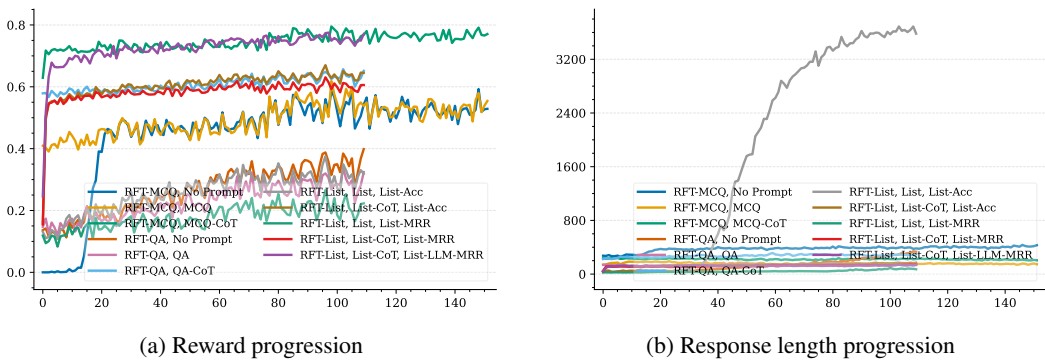

(a) Reward progression                    (b) Response length progression

Figure 17: Training dynamics comparison between different types of prior prompts and the case without any prior prompt.

Training dynamics are illustrated in Figures 16, 17 and 18b, which correspond respectively to experiments on removing the format reward, extending training, altering or removing the prior prompt, and changing the LLM judge.

Removing the format reward does not substantially affect final model performance or list behaviors. The primary differences lie in training dynamics: models without a format reward exhibit a lower reward range during training, as they must focus exclusively on accuracy without a steady signal of format reward. In addition, response length shows a slightly more pronounced decreasing trend compared to models trained with the format reward.

Extending training from two to four epochs does not substantially improve performance, except for a tendency toward longer responses. Interestingly, the run with longer training begins with a higher

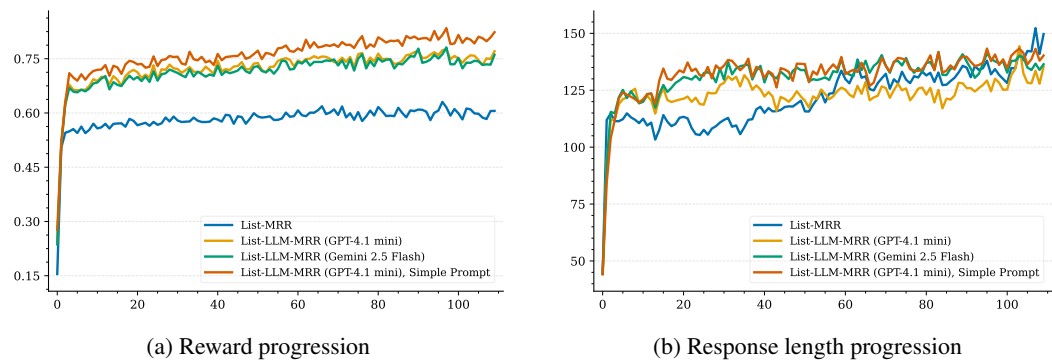

(a) Reward progression          (b) Response length progression

Figure 18: Training dynamics comparison across different Judge-MRR settings: GPT-4.1-mini and Gemini with the standard judge prompt, and GPT-4.1-mini with a simplified judge prompt.

initial reward and response length. Although, the overall training dynamics remain similar to the shorter run.

Models trained without prior prompts show lower initial rewards but experience a sharper increase later, while response length remains relatively stable throughout training.

## J.3 INITIAL MODELS

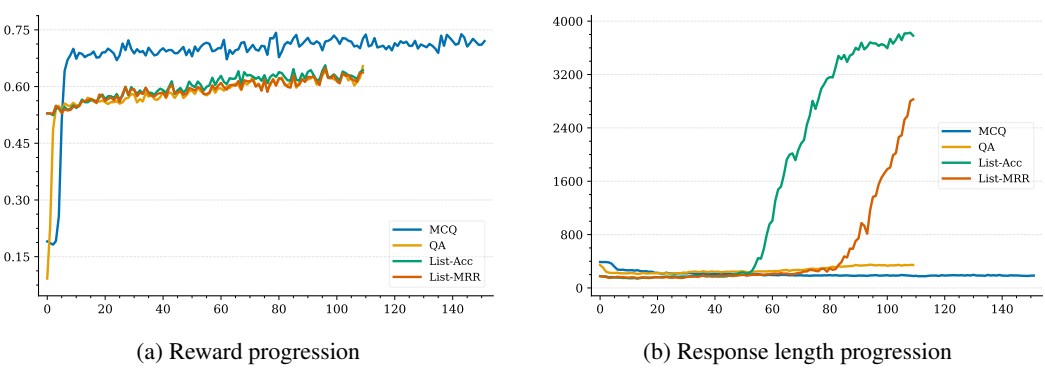

(a) Reward progression          (b) Response length progression

Figure 19: Training dynamics for Qwen2.5 3B Instruct.

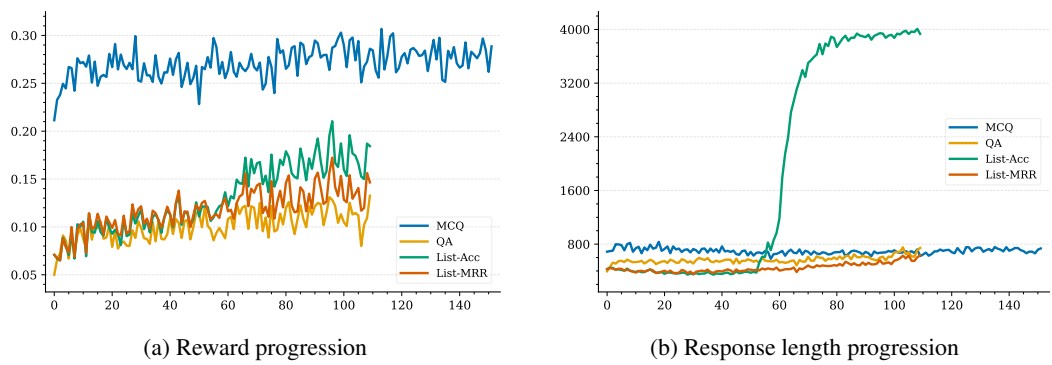

(a) Reward progression          (b) Response length progression

Figure 20: Training dynamics for Qwen3 4B.

Training dynamic of different initial models are illustrated in Figures 19 to 23.

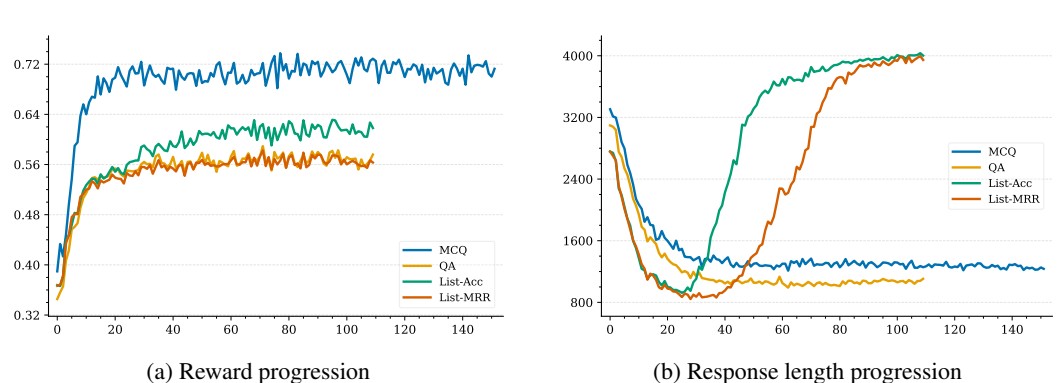

(a) Reward progression

(b) Response length progression

Figure 21: Training dynamics for OpenThinker 7B.

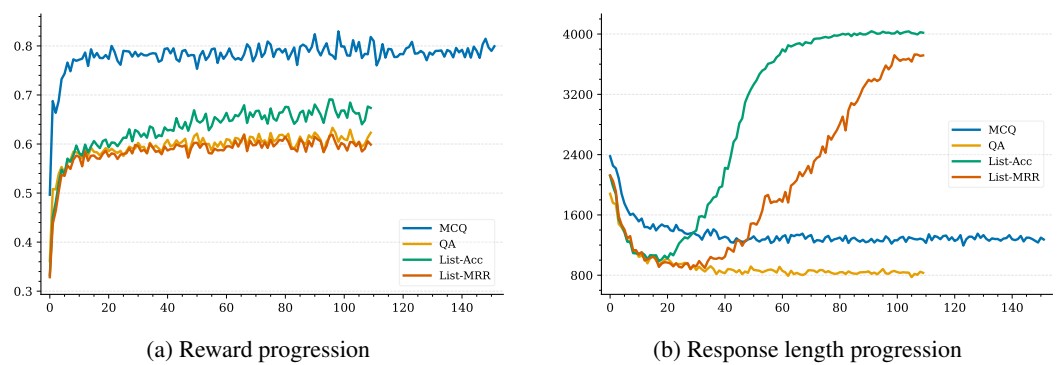

(a) Reward progression

(b) Response length progression

Figure 22: Training dynamics for m1 7B 23k.

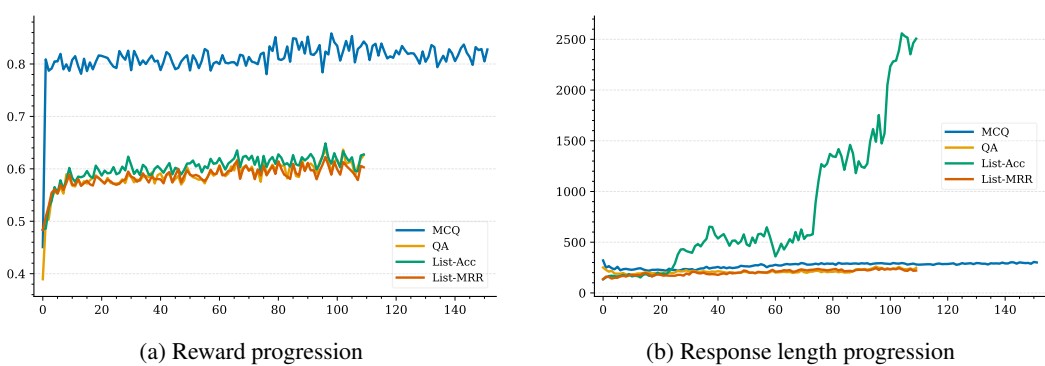

(a) Reward progression

(b) Response length progression

Figure 23: Training dynamics for AlphaMed 7B.

Table 9: Performance results of the ablation study on mixed datasets.

| | MCQ | QA | | List | | | |
|---|---|---|---|---|---|---|---|
| | $Acc_{MCQ}$ | $Acc_{QA}$ | $Acc_{QA}^{LLM}$ | $Acc_{List}$ | $Acc_{List}^{LLM}$ | $MRR_{List}$ | $MRR_{List}^{LLM}$ |
| **RFT-MCQ** | 39.34 | 9.96 | 46.33 | 9.89 | 40.06 | 8.16 | 33.00 |
| **RFT-QA** | 36.80 | 1.04 | 25.22 | 0.62 | 3.59 | 0.48 | 2.82 |
| **RFT-List-Acc** | 22.40 | 4.28 | 19.01 | **22.11** | 56.61 | **16.17** | 40.26 |
| **RFT-List-MRR** | 18.23 | 4.43 | 21.90 | 20.96 | **61.60** | 15.83 | **44.89** |
| RFT-MCQ+QA | 12.81 | 3.40 | 31.96 | 6.39 | 41.18 | 5.56 | 34.24 |
| RFT-MCQ+List-Acc | 39.70 | 11.24 | 46.89 | 21.47 | 61.52 | 15.10 | 41.93 |
| RFT-MCQ+List-MRR | **40.01** | **11.82** | **47.91** | 18.99 | 57.34 | 14.89 | 43.80 |

Table 10: Average response length (mean $\pm$ standard deviation) for MCQ, QA, and list-based answer formats across benchmarks and metrics related to the ranked list answer format from the generated evaluation responses for the mixed datasets ablation study.

| | MCQ | QA | List | CP | LL | VLL |
|---|---|---|---|---|---|---|
| **RFT-MCQ** | $204 \pm 106$ | $167 \pm 65$ | $33 \pm 123$ | 1.45 | 2.29 | 2.29 |
| **RFT-QA** | $296 \pm 302$ | $291 \pm 247$ | $46 \pm 91$ | 1.67 | 0.29 | 3.01 |
| **RFT-List-Acc** | $208 \pm 231$ | $195 \pm 433$ | $165 \pm 177$ | 2.07 | 5.94 | 5.94 |
| **RFT-List-MRR** | $174 \pm 110$ | $157 \pm 207$ | $319 \pm 1154$ | 2.11 | 16.97 | 16.97 |
| RFT-MCQ+QA | $233 \pm 170$ | $209 \pm 130$ | $116 \pm 190$ | 1.48 | 2.78 | 2.78 |
| RFT-MCQ+List-Acc | $246 \pm 209$ | $201 \pm 173$ | $193 \pm 541$ | 2.45 | 10.34 | 10.35 |
| RFT-MCQ+List-MRR | $210 \pm 202$ | $171 \pm 276$ | $134 \pm 73$ | 1.78 | 4.92 | 4.92 |

# K    RFT WITH MIXED ANSWER FORMAT DATASETS

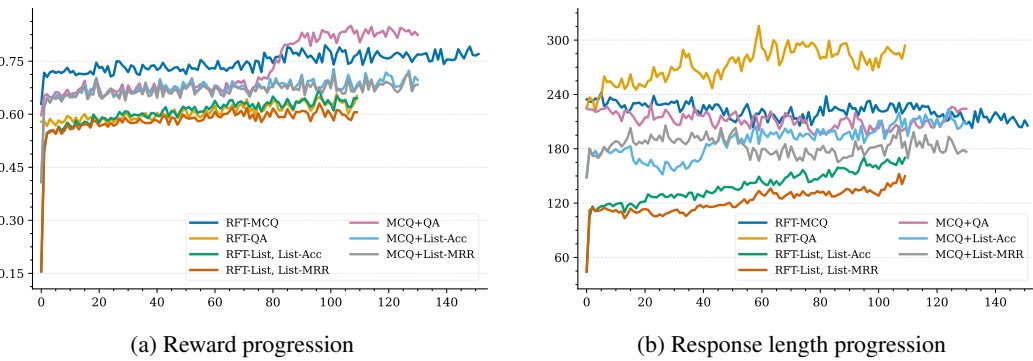

(a) Reward progression        (b) Response length progression

Figure 24: Training dynamics comparison across different dataset types: MCQ-only, QA-only, List-only, MCQ+QA, and MCQ+List.

To evaluate whether combining different answer formats can improve RFT, we construct mixed datasets by merging MCQ and QA data with appropriate prior prompts. Since mixing effectively doubles the dataset size, we train for one epoch instead of two to maintain a comparable number of optimization steps with the main experiments. Records are shuffled randomly, and rewards are computed according to the record type. Performance and list results are shown in Tables 9 and 10, and training dynamics are presented in Figure 24.

When training with a mixed dataset of MCQ and List using the $MRR_{List}$ reward function, we observe the best overall performance on MCQ and QA compared to training on either dataset alone. However, this setting does not reach the strongest performance on the a ranked-list answer formats, where training exclusively on list data remains superior. A similar pattern holds for MCQ and List under the $Acc_{List}$ reward.

In contrast, mixing MCQ and QA yields weaker results. While QA performance improves relative to training with QA alone, MCQ and list-format performance degrade noticeably. This outcome suggests that the mixture introduces instability, likely because QA is a sparse-reward task, making the overall reward signal less reliable when combined with denser MCQ data.

Table 11: Performance results of the ablation study on length penalty.

| | MCQ | QA | | List | | | |
|---|---|---|---|---|---|---|---|
| | $\text{Acc}_{\text{MCQ}}$ | $\text{Acc}_{\text{QA}}$ | $\text{Acc}_{\text{QA}}^{\text{LLM}}$ | $\text{Acc}_{\text{List}}$ | $\text{Acc}_{\text{List}}^{\text{LLM}}$ | $\text{MRR}_{\text{List}}$ | $\text{MRR}_{\text{List}}^{\text{LLM}}$ |
| Qwen2.5 3B Instruct | 32.16 | 7.25 | 39.29 | 20.66 | 59.82 | 12.08 | 34.68 |
| + $LP$ with $\lambda = 0.3$ | 31.16 | 6.32 | 30.05 | 9.91 | 26.65 | 9.60 | 25.06 |
| OpenThinker3 7B | 34.23 | 5.75 | 41.42 | 20.05 | 56.98 | 11.03 | 31.93 |
| + $LP$ with $\lambda = 0.3$ | 34.36 | 5.78 | 41.79 | 9.22 | 26.86 | 8.94 | 25.01 |
| m1 7B 23K | **44.04** | 8.96 | **43.92** | **26.48** | **66.34** | **15.52** | 37.11 |
| + $LP$ with $\lambda = 0.3$ | 43.70 | 8.43 | 43.51 | 13.16 | 33.23 | 12.81 | 32.16 |
| AlphaMed 7B | 38.35 | 5.82 | 25.38 | 17.86 | 57.29 | 14.32 | **44.16** |
| + $LP$ with $\lambda = 0.3$ | 21.82 | **9.65** | 38.60 | 13.48 | 33.55 | 13.25 | 31.94 |

Table 12: Average response length (mean $\pm$ standard deviation) for MCQ, QA, and list-based answer formats across benchmarks and metrics related to the ranked list answer format from the generated evaluation responses for the length penalty ablation study.

| | MCQ | QA | List | CP | LL | VLL |
|---|---|---|---|---|---|---|
| Qwen2.5 3B Instruct | $315 \pm 188$ | $217 \pm 195$ | $7881 \pm 1522$ | 4.47 | 807.73 | 808.08 |
| + $LP$ with $\lambda = 0.3$ | $230 \pm 188$ | $178 \pm 106$ | $131 \pm 107$ | 1.13 | 1.34 | 1.35 |
| OpenThinker3 7B | $1519 \pm 889$ | $1221 \pm 804$ | $8087 \pm 826$ | 5.95 | 828.88 | 831.05 |
| + $LP$ with $\lambda = 0.3$ | $1288 \pm 671$ | $1077 \pm 585$ | $720 \pm 479$ | 1.15 | 1.41 | 1.41 |
| m1 7B 23K | $1478 \pm 1208$ | $1148 \pm 1211$ | $8158 \pm 425$ | 7.40 | 772.45 | 773.23 |
| + $LP$ with $\lambda = 0.3$ | $1370 \pm 914$ | $890 \pm 538$ | $742 \pm 450$ | 1.09 | 1.22 | 1.22 |
| AlphaMed 7B | $256 \pm 114$ | $202 \pm 323$ | $3419 \pm 3782$ | 1.75 | 4.82 | 4.83 |
| + $LP$ with $\lambda = 0.3$ | $213 \pm 86$ | $174 \pm 217$ | $151 \pm 126$ | 1.11 | 1.31 | 1.31 |

Across all experiments, we find that mixing answer formats in the same dataset is not particularly effective. None of the models trained on mixed datasets produce excessively long lists (e.g., >100 items), but performance trade-offs prevent mixed training from outperforming single-answer-format training in most cases.

## L  LIST REWARD HACKING AND MITIGATION WITH LENGTH PENALTY

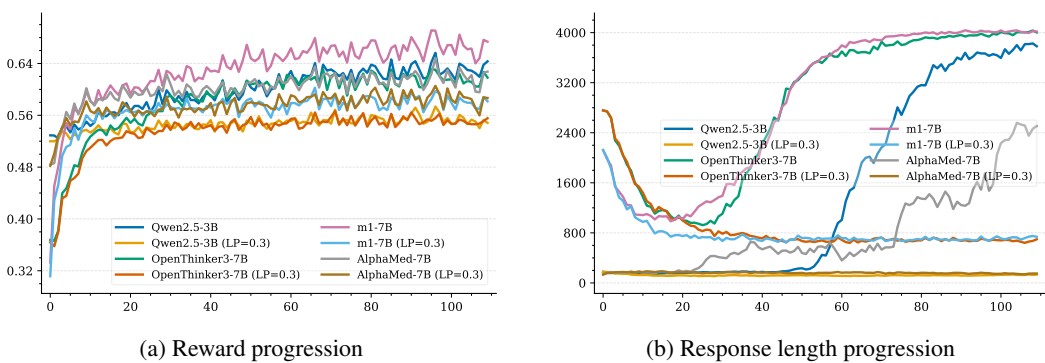

(a) Reward progression

(b) Response length progression

Figure 25: Training dynamics comparison for models before and after applied the length penalty.

As shown in Sections 4.3.1 and 4.3.2, some models exhibit a tendency to generate excessively long lists (sometimes exceeding 100 items) when trained with list-based reward functions. While training on mixed-format datasets (Appendix K) can partially mitigate this issue, we propose an alternative approach: modifying the list reward functions themselves by introducing **a length penalty term**.

Both $\text{Acc}_{\text{List}}$ and $\text{MRR}_{\text{List}}$ reward functions incentivze correctness but do not discourage unnecessarily long outputs. To address this, we introduce a length penalty term ($LP$) that scales the reward according to the number of items in the list. Let $L$ denote the length of the generated list and $\lambda$ the

Table 13: Performance results of the ablation study on length penalty.

| | MCQ | QA | | List | | | |
|---|---|---|---|---|---|---|---|
| | $Acc_{MCQ}$ | $Acc_{QA}$ | $Acc_{QA}^{LLM}$ | $Acc_{List}$ | $Acc_{List}^{LLM}$ | $MRR_{List}$ | $MRR_{List}^{LLM}$ |
| LP=0 | 12.93 | 8.23 | 34.55 | **19.89** | **54.95** | **15.78** | **41.59** |
| LP=0.1 | 37.73 | **11.85** | 47.23 | 13.90 | 36.02 | 13.46 | 33.40 |
| LP=0.3 | 33.52 | 11.54 | **48.30** | 12.64 | 28.64 | 12.34 | 27.71 |
| LP=0.5 | **37.78** | 11.24 | 46.25 | 11.86 | 26.02 | 11.86 | 26.01 |
| LP=0.7 | 34.71 | 11.56 | 47.35 | 11.68 | 27.39 | 11.68 | 27.34 |
| LP=0.9 | 35.47 | 10.34 | 46.08 | 11.83 | 27.07 | 11.83 | 27.05 |

Table 14: Average response length (mean $\pm$ standard deviation) for MCQ, QA, and list-based answer formats across benchmarks and metrics related to the ranked list answer format from the generated evaluation responses for the length penalty ablation study.

| | MCQ | QA | List | CP | LL | VLL |
|---|---|---|---|---|---|---|
| LP=0 | $170 \pm 141$ | $166 \pm 296$ | $132 \pm 141$ | 1.78 | 4.71 | 4.71 |
| LP=0.1 | $176 \pm 207$ | $170 \pm 423$ | $127 \pm 174$ | 1.18 | 1.52 | 1.52 |
| LP=0.3 | $160 \pm 272$ | $164 \pm 372$ | $11 \pm 6$ | 1.08 | 1.16 | 1.16 |
| LP=0.5 | $157 \pm 76$ | $139 \pm 100$ | $10 \pm 3$ | 1.00 | 1.01 | 1.01 |
| LP=0.7 | $161 \pm 186$ | $166 \pm 362$ | $11 \pm 4$ | 1.01 | 1.05 | 1.05 |
| LP=0.9 | $60 \pm 117$ | $64 \pm 80$ | $11 \pm 4$ | 1.00 | 1.03 | 1.03 |

penalty coefficient. The penalty term is defined as:

$$LP = \max\left(0,\, 1 - \lambda \cdot (L - 1)\right).$$

Since $LP$ is orthogonal to existing reward functions, it can be applied to both $Acc_{List}$ and $MRR_{List}$ by first computing the correctness reward and then scaling it by $LP$. In both cases, the penalty encourages concise and precise outputs rather than exhaustive or repetitive enumerations. This introduces an additional optimization constraint: the model must both produce the correct answer and avoid generating unnecessarily long lists. While we adopt the simplest linear form of length penalty here, future work may explore more flexible variants that allow stronger or weaker tolerance for longer outputs.

**Effectiveness of $LP$ in mitigating large lists**   We evaluate the effectiveness of the length penalty by applying it to four models that previously exhibited excessively long lists when trained with the $Acc_{List}$ reward function: Qwen2.5 3B Instruct, OpenThinker3 7B, m1 7B 23k, and AlphaMed 7B. We set $\lambda = 0.3$ as an arbitrary but fixed choice. Results in Table 12 show that the length penalty successfully reduces the large-list behavior across all models. Training dynamics in Figure 25 further demonstrate that response length remains more stable over time, with similar reward progression trends compared to the unpenalized setup, albeit at slightly lower reward values. We also observe that the gap between AccList and MRRList narrows in the evaluation results (Table 11), as the length penalty encourages correct items to appear in higher positions, thereby reducing the average correct position (Table 12).

**Trade-offs of $LP$ between list length and performance**   Although the length penalty effectively reduces uncontrolled list growth, it introduces trade-offs. Across models, we observe moderate reductions in MCQ and QA performance, and more substantial drops in a ranked-list answer format performance, where the penalty directly applies. This highlights the inherent trade-off between controlling undesirable behaviors and maximizing performance. As noted, our penalty term represents a simple first step and could be refined to better balance this trade-off. Larger models, which we do not investigate due to computational constraints, may also be less sensitive to such penalties. Additional experiments on varying $\lambda$ are presented in Appendix M.

## M   LENGTH PENALTY HYPERPARAMETERS

In this section, we examine how different values of $\lambda$ affect both performance and list length. We select $\lambda \in \{0.1, 0.3, 0.5, 0.7, 0.9\}$ to cover a range from mild to strong penalization. To demonstrate the generality of the $LP$ formulation, we study its application to the $MRR_{List}$ reward function using

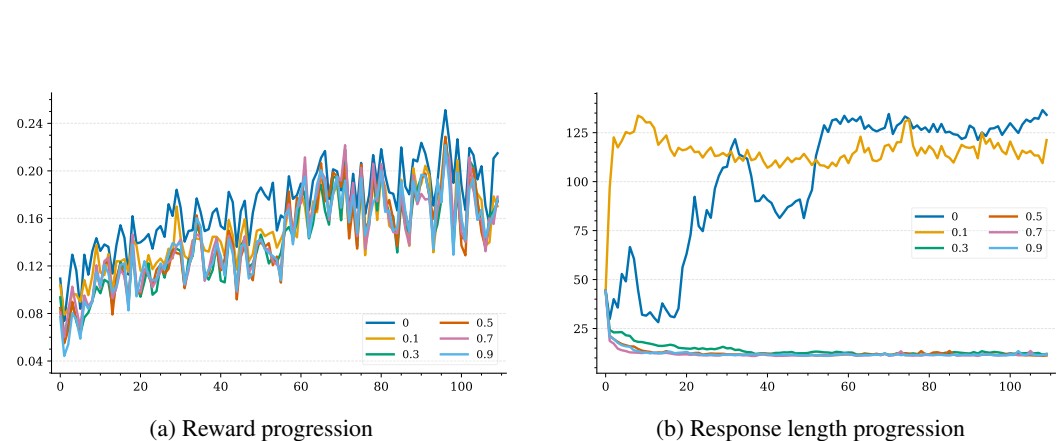

(a) Reward progression      (b) Response length progression

Figure 26: Training dynamics comparison for length penalty ablation.

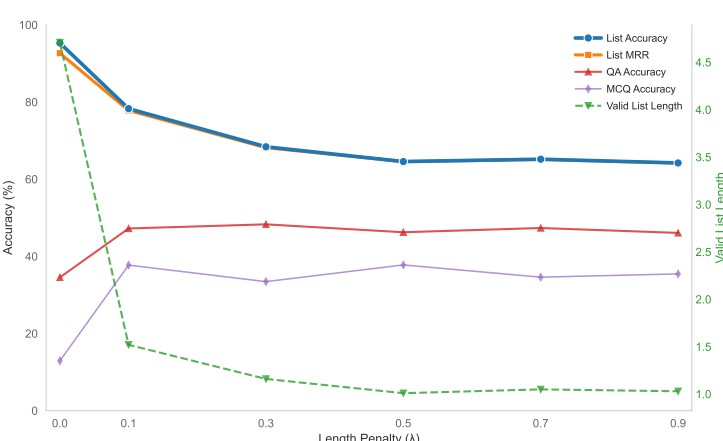

Figure 27: Impact of length penalty on model performance. Performance trends across different task types, where List tasks demonstrate highest baseline performance but steep degradation, QA tasks exhibit optimal performance at $\lambda = 0.1$–$0.3$, and MCQ tasks show consistent improvement with positive length penalty.

Table 15: Results table for prompting experiments evaluated on MCQ benchmarks using both direct **MCQ** and **MCQ-CoT** prompting strategies from Section 3. MXQA refers to MedXpertQA (text), and MLUP-H refers to MMLU Pro (Health).

| | MedQA | MedMCQA | MXQA | MLUP-H | Average |
|---|---|---|---|---|---|
| *Proprietary Reasoning Models* | | | | | |
| Gemini 2.5 Flash Lite | 82.01 | 18.80 | 20.82 | 72.25 | 48.47 |
| +CoT | 50.98 | 9.30 | 11.27 | 30.81 | 25.59 |
| Gemini 2.5 Flash | 92.22 | 18.05 | 37.39 | 73.11 | 55.19 |
| +CoT | 61.67 | 7.51 | 29.92 | 42.67 | 35.44 |
| Gemini 2.5 Pro | 94.34 | 18.78 | 44.94 | 76.65 | **58.68** |
| +CoT | **94.82** | 16.72 | **45.76** | 74.94 | 58.06 |
| *Proprietary Large Language Models* | | | | | |
| GPT-4.1 Mini | 90.49 | 18.91 | 31.59 | **77.87** | 54.72 |
| +CoT | 72.74 | 18.10 | 29.59 | 70.42 | 47.71 |
| *Open-weight Large Language Models* | | | | | |
| Qwen2.5 3B Instruct | 48.08 | **24.11** | 8.98 | 37.29 | 29.62 |
| +CoT | 52.24 | 23.63 | 11.84 | 43.03 | 32.68 |
| Qwen2.5 7B Instruct | 17.99 | 10.20 | 6.08 | 19.44 | 13.43 |
| +CoT | 49.10 | 17.58 | 12.33 | 45.97 | 31.24 |
| Qwen2.5 14B Instruct | 59.23 | 19.51 | 10.12 | 54.65 | 35.88 |
| +CoT | 54.91 | 19.56 | 10.20 | 60.64 | 36.33 |
| Qwen3 4B Instruct 2507 | 73.21 | 19.53 | 16.16 | 66.38 | 43.82 |
| +CoT | 68.34 | 17.48 | 16.29 | 62.35 | 41.12 |
| Gemma 3 4B IT | 49.80 | 21.04 | 12.00 | 38.88 | 30.43 |
| +CoT | 46.27 | 16.57 | 11.59 | 38.14 | 28.14 |
| *Open-weight Medical LLMs* | | | | | |
| MedGemma 4B IT | 63.79 | 21.48 | 14.69 | 48.41 | 37.09 |
| +CoT | 54.83 | 16.55 | 13.76 | 36.92 | 30.52 |
| MedGemma 27B IT | 81.46 | 20.21 | 22.45 | 71.76 | 48.97 |
| +CoT | 65.44 | 10.81 | 16.90 | 37.78 | 32.73 |
| *Open-weight Reasoning Models* | | | | | |
| OpenThinker3 7B | 48.15 | 18.07 | 5.55 | 38.51 | 27.57 |
| +CoT | 3.46 | 3.15 | 2.41 | 5.99 | 3.75 |
| *Open-weight Medical Reasoning Models* | | | | | |
| HuatuoGPT o1 7B | 17.67 | 9.19 | 9.06 | 35.09 | 17.75 |
| +CoT | 6.28 | 6.47 | 5.47 | 22.62 | 10.21 |
| m1 7B 23K | 64.10 | 20.65 | 15.92 | 56.36 | 39.26 |
| +CoT | 53.73 | 15.43 | 12.12 | 44.25 | 31.38 |
| AlphaMed 7B Instruct RL | 69.91 | 16.49 | 16.24 | 59.41 | 40.51 |
| +CoT | 60.41 | 14.54 | 16.73 | 58.19 | 37.47 |

Qwen2.5 7B Instruct. Performance and list results are presented in Tables 13 and 14, and training dynamics are shown in Figure 26.

Introducing the length penalty consistently reduces the length of generated lists across $\lambda$ values, confirming its effectiveness in controlling overly long outputs. However, this comes at the cost of reduced performance, as models become more reluctant to produce longer lists. The trade-offs discussed in Appendix L are observed across all choices of $\lambda$. Interestingly, applying the length penalty also improves performance on single-answer tasks such as MCQ and QA. This suggests that constraining the model to produce shorter, more focused outputs may indirectly benefit tasks where concise responses are essential.

# N   FULL RESULTS

This section provides a detailed breakdown of results for each benchmark based on experiments reported across papers.

## N.1   PROMPTING

Tables 15 to 17 present the non-aggregated versions of the aggregated results shown in the main body of this study (Table 1 in Section 3). While, Tables 18 to 20 show average response length

Table 16: Results table for prompting experiments evaluated on open-ended benchmarks using both direct **QA** and **QA-CoT** prompting strategies from Section 3. MXQA refers to MedXpertQA (text), and MLUP-H refers to MMLU Pro (Health).

| | MedQA | | MedMCQA | | MXQA | | MLUP-H | | Average | |
|---|---|---|---|---|---|---|---|---|---|---|
| | $Acc_{QA}$ | $Acc_{QA}^{LLM}$ | $Acc_{QA}$ | $Acc_{QA}^{LLM}$ | $Acc_{QA}$ | $Acc_{QA}^{LLM}$ | $Acc_{QA}$ | $Acc_{QA}^{LLM}$ | $Acc_{QA}$ | $Acc_{QA}^{LLM}$ |
| *Proprietary Reasoning Models* | | | | | | | | | | |
| Gemini 2.5 Flash Lite | 19.63 | 53.69 | **10.00** | 45.69 | 6.33 | 41.18 | 13.45 | 54.21 | 12.35 | 48.69 |
| +CoT | 18.17 | 48.34 | 8.81 | 37.75 | 6.14 | 34.66 | 12.23 | 48.91 | 11.34 | 42.42 |
| Gemini 2.5 Flash | 17.03 | 50.45 | 9.91 | 43.76 | **8.49** | 42.38 | 12.36 | 47.83 | 11.95 | 46.10 |
| +CoT | 17.84 | 49.88 | 9.86 | 41.56 | 6.90 | 40.27 | 11.14 | 45.92 | 11.44 | 44.41 |
| Gemini 2.5 Pro | 17.52 | **54.74** | 8.58 | 44.82 | 7.00 | **44.53** | 10.46 | 52.72 | 10.89 | **49.20** |
| +CoT | 16.46 | 52.55 | 7.75 | 45.83 | 6.57 | 42.86 | 10.46 | 51.09 | 10.31 | 48.08 |
| *Proprietary Large Language Models* | | | | | | | | | | |
| GPT-4.1 Mini | 13.87 | 52.31 | 8.90 | 45.64 | 5.75 | 39.98 | 8.70 | 50.14 | 9.30 | 47.02 |
| +CoT | 16.22 | 52.72 | 9.40 | 46.24 | 6.66 | 42.67 | 10.46 | **54.35** | 10.68 | 49.00 |
| *Open-weight Large Language Models* | | | | | | | | | | |
| Qwen2.5 3B Instruct | 7.38 | 36.25 | 4.72 | 36.06 | 2.49 | 33.22 | 8.15 | 37.09 | 5.68 | 35.66 |
| +CoT | 8.27 | 41.28 | 4.95 | 40.46 | 2.73 | 34.85 | 7.61 | 44.16 | 5.89 | 40.19 |
| Qwen2.5 7B Instruct | 11.76 | 45.90 | 6.42 | 43.35 | 3.50 | 35.14 | 9.92 | 48.37 | 7.90 | 43.19 |
| +CoT | 12.08 | 46.80 | 6.47 | 41.19 | 3.88 | 35.67 | 11.41 | 49.73 | 8.46 | 43.35 |
| Qwen2.5 14B Instruct | 15.25 | 49.64 | 9.13 | 44.68 | 4.89 | 38.64 | 12.36 | 48.37 | 10.41 | 45.33 |
| +CoT | 15.33 | 53.20 | 9.04 | **47.11** | 4.65 | 39.41 | 11.41 | 51.49 | 10.11 | 47.80 |
| Qwen3 4B Instruct 2507 | 15.49 | 51.50 | 8.30 | 45.00 | 5.18 | 40.36 | 12.91 | 52.04 | 10.47 | 47.22 |
| +CoT | 16.06 | 50.45 | 7.94 | 40.50 | 4.55 | 39.65 | 11.55 | 48.10 | 10.02 | 44.68 |
| Gemma 3 4B IT | 8.19 | 36.58 | 5.60 | 40.28 | 3.07 | 30.92 | 8.02 | 38.72 | 6.22 | 36.62 |
| +CoT | 9.73 | 40.31 | 5.28 | 40.92 | 2.88 | 35.67 | 9.10 | 43.75 | 6.75 | 40.16 |
| *Open-weight Medical LLMs* | | | | | | | | | | |
| MedGemma 4B IT | 12.81 | 46.55 | 6.33 | 42.89 | 3.02 | 35.76 | 9.24 | 47.55 | 7.85 | 43.19 |
| +CoT | 12.33 | 47.61 | 6.97 | 41.93 | 3.31 | 34.80 | 10.46 | 47.01 | 8.27 | 42.84 |
| MedGemma 27B IT | **20.36** | 54.66 | 9.91 | 44.04 | 7.00 | 40.22 | **13.86** | 51.63 | **12.78** | 47.64 |
| +CoT | 15.82 | 45.01 | 9.04 | 40.28 | 5.32 | 32.36 | 12.23 | 46.20 | 10.60 | 40.96 |
| *Open-weight Reasoning Models* | | | | | | | | | | |
| OpenThinker3 7B | 6.33 | 32.77 | 4.72 | 31.28 | 1.87 | 24.35 | 8.15 | 35.73 | 5.27 | 31.03 |
| +CoT | 5.92 | 33.41 | 4.36 | 31.56 | 1.97 | 23.30 | 7.20 | 34.51 | 4.86 | 30.70 |
| *Open-weight Medical Reasoning Models* | | | | | | | | | | |
| HuatuoGPT o1 7B | 1.30 | 3.24 | 0.87 | 4.04 | 0.58 | 2.88 | 1.63 | 5.16 | 1.10 | 3.83 |
| +CoT | 0.65 | 2.43 | 0.37 | 1.79 | 0.19 | 1.87 | 1.09 | 2.72 | 0.57 | 2.20 |
| m1 7B 23K | 12.33 | 43.15 | 6.38 | 38.99 | 3.74 | 32.17 | 8.15 | 41.58 | 7.65 | 38.97 |
| +CoT | 12.00 | 43.80 | 6.10 | 35.78 | 2.83 | 28.04 | 9.92 | 41.44 | 7.71 | 37.26 |
| AlphaMed 7B Instruct RL | 0.41 | 5.52 | 1.10 | 13.53 | 0.00 | 7.38 | 1.77 | 11.41 | 0.82 | 9.46 |
| +CoT | 8.76 | 25.22 | 5.28 | 28.30 | 2.35 | 17.40 | 6.39 | 26.90 | 5.70 | 24.46 |

Table 17: Results table for prompting experiments evaluated on open-ended benchmarks using both direct **List** and **List-CoT** prompting strategies from Section 3.

| Model | MedQA | | | | MedMCQA | | | | MedXpertQA (Text) | | | | MMLU Pro (Health) | | | | Average | | | |
|---|---|---|---|---|---|---|---|---|---|---|---|---|---|---|---|---|---|---|---|---|
| | $Acc_{List}$ | $Acc_{List}^{LLM}$ | $MRR_{List}$ | $MRR_{List}^{LLM}$ | $Acc_{List}$ | $Acc_{List}^{LLM}$ | $MRR_{List}$ | $MRR_{List}^{LLM}$ | $Acc_{List}$ | $Acc_{List}^{LLM}$ | $MRR_{List}$ | $MRR_{List}^{LLM}$ | $Acc_{List}$ | $Acc_{List}^{LLM}$ | $MRR_{List}$ | $MRR_{List}^{LLM}$ | $Acc_{List}$ | $Acc_{List}^{LLM}$ | $MRR_{List}$ | $MRR_{List}^{LLM}$ |
| *Proprietary Reasoning Models* | | | | | | | | | | | | | | | | | | | | |
| Gemini 2.5 Flash Lite | 21.74 | 63.83 | 19.96 | 57.07 | 13.17 | 41.93 | 11.54 | 34.71 | 9.92 | 46.60 | 7.92 | 35.93 | 14.54 | 64.27 | 13.94 | 58.74 | 14.84 | 54.16 | 13.34 | 46.61 |
| +CoT | 10.14 | 25.22 | 9.70 | 24.11 | 9.17 | 24.54 | 8.72 | 22.92 | 3.84 | 13.52 | 3.46 | 11.97 | 9.24 | 34.92 | 9.01 | 33.42 | 8.10 | 24.55 | 7.72 | 23.10 |
| Gemini 2.5 Flash | **25.79** | 74.37 | 23.18 | 65.24 | 13.72 | 44.82 | 12.06 | 37.38 | 13.47 | 60.93 | 11.46 | 49.40 | **16.98** | 72.01 | 15.35 | 62.80 | 17.49 | 63.03 | 15.51 | 53.70 |
| +CoT | 9.65 | 27.90 | 9.06 | 25.08 | 8.21 | 26.19 | 7.69 | 23.62 | 4.12 | 18.70 | 3.47 | 16.28 | 8.02 | 36.14 | 7.44 | 32.55 | 7.50 | 27.23 | 6.92 | 24.38 |
| Gemini 2.5 Pro | 25.71 | **78.51** | 23.86 | 70.88 | 15.41 | 48.39 | 13.11 | 38.58 | **15.77** | 67.26 | 12.98 | 55.93 | 16.85 | 76.49 | 15.81 | 67.92 | 18.44 | 67.66 | 16.44 | 58.33 |
| +CoT | 23.28 | 76.24 | 21.24 | 68.21 | 14.45 | 46.61 | 12.17 | 37.21 | 13.71 | 64.96 | 11.21 | 53.62 | 14.54 | 73.10 | 13.32 | 63.71 | 16.50 | 65.23 | 14.48 | 55.69 |
| *Proprietary Large Language Models* | | | | | | | | | | | | | | | | | | | | |
| GPT-4.1 Mini | 18.98 | 73.15 | 17.82 | 67.10 | 11.88 | 47.89 | 10.29 | 39.48 | 9.40 | 56.09 | 7.39 | 45.96 | 10.87 | 70.11 | 10.20 | 63.60 | 12.78 | 61.81 | 11.42 | 54.04 |
| +CoT | 17.84 | 75.59 | 16.66 | 69.57 | 12.34 | **50.09** | 10.36 | **40.57** | 10.35 | 59.59 | 8.39 | 48.96 | 10.87 | 74.05 | 10.12 | 67.41 | 12.85 | 64.83 | 11.38 | 56.63 |
| *Open-weight Large Language Models* | | | | | | | | | | | | | | | | | | | | |
| Qwen2.5 3B Instruct | 13.38 | 46.55 | 10.45 | 35.76 | 7.57 | 29.72 | 6.08 | 23.23 | 4.89 | 29.67 | 3.55 | 20.28 | 10.87 | 50.41 | 9.32 | 39.92 | 9.18 | 39.09 | 7.35 | 29.80 |
| +CoT | 11.44 | 35.20 | 10.63 | 31.88 | 6.70 | 23.21 | 6.24 | 22.09 | 3.55 | 18.94 | 3.30 | 16.22 | 8.42 | 41.17 | 8.22 | 37.33 | 7.53 | 29.63 | 7.10 | 26.59 |
| Qwen2.5 7B Instruct | 14.76 | 48.91 | 11.97 | 40.36 | 7.43 | 27.84 | 6.78 | 24.29 | 5.90 | 30.87 | 4.89 | 23.44 | 13.45 | 53.40 | 11.61 | 45.35 | 10.38 | 40.26 | 8.81 | 33.36 |
| +CoT | 16.30 | 54.66 | 13.40 | 44.62 | 10.00 | 40.78 | 7.90 | 31.20 | 7.53 | 38.40 | 5.11 | 27.36 | 10.33 | 60.46 | 9.06 | 50.84 | 11.04 | 48.58 | 8.87 | 38.50 |
| Qwen2.5 14B Instruct | 18.33 | 60.02 | 15.83 | 50.33 | 12.71 | 42.84 | 10.71 | 34.42 | 8.01 | 41.80 | 6.15 | 30.67 | 13.32 | 66.30 | 12.09 | 58.36 | 13.09 | 52.74 | 11.20 | 43.44 |
| +CoT | 19.22 | 59.61 | 17.00 | 53.33 | 10.83 | 36.15 | 10.02 | 31.89 | 7.19 | 37.34 | 5.91 | 29.87 | 13.32 | 60.87 | 12.16 | 56.01 | 12.64 | 48.49 | 11.27 | 42.78 |
| Qwen3 4B Instruct 2507 | 16.95 | 60.91 | 13.34 | 47.10 | 10.60 | 43.12 | 8.21 | 31.89 | 6.86 | 42.14 | 4.74 | 29.25 | 11.14 | 66.03 | 9.57 | 54.83 | 11.39 | 53.05 | 8.96 | 40.77 |
| +CoT | 19.38 | 61.64 | 16.76 | 52.60 | 10.78 | 37.20 | 9.22 | 30.19 | 7.29 | 42.57 | 5.69 | 32.10 | 12.91 | 58.15 | 11.80 | 51.27 | 12.59 | 49.89 | 10.87 | 41.54 |
| Gemma 3 4B IT | 17.52 | 53.93 | 12.93 | 40.65 | 10.60 | 38.58 | 7.40 | 26.52 | 7.14 | 38.11 | 4.63 | 25.88 | 11.55 | 58.83 | 9.94 | 46.17 | 11.77 | 47.36 | 8.72 | 34.80 |
| +CoT | 12.81 | 45.26 | 10.17 | 36.10 | 7.25 | 27.75 | 5.61 | 20.44 | 5.56 | 28.04 | 3.90 | 20.56 | 11.55 | 47.28 | 9.07 | 37.90 | 9.29 | 37.08 | 7.19 | 28.75 |
| *Open-weight Medical LLMs* | | | | | | | | | | | | | | | | | | | | |
| MedGemma 4B IT | 18.09 | 60.58 | 13.62 | 46.27 | 13.81 | 44.54 | 9.07 | 29.71 | 8.29 | 42.43 | 5.11 | 27.58 | 13.18 | 65.08 | 10.52 | 51.55 | 13.34 | 53.16 | 9.58 | 38.78 |
| +CoT | 20.44 | 61.64 | 15.73 | 47.09 | 12.89 | 42.48 | 8.91 | 27.84 | 8.96 | 40.84 | 4.98 | 25.00 | 11.96 | 57.47 | 9.70 | 44.89 | 13.56 | 50.61 | 9.83 | 36.20 |
| MedGemma 27B IT | 21.98 | 63.50 | 19.30 | 55.73 | 13.21 | 38.76 | 11.31 | 32.65 | 9.16 | 40.65 | 7.22 | 31.15 | 13.45 | 59.51 | 13.29 | 53.44 | 14.45 | 50.60 | 12.78 | 43.24 |
| +CoT | 22.30 | 66.10 | 20.33 | 58.91 | 12.80 | 39.91 | 11.20 | 33.39 | 10.55 | 43.53 | 8.36 | 34.38 | 15.00 | 63.18 | 14.21 | 57.07 | 15.18 | 53.18 | 13.52 | 45.94 |
| *Open-weight Reasoning Models* | | | | | | | | | | | | | | | | | | | | |
| OpenThinker3 7B | 6.97 | 33.25 | 6.39 | 29.15 | 4.51 | 22.25 | 4.10 | 19.12 | 2.11 | 18.07 | 1.77 | 14.19 | 7.61 | 40.76 | 7.34 | 36.25 | 5.44 | 28.58 | 5.00 | 24.68 |
| +CoT | 0.16 | 0.81 | 0.16 | 0.77 | 0.14 | 0.50 | 0.10 | 0.39 | 0.05 | 0.24 | 0.05 | 0.24 | 0.14 | 2.17 | 0.14 | 1.94 | 0.12 | 0.93 | 0.11 | 0.84 |
| *Open-weight Medical Reasoning Models* | | | | | | | | | | | | | | | | | | | | |
| HuatuoGPT o1 7B | 13.06 | 36.66 | 10.27 | 29.42 | 11.56 | 36.47 | 8.51 | 26.50 | 4.89 | 23.01 | 3.53 | 16.21 | 11.01 | 43.48 | 9.17 | 36.07 | 10.13 | 34.90 | 7.87 | 27.05 |
| +CoT | 0.57 | 1.62 | 0.51 | 1.47 | 0.32 | 1.06 | 0.26 | 0.91 | 0.14 | 0.62 | 0.14 | 0.47 | 0.14 | 1.09 | 0.14 | 1.02 | 0.29 | 1.10 | 0.26 | 0.97 |
| m1 7B 23K | 15.33 | 56.04 | 11.65 | 42.59 | 11.93 | 41.93 | 8.29 | 29.73 | 7.00 | 39.50 | 4.66 | 26.10 | 9.92 | 59.65 | 8.46 | 47.23 | 11.04 | 49.28 | 8.26 | 36.41 |
| +CoT | 7.95 | 40.39 | 6.81 | 35.17 | 5.41 | 34.36 | 4.07 | 25.80 | 3.21 | 26.03 | 2.18 | 19.08 | 6.25 | 44.57 | 5.42 | 38.89 | 5.70 | 36.34 | 4.62 | 29.74 |
| AlphaMed 7B Instruct RL | 4.30 | 18.25 | 2.99 | 13.12 | 3.99 | 14.29 | 2.95 | 10.60 | 2.30 | 12.56 | 1.44 | 8.13 | 7.34 | 32.74 | 5.69 | 25.78 | 4.48 | 19.46 | 3.27 | 14.41 |
| +CoT | 6.00 | 19.71 | 4.59 | 15.42 | 3.94 | 18.39 | 3.27 | 14.08 | 2.16 | 14.09 | 1.49 | 10.24 | 5.30 | 30.57 | 4.52 | 24.97 | 4.35 | 20.69 | 3.47 | 16.18 |

Table 18: Response length (mean $\pm$ standard deviation) for **MCQ** answer format prompting across benchmarks from Section 3.

| | MedQA | MedMCQA | MXQA | MMLU Pro-H | Average |
|---|---|---|---|---|---|
| *Proprietary Reasoning Models* | | | | | |
| Gemini 2.5 Flash Lite | 1257.47 ± 3191.23 | 1119.54 ± 3512.21 | 2628.94 ± 4957.93 | 1335.20 ± 3220.06 | 1585.29 |
| +CoT | 2818.61 ± 4419.52 | 2139.62 ± 4667.54 | 8736.47 ± 7822.16 | 3745.60 ± 5907.18 | 4360.07 |
| Gemini 2.5 Flash | 545.11 ± 197.13 | 123.44 ± 192.86 | 844.54 ± 266.83 | 377.23 ± 346.71 | 472.58 |
| +CoT | 1587.31 ± 1154.53 | 1033.27 ± 1025.54 | 2967.20 ± 2043.39 | 1449.88 ± 1243.58 | 1759.41 |
| Gemini 2.5 Pro | 490.35 ± 138.99 | 248.95 ± 127.13 | 627.53 ± 270.05 | 334.02 ± 217.40 | 425.21 |
| +CoT | 1465.06 ± 266.79 | 954.33 ± 272.60 | 1631.28 ± 390.26 | 1252.48 ± 375.49 | 1325.79 |
| *Proprietary Large Language Models* | | | | | |
| GPT-4.1 Mini | 415.21 ± 150.39 | 155.87 ± 75.99 | 539.88 ± 224.97 | 302.50 ± 171.92 | 353.37 |
| +CoT | 382.19 ± 118.66 | 200.29 ± 67.40 | 509.30 ± 165.14 | 334.75 ± 138.41 | 356.63 |
| *Open-weight Large Language Models* | | | | | |
| Qwen2.5 3B Instruct | 355.70 ± 247.30 | 209.95 ± 233.14 | 197.95 ± 235.02 | 318.69 ± 169.92 | 270.57 |
| +CoT | 368.60 ± 328.27 | 263.84 ± 257.55 | 431.82 ± 501.80 | 381.64 ± 133.37 | 361.48 |
| Qwen2.5 7B Instruct | 112.42 ± 83.21 | 39.83 ± 48.06 | 83.12 ± 70.98 | 53.92 ± 68.11 | 72.32 |
| +CoT | 1066.64 ± 2476.43 | 1819.36 ± 3206.03 | 3376.30 ± 3906.91 | 3310.99 ± 3829.30 | 2393.32 |
| Qwen2.5 14B Instruct | 169.53 ± 81.65 | 89.49 ± 65.35 | 149.53 ± 68.53 | 108.68 ± 72.01 | 129.31 |
| +CoT | 227.06 ± 79.52 | 171.93 ± 190.53 | 239.99 ± 104.90 | 228.72 ± 116.49 | 216.93 |
| Qwen3 4B Instruct 2507 | 917.31 ± 1093.89 | 364.78 ± 640.04 | 1074.26 ± 899.40 | 669.64 ± 835.01 | 756.50 |
| +CoT | 1055.03 ± 1189.65 | 536.58 ± 793.92 | 1191.33 ± 936.30 | 794.32 ± 797.72 | 894.32 |
| Gemma 3 4B IT | 407.36 ± 98.68 | 255.02 ± 81.38 | 529.57 ± 136.83 | 406.46 ± 131.04 | 399.60 |
| +CoT | 403.89 ± 782.49 | 273.21 ± 607.06 | 425.57 ± 544.08 | 424.09 ± 740.47 | 381.69 |
| *Open-weight Medical LLMs* | | | | | |
| MedGemma 4B IT | 376.22 ± 992.80 | 122.10 ± 541.86 | 427.47 ± 1056.55 | 263.80 ± 646.20 | 297.40 |
| +CoT | 7657.11 ± 1975.69 | 8060.86 ± 1009.42 | 8188.78 ± 155.49 | 7924.15 ± 1423.73 | 7957.73 |
| MedGemma 27B IT | 1193.72 ± 855.60 | 572.05 ± 732.17 | 1585.79 ± 1010.40 | 974.36 ± 789.28 | 1081.48 |
| +CoT | 1419.63 ± 826.18 | 852.78 ± 783.07 | 2090.85 ± 1027.10 | 1331.33 ± 874.37 | 1423.65 |
| *Open-weight Reasoning Models* | | | | | |
| OpenThinker3 7B | 4386.88 ± 2777.18 | 3672.76 ± 2787.49 | 6164.33 ± 2439.96 | 4933.49 ± 2773.43 | 4789.36 |
| +CoT | 7538.26 ± 1933.86 | 7284.63 ± 2249.75 | 7139.14 ± 2387.52 | 7399.08 ± 1985.14 | 7340.28 |
| *Open-weight Medical Reasoning Models* | | | | | |
| HuatuoGPT o1 7B | 501.45 ± 83.88 | 399.07 ± 124.61 | 515.03 ± 179.38 | 472.94 ± 287.54 | 472.12 |
| +CoT | 524.25 ± 315.82 | 410.86 ± 189.21 | 535.78 ± 238.44 | 479.55 ± 96.68 | 487.61 |
| m1 7B 23K | 1423.43 ± 2067.20 | 1117.32 ± 1798.36 | 1958.03 ± 2502.58 | 1811.57 ± 2445.53 | 1577.59 |
| +CoT | 8185.63 ± 195.05 | 8184.38 ± 214.57 | 8188.19 ± 135.88 | 8182.06 ± 262.03 | 8185.07 |
| AlphaMed 7B Instruct RL | 336.71 ± 252.38 | 197.61 ± 155.88 | 362.00 ± 210.89 | 346.32 ± 129.34 | 310.66 |
| +CoT | 400.09 ± 330.13 | 322.74 ± 357.22 | 489.54 ± 744.54 | 452.29 ± 488.71 | 416.16 |

Table 19: Response length (mean ± standard deviation) for **QA** answer format prompting across benchmarks from Section 3.

| | MedQA | MedMCQA | MXQA | MMLU Pro-H | Average |
|---|---|---|---|---|---|
| *Proprietary Reasoning Models* | | | | | |
| Gemini 2.5 Flash Lite | 539.03 ± 713.66 | 366.96 ± 1111.66 | 752.54 ± 1854.21 | 386.84 ± 306.99 | 511.34 |
| +CoT | 2578.72 ± 3754.91 | 2193.28 ± 4331.44 | 4146.62 ± 5493.75 | 2481.86 ± 4558.74 | 2850.12 |
| Gemini 2.5 Flash | 312.46 ± 224.98 | 181.34 ± 184.48 | 341.99 ± 238.11 | 258.09 ± 232.62 | 273.47 |
| +CoT | 1637.81 ± 980.09 | 1206.37 ± 885.25 | 2051.34 ± 1222.96 | 1406.75 ± 1055.64 | 1575.57 |
| Gemini 2.5 Pro | 499.26 ± 286.12 | 363.56 ± 278.51 | 545.75 ± 290.95 | 428.83 ± 298.50 | 459.35 |
| +CoT | 1555.26 ± 463.10 | 1402.16 ± 784.48 | 1655.74 ± 479.10 | 1494.77 ± 545.76 | 1526.98 |
| *Proprietary Large Language Models* | | | | | |
| GPT-4.1 Mini | 340.36 ± 178.95 | 179.09 ± 103.89 | 388.24 ± 199.08 | 231.40 ± 148.30 | 284.77 |
| +CoT | 320.84 ± 119.89 | 238.50 ± 85.35 | 355.43 ± 134.23 | 253.05 ± 102.18 | 291.95 |
| *Open-weight Large Language Models* | | | | | |
| Qwen2.5 3B Instruct | 246.31 ± 343.36 | 170.46 ± 268.18 | 245.44 ± 279.15 | 193.74 ± 123.33 | 213.99 |
| +CoT | 314.92 ± 117.15 | 307.22 ± 595.46 | 340.17 ± 402.12 | 296.82 ± 426.34 | 314.78 |
| Qwen2.5 7B Instruct | 171.16 ± 251.37 | 108.60 ± 89.28 | 170.38 ± 108.71 | 133.57 ± 101.85 | 145.93 |
| +CoT | 214.56 ± 82.16 | 164.49 ± 252.89 | 225.74 ± 85.32 | 179.28 ± 78.02 | 196.02 |
| Qwen2.5 14B Instruct | 160.09 ± 244.07 | 118.92 ± 78.74 | 156.92 ± 92.65 | 122.62 ± 78.94 | 139.64 |
| +CoT | 217.01 ± 75.96 | 195.16 ± 305.39 | 240.56 ± 261.87 | 199.52 ± 303.65 | 213.06 |
| Qwen3 4B Instruct 2507 | 474.31 ± 438.88 | 419.78 ± 1037.35 | 563.43 ± 679.78 | 341.24 ± 485.06 | 449.69 |
| +CoT | 565.40 ± 482.77 | 521.64 ± 1000.22 | 624.52 ± 600.58 | 441.24 ± 500.05 | 538.20 |
| Gemma 3 4B IT | 480.07 ± 201.84 | 339.02 ± 188.74 | 504.22 ± 198.02 | 405.77 ± 210.79 | 432.27 |
| +CoT | 343.19 ± 150.29 | 251.72 ± 210.98 | 343.45 ± 154.01 | 275.65 ± 136.41 | 303.50 |
| *Open-weight Medical LLMs* | | | | | |
| MedGemma 4B IT | 166.35 ± 345.02 | 108.60 ± 528.33 | 183.20 ± 381.67 | 150.28 ± 531.34 | 152.11 |
| +CoT | 343.13 ± 962.56 | 600.20 ± 1810.89 | 526.34 ± 1473.62 | 502.38 ± 1534.13 | 493.01 |
| MedGemma 27B IT | 797.26 ± 678.04 | 395.62 ± 483.65 | 901.12 ± 929.02 | 518.22 ± 454.65 | 653.06 |
| +CoT | 1020.20 ± 863.48 | 849.69 ± 1055.90 | 1168.26 ± 1020.78 | 826.97 ± 854.14 | 966.28 |
| *Open-weight Reasoning Models* | | | | | |
| OpenThinker3 7B | 4372.27 ± 2887.58 | 4053.37 ± 2843.18 | 5123.57 ± 2780.84 | 3843.64 ± 2864.54 | 4348.21 |
| +CoT | 4483.61 ± 2925.64 | 4330.32 ± 2859.00 | 5184.47 ± 2810.99 | 3801.95 ± 2852.59 | 4450.09 |
| *Open-weight Medical Reasoning Models* | | | | | |
| HuatuoGPT o1 7B | 509.17 ± 92.77 | 456.40 ± 299.47 | 530.70 ± 252.94 | 484.12 ± 413.76 | 495.10 |
| +CoT | 532.43 ± 323.88 | 455.65 ± 84.25 | 543.79 ± 303.99 | 471.90 ± 96.36 | 500.94 |
| m1 7B 23K | 1628.53 ± 2083.86 | 1376.54 ± 1753.36 | 1909.36 ± 2242.62 | 1254.38 ± 1786.37 | 1542.20 |
| +CoT | 2355.00 ± 2511.52 | 2068.92 ± 2256.22 | 2795.45 ± 2593.80 | 2091.34 ± 2332.12 | 2327.68 |
| AlphaMed 7B Instruct RL | 244.48 ± 406.33 | 186.20 ± 312.77 | 235.77 ± 272.32 | 198.74 ± 108.44 | 216.30 |
| +CoT | 284.31 ± 605.91 | 223.78 ± 549.05 | 309.03 ± 678.04 | 245.06 ± 595.63 | 265.55 |

Table 20: Response length (mean $\pm$ standard deviation) for ranked-list answer format prompting across benchmarks from Section 3.

| | MedQA | MedMCQA | MXQA | MMLU Pro-H | Average |
|---|---|---|---|---|---|
| *Proprietary Reasoning Models* | | | | | |
| Gemini 2.5 Flash Lite | 98.61 ± 744.16 | 88.90 ± 997.32 | 135.87 ± 1085.66 | 50.13 ± 111.02 | 93.38 |
| +CoT | 7697.65 ± 8361.72 | 5191.43 ± 7473.53 | 10339.69 ± 8598.17 | 6364.13 ± 8108.09 | 7398.23 |
| Gemini 2.5 Flash | 34.67 ± 28.84 | 21.87 ± 24.46 | 39.85 ± 41.35 | 31.26 ± 38.97 | 31.91 |
| +CoT | 1090.39 ± 793.30 | 820.04 ± 755.68 | 1209.99 ± 1023.00 | 962.94 ± 854.95 | 1020.84 |
| Gemini 2.5 Pro | 41.62 ± 45.35 | 26.98 ± 41.48 | 44.23 ± 45.51 | 41.07 ± 88.56 | 38.47 |
| +CoT | 1073.70 ± 254.87 | 885.01 ± 275.27 | 1133.54 ± 264.44 | 985.43 ± 332.91 | 1019.42 |
| *Proprietary Large Language Models* | | | | | |
| GPT-4.1 Mini | 163.78 ± 98.56 | 86.79 ± 60.53 | 186.19 ± 107.64 | 111.20 ± 77.52 | 136.99 |
| +CoT | 282.98 ± 100.23 | 206.23 ± 69.57 | 302.09 ± 99.34 | 216.46 ± 80.95 | 251.94 |
| *Open-weight Large Language Models* | | | | | |
| Qwen2.5 3B Instruct | 42.18 ± 36.45 | 24.36 ± 20.55 | 46.25 ± 38.97 | 33.09 ± 30.90 | 36.47 |
| +CoT | 180.81 ± 329.07 | 157.29 ± 350.21 | 176.33 ± 188.08 | 153.23 ± 67.16 | 166.92 |
| Qwen2.5 7B Instruct | 30.90 ± 18.33 | 19.26 ± 13.26 | 33.21 ± 21.03 | 26.23 ± 18.85 | 27.40 |
| +CoT | 4705.88 ± 3976.94 | 3723.11 ± 4008.96 | 4801.62 ± 3962.49 | 4504.85 ± 4015.31 | 4433.86 |
| Qwen2.5 14B Instruct | 37.05 ± 20.16 | 26.90 ± 16.62 | 42.33 ± 25.19 | 31.74 ± 21.19 | 34.51 |
| +CoT | 167.78 ± 54.69 | 142.31 ± 248.58 | 178.23 ± 58.97 | 162.10 ± 422.30 | 162.61 |
| Qwen3 4B Instruct 2507 | 61.19 ± 48.45 | 98.58 ± 676.35 | 69.45 ± 93.76 | 47.98 ± 39.37 | 69.30 |
| +CoT | 490.88 ± 778.15 | 431.86 ± 1009.89 | 500.40 ± 647.48 | 409.64 ± 796.57 | 458.19 |
| Gemma 3 4B IT | 60.97 ± 88.76 | 56.17 ± 71.18 | 57.65 ± 68.12 | 70.92 ± 310.54 | 61.43 |
| +CoT | 272.78 ± 568.13 | 228.08 ± 566.71 | 291.60 ± 637.28 | 210.28 ± 336.13 | 250.69 |
| *Open-weight Medical LLMs* | | | | | |
| MedGemma 4B IT | 885.26 ± 2464.30 | 1032.49 ± 2662.78 | 966.32 ± 2566.97 | 921.43 ± 2523.40 | 951.38 |
| +CoT | 8179.20 ± 310.00 | 8177.36 ± 334.93 | 8180.46 ± 297.15 | 8181.04 ± 293.15 | 8179.52 |
| MedGemma 27B IT | 75.15 ± 363.27 | 33.21 ± 249.06 | 66.23 ± 202.91 | 33.26 ± 42.59 | 51.96 |
| +CoT | 1011.49 ± 842.55 | 945.87 ± 1202.11 | 1237.82 ± 1249.42 | 866.86 ± 904.18 | 1015.51 |
| *Open-weight Reasoning Models* | | | | | |
| OpenThinker3 7B | 4915.88 ± 3022.94 | 4411.53 ± 3028.07 | 5495.49 ± 2829.85 | 4154.27 ± 3049.39 | 4744.29 |
| +CoT | 7842.04 ± 1404.18 | 7561.84 ± 1881.72 | 7737.29 ± 1577.02 | 7377.80 ± 2054.35 | 7629.74 |
| *Open-weight Medical Reasoning Models* | | | | | |
| HuatuoGPT o1 7B | 55.94 ± 330.24 | 34.55 ± 176.57 | 61.56 ± 358.68 | 36.49 ± 31.11 | 47.14 |
| +CoT | 395.63 ± 275.05 | 361.73 ± 326.84 | 404.67 ± 150.04 | 336.29 ± 335.71 | 374.58 |
| m1 7B 23K | 1629.10 ± 2926.38 | 1564.04 ± 2882.66 | 1733.96 ± 2945.14 | 1702.78 ± 2966.04 | 1657.47 |
| +CoT | 8146.77 ± 548.03 | 8149.01 ± 552.57 | 8166.48 ± 418.94 | 8133.05 ± 648.74 | 8148.83 |
| AlphaMed 7B Instruct RL | 20.58 ± 28.28 | 17.68 ± 175.47 | 20.46 ± 32.36 | 17.26 ± 17.24 | 19.00 |
| +CoT | 1340.47 ± 2803.22 | 852.62 ± 2297.19 | 1033.94 ± 2461.34 | 979.52 ± 2451.48 | 1051.64 |

Table 21: Metrics related to the ranked **list** answer format from the generated evaluation responses across benchmarks from Section 3. **CP** denotes the average rank position of the correct item within the generated list, **LL** represents the average list length across all responses, and **VLL** corresponds to the average valid list length, computed only over non-empty outputs.

| | MedQA | | | MedMCQA | | | MXQA | | | MLUP-H | | |
|---|---|---|---|---|---|---|---|---|---|---|---|---|
| | LL | VLL | CP | LL | VLL | CP | LL | VLL | CP | LL | VLL | CP |
| *Proprietary Reasoning Models* | | | | | | | | | | | | |
| Gemini 2.5 Flash Lite | 2.84 | 3.00 | 1.27 | 2.70 | 2.73 | 1.46 | 2.96 | 3.07 | 1.59 | 2.59 | 2.61 | 1.22 |
| +CoT | 0.75 | 2.05 | 1.10 | 1.16 | 1.76 | 1.17 | 0.60 | 2.11 | 1.32 | 0.96 | 1.74 | 1.09 |
| Gemini 2.5 Flash | 3.26 | 3.31 | 1.34 | 2.42 | 2.42 | 1.44 | 3.39 | 3.42 | 1.53 | 2.86 | 2.87 | 1.32 |
| +CoT | 1.08 | 2.92 | 1.30 | 1.33 | 2.28 | 1.26 | 0.92 | 2.93 | 1.34 | 1.30 | 2.53 | 1.28 |
| Gemini 2.5 Pro | 3.66 | 3.71 | 1.27 | 3.02 | 3.02 | 1.54 | 3.58 | 3.62 | 1.48 | 3.28 | 3.30 | 1.30 |
| +CoT | 3.56 | 3.69 | 1.28 | 2.92 | 3.09 | 1.56 | 3.51 | 3.69 | 1.47 | 3.20 | 3.37 | 1.34 |
| *Proprietary Large Language Models* | | | | | | | | | | | | |
| GPT-4.1 Mini | 3.44 | 3.59 | 1.21 | 2.90 | 2.90 | 1.48 | 3.40 | 3.46 | 1.50 | 3.07 | 3.09 | 1.24 |
| +CoT | 3.99 | 4.00 | 1.22 | 3.39 | 3.39 | 1.51 | 3.88 | 3.88 | 1.49 | 3.59 | 3.59 | 1.22 |
| *Open-weight Large Language Models* | | | | | | | | | | | | |
| Qwen2.5 3B Instruct | 3.56 | 3.56 | 1.67 | 2.42 | 2.42 | 1.57 | 3.53 | 3.53 | 1.95 | 2.86 | 2.86 | 1.56 |
| +CoT | 1.79 | 1.83 | 1.24 | 1.61 | 1.61 | 1.25 | 1.90 | 1.93 | 1.35 | 1.83 | 1.84 | 1.23 |
| Qwen2.5 7B Instruct | 2.73 | 2.73 | 1.43 | 1.84 | 1.84 | 1.34 | 2.68 | 2.68 | 1.61 | 2.29 | 2.29 | 1.41 |
| +CoT | 195.49 | 196.93 | 1.65 | 153.75 | 154.89 | 2.04 | 210.52 | 212.15 | 2.25 | 176.98 | 178.92 | 1.70 |
| Qwen2.5 14B Instruct | 3.23 | 3.25 | 1.43 | 2.76 | 2.76 | 1.52 | 3.28 | 3.29 | 1.70 | 2.89 | 2.89 | 1.29 |
| +CoT | 2.57 | 2.57 | 1.27 | 2.07 | 2.07 | 1.28 | 2.57 | 2.57 | 1.49 | 2.24 | 2.24 | 1.17 |
| Qwen3 4B Instruct 2507 | 4.42 | 4.42 | 1.68 | 3.40 | 3.40 | 1.75 | 4.38 | 4.38 | 1.90 | 3.64 | 3.64 | 1.48 |
| +CoT | 3.75 | 3.95 | 1.41 | 2.69 | 2.93 | 1.53 | 3.76 | 3.96 | 1.74 | 3.10 | 3.41 | 1.32 |
| Gemma 3 4B IT | 4.40 | 4.58 | 1.78 | 4.00 | 4.02 | 1.94 | 4.29 | 4.40 | 1.94 | 5.66 | 5.72 | 1.64 |
| +CoT | 3.60 | 4.22 | 1.58 | 2.94 | 3.63 | 1.75 | 3.79 | 4.64 | 1.78 | 3.37 | 4.03 | 1.57 |
| *Open-weight Medical LLMs* | | | | | | | | | | | | |
| MedGemma 4B IT | 90.18 | 90.33 | 2.12 | 104.13 | 104.13 | 2.33 | 87.22 | 87.31 | 2.43 | 98.49 | 98.49 | 1.72 |
| +CoT | 491.91 | 546.91 | 2.44 | 331.35 | 387.53 | 3.30 | 429.49 | 515.78 | 3.54 | 423.99 | 480.82 | 3.26 |
| MedGemma 27B IT | 3.66 | 3.82 | 1.37 | 2.99 | 2.99 | 1.48 | 3.29 | 3.36 | 1.72 | 2.86 | 2.89 | 1.28 |
| +CoT | 3.95 | 4.01 | 1.31 | 3.14 | 3.21 | 1.53 | 4.70 | 4.83 | 1.61 | 3.44 | 3.49 | 1.25 |
| *Open-weight Reasoning Models* | | | | | | | | | | | | |
| OpenThinker3 7B | 2.25 | 3.45 | 1.36 | 1.99 | 2.73 | 1.38 | 1.84 | 3.04 | 1.60 | 2.17 | 3.01 | 1.37 |
| +CoT | 0.07 | 1.84 | 1.00 | 0.13 | 2.30 | 1.83 | 0.57 | 13.09 | 1.00 | 0.16 | 1.80 | 1.33 |
| *Open-weight Medical Reasoning Models* | | | | | | | | | | | | |
| HuatuoGPT o1 7B | 3.14 | 5.36 | 1.58 | 3.21 | 3.76 | 1.84 | 2.46 | 4.81 | 1.89 | 2.41 | 3.64 | 1.50 |
| +CoT | 0.04 | 2.61 | 1.16 | 0.03 | 1.99 | 1.54 | 0.03 | 3.12 | 2.00 | 0.02 | 2.07 | 1.12 |
| m1 7B 23K | 14.44 | 14.44 | 1.85 | 9.82 | 9.82 | 2.06 | 18.23 | 18.23 | 2.36 | 11.06 | 11.06 | 1.77 |
| +CoT | 13.00 | 17.14 | 1.45 | 14.85 | 18.01 | 1.83 | 16.22 | 22.58 | 1.89 | 14.42 | 18.39 | 1.37 |
| AlphaMed 7B Instruct RL | 2.00 | 2.68 | 1.92 | 1.58 | 2.35 | 1.72 | 2.03 | 2.73 | 2.08 | 2.11 | 2.61 | 1.58 |
| +CoT | 61.01 | 71.04 | 1.68 | 35.16 | 40.67 | 1.84 | 50.63 | 60.62 | 2.42 | 44.65 | 50.64 | 1.60 |

Table 22: Results table for SFT experiments evaluated on the benchmarks with **MCQ** answer format from Section 4.1. MXQA refers to MedXpertQA (text), and MLUP-H refers to MMLU Pro (Health).

|  | MedQA | MedMCQA | MXQA | MLUP-H | Average |
|---|---|---|---|---|---|
| SFT-MCQ | 66.46 | 18.44 | 14.20 | 59.29 | 39.60 |
| +CoT | **70.23** | **19.09** | **15.71** | **61.12** | **41.54** |
| SFT-QA | 67.32 | 15.59 | 13.63 | 54.16 | 37.67 |
| +CoT | 66.06 | 16.08 | 13.18 | 55.87 | 37.80 |
| SFT-List | 26.24 | 0.31 | 5.67 | 10.51 | 10.68 |
| +CoT | 29.85 | 0.46 | 6.61 | 13.57 | 12.62 |

Table 23: Results table for SFT experiments evaluated on the benchmarks with **QA** answer format from Section 4.1. MXQA refers to MedXpertQA (text), and MLUP-H refers to MMLU Pro (Health).

|  | MedQA | | MedMCQA | | MXQA | | MLUP-H | | Average | |
|---|---|---|---|---|---|---|---|---|---|---|
|  | $Acc_{QA}$ | $Acc_{QA}^{LLM}$ | $Acc_{QA}$ | $Acc_{QA}^{LLM}$ | $Acc_{QA}$ | $Acc_{QA}^{LLM}$ | $Acc_{QA}$ | $Acc_{QA}^{LLM}$ | $Acc_{QA}$ | $Acc_{QA}^{LLM}$ |
| SFT-QA | 18.25 | 52.72 | 7.75 | 42.48 | 5.27 | 40.41 | 13.72 | 50.68 | 11.25 | 46.57 |
| +CoT | 18.09 | 53.28 | 8.35 | 42.11 | 4.89 | 40.17 | 13.72 | 52.58 | 11.26 | 47.04 |
| SFT-MCQ | 17.19 | 53.69 | 8.26 | 43.53 | **5.42** | **41.56** | 13.45 | **53.40** | 11.08 | **48.04** |
| +CoT | **19.14** | **56.85** | **8.62** | **43.99** | 4.84 | 38.59 | **14.27** | 50.82 | **11.72** | 47.56 |
| SFT-List | 0.32 | 0.97 | 0.05 | 0.64 | 0.19 | 1.63 | 0.41 | 1.36 | 0.24 | 1.15 |
| +CoT | 0.41 | 1.95 | 0.23 | 0.78 | 0.24 | 1.87 | 0.82 | 2.58 | 0.42 | 1.80 |

per benchmark. Table 21 shows metrics related to list responses, e.g., list length and position of a correct item.

## N.2 FINE-TUNING

### N.2.1 SFT

Tables 22 to 24 present the non-aggregated versions of the aggregated results shown in the main body of this study (Table 1 in Section 4.1). While, Tables 26 to 28 show average response length per benchmark. Table 29 shows metrics related to list responses, e.g., list length and position of a correct item.

### N.2.2 RFT

Tables 30 to 32 present the non-aggregated versions of the aggregated results shown in the main body of this study (Table 1 in Section 4.2). While, Tables 33 to 35 show average response length per benchmark. Table 36 shows metrics related to list responses, e.g., list length and position of a correct item.

## O QUALITATIVE EXAMPLES

Figure 28 shows an example in which HuatuoGPT-o1 fails to follow the instruction to place the final answer inside a boxed environment for the question: "A junior orthopaedic surgery resident is completing a carpal tunnel repair with the department chairman as the attending physician. During the case, the resident inadvertently cuts a flexor tendon. The tendon is repaired without complication.

Table 24: Results table for SFT experiments evaluated on the benchmarks with a ranked **list** answer format from Section 4.1. MXQA refers to MedXpertQA (text), and MLUP-H refers to MMLU Pro (Health).

|  | MedQA | | | | MedMCQA | | | | MXQA | | | | MLUP-H | | | | Average | | | |
|---|---|---|---|---|---|---|---|---|---|---|---|---|---|---|---|---|---|---|---|---|
|  | $Acc_{List}$ | $Acc_{List}^{LLM}$ | $MRR_{List}$ | $MRR_{List}^{LLM}$ | $Acc_{List}$ | $Acc_{List}^{LLM}$ | $MRR_{List}$ | $MRR_{List}^{LLM}$ | $Acc_{List}$ | $Acc_{List}^{LLM}$ | $MRR_{List}$ | $MRR_{List}^{LLM}$ | $Acc_{List}$ | $Acc_{List}^{LLM}$ | $MRR_{List}$ | $MRR_{List}^{LLM}$ | $Acc_{List}$ | $Acc_{List}^{LLM}$ | $MRR_{List}$ | $MRR_{List}^{LLM}$ |
| SFT-List | **24.90** | 61.64 | **22.40** | **54.89** | 11.33 | 36.42 | 9.76 | 30.32 | 8.92 | 37.73 | 6.96 | **29.40** | 15.90 | 60.60 | **14.83** | **54.30** | 15.26 | 49.10 | **13.49** | **42.23** |
| +CoT | 24.01 | 60.50 | 21.63 | 53.91 | 11.10 | 37.75 | 9.71 | 31.38 | 8.96 | 37.87 | **7.00** | 28.86 | 15.76 | 60.19 | 14.79 | 52.27 | 14.96 | 49.08 | 13.28 | 41.60 |
| SFT-MCQ | 20.84 | **63.42** | 15.36 | 45.91 | **15.50** | **51.28** | **10.18** | **33.16** | **9.97** | **46.26** | 6.36 | 28.67 | 15.62 | **69.02** | 12.79 | 51.00 | **15.48** | **57.50** | 11.17 | 39.68 |
| +CoT | 17.84 | 47.69 | 17.43 | 46.31 | 8.39 | 25.37 | 8.37 | 24.59 | 5.90 | 24.16 | 5.68 | 22.58 | 13.86 | 47.15 | 13.86 | 46.00 | 11.50 | 36.09 | 11.34 | 34.87 |
| SFT-QA | 18.41 | 54.66 | 15.22 | 44.62 | 12.84 | 46.56 | 9.22 | 32.77 | 7.19 | 38.30 | 5.31 | 27.17 | 13.32 | 65.76 | 11.15 | 51.53 | 12.94 | 51.32 | 10.22 | 39.02 |
| +CoT | 12.49 | 31.87 | 12.17 | 30.62 | 6.01 | 18.39 | 5.82 | 17.54 | 3.64 | 14.81 | 3.43 | 13.36 | 9.38 | 33.29 | 9.24 | 32.22 | 7.88 | 24.59 | 7.67 | 23.44 |

Table 25: Results table for prompting experiments evaluated on open-ended benchmarks using **List** prompting strategies. The highest overall score in each benchmark column is shown in **bold**.

| Model | MedQA | | MedMCQA | | MXQA | | MLUP-H | | Average | |
|---|---|---|---|---|---|---|---|---|---|---|
| | $Acc_{List}^{LLM}$ | $MRR_{List}^{LLM}$ | $Acc_{List}^{LLM}$ | $MRR_{List}^{LLM}$ | $Acc_{List}^{LLM}$ | $MRR_{List}^{LLM}$ | $Acc_{List}^{LLM}$ | $MRR_{List}^{LLM}$ | $Acc_{List}^{LLM}$ | $MRR_{List}^{LLM}$ |
| SFT-List | 61.64 | **54.89** | 36.42 | 30.32 | 37.73 | **29.40** | 60.60 | **54.30** | 49.10 | **42.23** |
| SFT-MCQ | **63.42** | 45.91 | **51.28** | **33.16** | **46.26** | 28.67 | **69.02** | 51.00 | **57.50** | 39.68 |
| SFT-QA | 54.66 | 44.62 | 46.56 | 32.77 | 38.30 | 27.17 | 65.76 | 51.53 | 51.32 | 39.02 |

Table 26: Response length (mean ± standard deviation) for **MCQ** answer format for the experiments from Section 4.1.

| | MedQA | MedMCQA | MXQA | MLUP-H | Average |
|---|---|---|---|---|---|
| SFT-MCQ | 3332.74 ± 1671.04 | 1344.50 ± 1316.94 | 3918.01 ± 1859.36 | 2399.41 ± 1657.66 | 2748.66 |
| +CoT | 3148.79 ± 1718.24 | 1278.25 ± 1211.38 | 3858.86 ± 1820.45 | 2287.22 ± 1558.91 | 2643.28 |
| SFT-QA | 2933.34 ± 1710.83 | 1267.93 ± 1394.58 | 3152.43 ± 1819.12 | 2105.04 ± 1657.94 | 2364.69 |
| +CoT | 2870.24 ± 1623.05 | 1361.11 ± 1477.56 | 3307.65 ± 1723.66 | 2162.59 ± 1595.87 | 2425.40 |
| SFT-List | 3656.68 ± 1433.71 | 1218.50 ± 1504.40 | 4169.00 ± 1664.91 | 2378.76 ± 1851.69 | 2855.74 |
| +CoT | 3525.13 ± 1581.85 | 1257.90 ± 1481.72 | 4063.66 ± 1674.81 | 2256.44 ± 1626.20 | 2775.78 |

Table 27: Response length (mean ± standard deviation) for **QA** answer format for the experiments from Section 4.1.

| | MedQA | MedMCQA | MXQA | MLUP-H | Average |
|---|---|---|---|---|---|
| SFT-QA | 1295.77 ± 991.87 | 1082.37 ± 1028.77 | 1468.30 ± 1235.48 | 1092.76 ± 923.33 | 1234.80 |
| +CoT | 1450.64 ± 1084.56 | 1195.57 ± 1168.67 | 1719.79 ± 1428.31 | 1187.19 ± 864.02 | 1388.30 |
| SFT-MCQ | 1448.70 ± 1168.88 | 1123.19 ± 978.47 | 1823.66 ± 1542.80 | 1281.55 ± 1125.39 | 1419.28 |
| +CoT | 1442.76 ± 1233.05 | 1146.67 ± 1069.65 | 1769.96 ± 1506.86 | 1200.58 ± 1003.70 | 1389.99 |
| SFT-List | 1417.71 ± 1231.00 | 1019.12 ± 1068.58 | 1579.55 ± 1346.94 | 1133.24 ± 1090.30 | 1287.40 |
| +CoT | 1488.67 ± 1253.06 | 1073.88 ± 1056.35 | 1648.48 ± 1418.77 | 1120.35 ± 883.86 | 1332.85 |

Table 28: Response length (mean ± standard deviation) for a ranked-**list** answer format for the experiments from Section 4.1.

| | MedQA | MedMCQA | MXQA | MLUP-H | Average |
|---|---|---|---|---|---|
| SFT-List | 1365.25 ± 1074.55 | 1154.27 ± 960.43 | 1485.45 ± 1156.37 | 1120.23 ± 775.11 | 1281.30 |
| +CoT | 1527.38 ± 1109.57 | 1225.99 ± 1073.97 | 1704.15 ± 1294.16 | 1242.31 ± 968.28 | 1424.96 |
| SFT-MCQ | 2497.83 ± 3257.88 | 1948.54 ± 3239.73 | 2984.07 ± 3433.67 | 2320.56 ± 3310.11 | 2437.75 |
| +CoT | 1801.25 ± 1373.75 | 1233.82 ± 935.18 | 2192.48 ± 1561.14 | 1456.62 ± 1238.29 | 1671.04 |
| SFT-QA | 1634.33 ± 2060.58 | 1835.73 ± 2674.40 | 1798.66 ± 2182.52 | 1761.27 ± 2501.29 | 1757.50 |
| +CoT | 1629.20 ± 1134.62 | 1258.18 ± 1244.22 | 1913.98 ± 1349.85 | 1325.79 ± 960.62 | 1531.79 |

Table 29: Metrics related to the ranked **list** answer format from the generated evaluation responses across benchmarks from Section 4.1.

| | MedQA | | | MedMCQA | | | MXQA | | | MLUP-H | | |
|---|---|---|---|---|---|---|---|---|---|---|---|---|
| | LL | VLL | CP | LL | VLL | CP | LL | VLL | CP | LL | VLL | CP |
| SFT-List | 2.72 | 2.74 | 1.29 | 2.11 | 2.13 | 1.46 | 2.82 | 2.84 | 1.60 | 2.37 | 2.37 | 1.28 |
| +CoT | 2.78 | 2.80 | 1.28 | 2.10 | 2.13 | 1.44 | 2.85 | 2.88 | 1.63 | 2.47 | 2.49 | 1.33 |
| SFT-MCQ | 144.84 | 146.14 | 2.65 | 135.98 | 136.04 | 2.82 | 151.28 | 151.79 | 3.41 | 132.36 | 132.90 | 2.43 |
| +CoT | 1.51 | 1.54 | 1.07 | 1.29 | 1.31 | 1.07 | 1.46 | 1.50 | 1.16 | 1.47 | 1.50 | 1.07 |
| SFT-QA | 8.42 | 8.75 | 1.64 | 15.52 | 15.70 | 1.96 | 13.03 | 13.66 | 2.01 | 17.01 | 17.27 | 1.65 |
| +CoT | 1.17 | 2.00 | 1.09 | 0.97 | 1.51 | 1.13 | 0.99 | 1.93 | 1.28 | 1.03 | 1.67 | 1.08 |

Table 30: Results table for RFT experiments evaluated on the benchmarks with MCQ answer format from Section 4.2.

| | Dataset | Prior Prompt | Reward Function | Prompting | MedQA | MedMCQA | MXQA | MLUP-H | Average |
|---|---|---|---|---|---|---|---|---|---|
| RFT-MCQ (Baseline, No Prompt) | MCQ | No Prompt | MCQ-Acc | MCQ-CoT | 66.30 | 16.99 | 14.20 | 58.31 | 38.95 |
| RFT-MCQ (Baseline, MCQ) | MCQ | MCQ | MCQ-Acc | MCQ | 67.48 | 18.73 | 14.33 | 58.68 | 39.80 |
| RFT-MCQ (MCQ-CoT) | MCQ | MCQ-CoT | MCQ-Acc | MCQ-CoT | 66.93 | 17.45 | 14.16 | 58.80 | 39.34 |
| RFT-QA (Baseline, No Prompt) | QA | No Prompt | QA-Acc | MCQ-CoT | 47.05 | 10.54 | 12.49 | 40.95 | 27.76 |
| RFT-QA (Baseline, QA) | QA | QA | QA-Acc | MCQ | 51.06 | 14.93 | 10.73 | 36.19 | 28.23 |
| RFT-List (Baseline, List, Acc) | QA | List | List-Acc | MCQ | 16.03 | 6.21 | 8.12 | 21.52 | 12.97 |
| RFT-List (List-CoT, Acc) | QA | List-CoT | List-Acc | MCQ-CoT | 36.53 | 7.97 | 9.55 | 35.57 | 22.40 |
| RFT-List (List, MRR) | QA | List | List-MRR | MCQ | 12.18 | 5.79 | 7.18 | 16.50 | 10.41 |
| RFT-List (List-CoT, MRR) | QA | List-CoT | List-MRR | MCQ-CoT | 23.96 | 10.65 | 7.51 | 30.81 | 18.23 |
| RFT-List (List-CoT, Judge-MRR) | QA | List-CoT | LLM-List-MRR | MCQ-CoT | 30.09 | 11.37 | 5.55 | 34.96 | 20.49 |
| RFT-MCQ (MCQ-CoT, 4 Epochs) | MCQ | MCQ-CoT | MCQ-Acc | MCQ-CoT | 68.34 | 18.24 | 14.37 | 58.92 | 39.97 |
| RFT-MCQ (MCQ-CoT, No format) | MCQ | No Prompt | MCQ-Acc-NF | MCQ-CoT | 67.56 | 18.18 | 13.96 | 58.56 | 39.57 |
| RFT-List (List-CoT, Judge-MRR-Simple) | QA | List-CoT | LLM-List-MRR-Simple | MCQ-CoT | 24.27 | 12.15 | 7.76 | 31.91 | 19.02 |
| RFT-List (List-CoT, Judge-MRR-Gemini) | QA | List-CoT | LLM-List-MRR-Gemini | MCQ-CoT | 54.12 | 18.46 | 9.76 | 50.12 | 33.12 |
| RFT-MCQ (Acc) + QA (Acc) | MCQ,QA | MCQ-CoT,QA-CoT | MCQ-Acc,QA-Acc | MCQ-CoT | 28.04 | 0.00 | 9.88 | 13.33 | 12.81 |
| RFT-MCQ (Acc) + List (Acc) | MCQ,QA | MCQ-CoT,List-CoT | MCQ-Acc,List-Acc | MCQ-CoT | 68.03 | 17.84 | 14.37 | 58.56 | 39.70 |
| RFT-MCQ (Acc) + List (MRR) | MCQ,QA | MCQ-CoT,List-CoT | MCQ-Acc,List-MRR | MCQ-CoT | 68.74 | 17.93 | 13.59 | 59.78 | 40.01 |
| Qwen2.5 3B: RFT-MCQ | MCQ | MCQ-CoT | MCQ-Acc | MCQ-CoT | 54.05 | 14.96 | 12.33 | 43.77 | 31.28 |
| Qwen2.5 3B: RFT-QA | QA | QA-CoT | QA-Acc | MCQ-CoT | 52.95 | **25.09** | 11.43 | 41.08 | 32.64 |
| Qwen2.5 3B: RFT-List (Acc) | QA | List-CoT | List-Acc | MCQ-CoT | 52.24 | 24.67 | 10.78 | 40.95 | 32.16 |
| Qwen2.5 3B: RFT-List (MRR) | QA | List-CoT | List-MRR | MCQ-CoT | 50.67 | 24.70 | 11.59 | 41.08 | 32.01 |
| Qwen3 4B: RFT-MCQ | MCQ | MCQ-CoT | MCQ-Acc | MCQ-CoT | 75.88 | 18.81 | 18.12 | **68.09** | **45.22** |
| Qwen3 4B: RFT-QA | QA | QA-CoT | QA-Acc | MCQ-CoT | 75.88 | 19.80 | 17.59 | 67.36 | 45.16 |
| Qwen3 4B: RFT-List (Acc) | QA | List-CoT | List-Acc | MCQ-CoT | 73.76 | 20.07 | 15.67 | 65.40 | 43.73 |
| Qwen3 4B: RFT-List (MRR) | QA | List-CoT | List-MRR | MCQ | 75.18 | 19.37 | 16.82 | 65.89 | 44.32 |
| OpenThinker3 7B: RFT-MCQ | MCQ | MCQ-CoT | MCQ-Acc | MCQ-CoT | 56.79 | 17.11 | 13.02 | 48.04 | 33.74 |
| OpenThinker3 7B: RFT-QA | QA | QA-CoT | QA-Acc | MCQ-CoT | 56.64 | 21.66 | 11.51 | 46.21 | 34.01 |
| OpenThinker3 7B: RFT-List (Acc) | QA | List-CoT | List-Acc | MCQ-CoT | 56.25 | 22.57 | 11.02 | 47.07 | 34.23 |
| OpenThinker3 7B: RFT-List (MRR) | QA | List-CoT | List-MRR | MCQ-CoT | 57.50 | 21.76 | 11.35 | 47.80 | 34.60 |
| m1 7B 23K: RFT-MCQ | MCQ | MCQ-CoT | MCQ-Acc | MCQ-CoT | **76.28** | 19.95 | **19.63** | 64.06 | 44.98 |
| m1 7B 23K: RFT-QA | QA | QA-CoT | QA-Acc | MCQ-CoT | 72.51 | 21.79 | 16.33 | 61.61 | 43.06 |
| m1 7B 23K: RFT-List (Acc) | QA | List-CoT | List-Acc | MCQ-CoT | 73.06 | 21.69 | 17.96 | 63.45 | 44.04 |
| m1 7B 23K: RFT-List (MRR) | QA | List-CoT | List-MRR | MCQ-CoT | 75.88 | 20.60 | 16.90 | 60.76 | 43.54 |
| AlphaMed 7B: RFT-MCQ | MCQ | MCQ-CoT | MCQ-Acc | MCQ-CoT | 71.88 | 17.58 | 17.06 | 61.61 | 42.03 |
| AlphaMed 7B: RFT-QA | QA | QA-CoT | QA-Acc | MCQ-CoT | 71.25 | 17.06 | 15.71 | 62.96 | 41.74 |
| AlphaMed 7B: RFT-List (Acc) | QA | List-CoT | List-Acc | MCQ-CoT | 68.19 | 15.50 | 12.61 | 57.09 | 38.35 |
| AlphaMed 7B: RFT-List (MRR) | QA | List-CoT | List-MRR | MCQ-CoT | 60.57 | 14.75 | 14.20 | 56.23 | 36.44 |
| RFT-List (List-CoT, MRR, LP=0) | QA | List-CoT | List-MRR-0 | MCQ-CoT | 17.12 | 8.11 | 5.10 | 21.39 | 12.93 |
| RFT-List (List-CoT, MRR, LP=0.1) | QA | List-CoT | List-MRR-0.1 | MCQ-CoT | 65.04 | 20.00 | 12.94 | 52.93 | 37.73 |
| RFT-List (List-CoT, MRR, LP=0.3) | QA | List-CoT | List-MRR-0.3 | MCQ-CoT | 55.22 | 18.07 | 10.53 | 50.24 | 33.52 |
| RFT-List (List-CoT, MRR, LP=0.5) | QA | List-CoT | List-MRR-0.5 | MCQ-CoT | 63.00 | 20.70 | 12.65 | 54.77 | 37.78 |
| RFT-List (List-CoT, MRR, LP=0.7) | QA | List-CoT | List-MRR-0.7 | MCQ-CoT | 56.48 | 18.78 | 12.24 | 51.34 | 34.71 |
| RFT-List (List-CoT, MRR, LP=0.9) | QA | List-CoT | List-MRR-0.9 | MCQ-CoT | 60.88 | 19.01 | 11.51 | 50.49 | 35.47 |
| AlphaMed 7B: RFT-List (Acc, LP=0.3) | QA | List-CoT | List-Acc-LP-0.3 | MCQ-CoT | 23.88 | 11.72 | 7.67 | 44.01 | 21.82 |
| m1 7B 23K: RFT-List (Acc, LP=0.3) | QA | List-CoT | List-Acc-LP-0.3 | MCQ-CoT | 74.78 | 21.50 | 17.27 | 61.25 | 43.70 |
| OpenThinker3 7B: RFT-List (Acc, LP=0.3) | QA | List-CoT | List-Acc-LP-0.3 | MCQ-CoT | 56.64 | 21.76 | 11.02 | 48.04 | 34.36 |
| Qwen2.5 3B: RFT-List (Acc, LP=0.3) | QA | List-CoT | List-Acc-LP-0.3 | MCQ-CoT | 49.57 | 21.38 | 11.88 | 41.81 | 31.16 |

Table 31: Results table for RFT experiments evaluated on the benchmarks with QA answer format from Section 4.2.

| | Dataset | Prior Prompt | Reward Function | Prompting | MedQA | | MedMCQA | | MXQA | | MLUP-H | | Average | |
|---|---|---|---|---|---|---|---|---|---|---|---|---|---|---|
| | | | | | $Acc_{QA}$ | $Acc_{QA}^{LLM}$ | $Acc_{QA}$ | $Acc_{QA}^{LLM}$ | $Acc_{QA}$ | $Acc_{QA}^{LLM}$ | $Acc_{QA}$ | $Acc_{QA}^{LLM}$ | $Acc_{QA}$ | $Acc_{QA}^{LLM}$ |
| RFT-QA (Baseline, No Prompt) | QA | No Prompt | QA-Acc | QA-CoT | 5.84 | 28.71 | 3.26 | 31.74 | 1.53 | 24.02 | 5.71 | 32.20 | 4.08 | 29.17 |
| RFT-QA (Baseline, QA) | QA | QA | QA-Acc | QA | 0.65 | 20.60 | 1.38 | 31.93 | 0.29 | 19.42 | 1.49 | 27.17 | 0.95 | 24.78 |
| RFT-QA (Baseline, QA) | QA | QA | QA-Acc | QA-CoT | 1.05 | 21.74 | 1.28 | 30.87 | 0.14 | 16.78 | 2.31 | 26.49 | 1.20 | 23.97 |
| RFT-QA (QA-CoT) | QA | QA-CoT | QA-Acc | QA-CoT | 1.30 | 22.38 | 1.33 | 31.51 | 0.29 | 19.13 | 1.22 | 27.85 | 1.03 | 25.22 |
| RFT-MCQ (Baseline, No Prompt) | MCQ | No Prompt | MCQ-Acc | QA-CoT | 14.27 | 51.50 | 7.39 | 44.40 | 5.08 | 38.59 | 12.91 | 50.00 | 9.91 | 46.12 |
| RFT-MCQ (Baseline, MCQ) | MCQ | MCQ | MCQ-Acc | QA | 12.98 | 47.69 | 6.93 | 44.27 | 3.74 | 36.53 | 13.18 | 49.86 | 9.21 | 44.59 |
| RFT-MCQ (Baseline, MCQ) | MCQ | MCQ | MCQ-Acc | QA-CoT | 13.38 | 47.20 | 7.48 | 44.77 | 4.27 | 37.49 | 11.82 | 47.01 | 9.24 | 44.12 |
| RFT-MCQ (MCQ-CoT) | MCQ | MCQ-CoT | MCQ-Acc | QA-CoT | 14.76 | 50.36 | 7.20 | 45.32 | 4.41 | 37.06 | 13.45 | 52.58 | 9.96 | 46.33 |
| RFT-List (Baseline, List, Acc) | QA | List | List-Acc | QA | 13.54 | 44.93 | 6.79 | 43.67 | 4.36 | 37.63 | 13.04 | 51.63 | 9.43 | 44.46 |
| RFT-List (Baseline, List, Acc) | QA | List | List-Acc | QA-CoT | 12.41 | 41.12 | 5.60 | 36.33 | 3.79 | 33.41 | 10.73 | 44.78 | 8.11 | 38.21 |
| RFT-List (List-CoT, Acc) | QA | List-CoT | List-Acc | QA-CoT | 6.24 | 20.03 | 1.97 | 17.48 | 1.58 | 14.48 | 7.34 | 24.05 | 4.28 | 19.01 |
| RFT-List (List, MRR) | QA | List | List-MRR | QA | 15.49 | 50.69 | 7.48 | 44.13 | 4.65 | 38.11 | 13.86 | 51.90 | 10.37 | 46.21 |
| RFT-List (List, MRR) | QA | List | List-MRR | QA-CoT | 17.52 | 50.53 | 7.89 | 44.27 | 4.51 | 37.15 | 13.45 | 51.49 | 10.84 | 45.86 |
| RFT-List (List-CoT, MRR) | QA | List-CoT | List-MRR | QA-CoT | 5.68 | 20.60 | 3.21 | 22.75 | 1.92 | 16.68 | 6.93 | 27.58 | 4.44 | 21.90 |
| RFT-List (List-CoT, Judge-MRR) | QA | List-CoT | LLM-List-MRR | QA-CoT | 9.73 | 30.49 | 4.77 | 30.28 | 2.78 | 25.60 | 8.70 | 35.05 | 6.50 | 30.36 |
| RFT-MCQ (MCQ-CoT, 4 Epochs) | MCQ | MCQ-CoT | MCQ-Acc | QA-CoT | 15.09 | 48.66 | 7.29 | 43.94 | 4.65 | 37.68 | 12.09 | 50.00 | 9.78 | 45.07 |
| RFT-MCQ (MCQ-CoT, No format) | MCQ | No Prompt | MCQ-Acc-NF | QA-CoT | 13.87 | 48.62 | 7.34 | 44.91 | 4.36 | 39.12 | 12.36 | 52.17 | 9.48 | 46.24 |
| RFT-List (List-CoT, Judge-MRR-Simple) | QA | List-CoT | LLM-List-MRR-Simple | QA-CoT | 4.70 | 26.85 | 4.13 | 28.35 | 1.87 | 21.96 | 6.79 | 31.93 | 4.37 | 27.27 |
| RFT-List (List-CoT, Judge-MRR-Gemini) | QA | List-CoT | LLM-List-MRR-Gemini | QA-CoT | 14.19 | 46.96 | 7.39 | 42.20 | 4.22 | 34.71 | 11.82 | 48.78 | 9.40 | 43.16 |
| RFT-MCQ (Acc) + QA (Acc) | MCQ,QA | MCQ-CoT,QA-CoT | MCQ-Acc,QA-Acc | QA-CoT | 4.70 | 29.85 | 3.30 | 37.61 | 0.96 | 23.54 | 4.62 | 36.82 | 3.40 | 31.96 |
| RFT-MCQ (Acc) + List (Acc) | MCQ,QA | MCQ-CoT,List-CoT | MCQ-Acc,List-Acc | QA-CoT | 16.79 | 52.47 | 8.53 | 44.04 | 5.80 | 40.36 | 13.86 | 50.68 | 11.24 | 46.89 |
| RFT-MCQ (Acc) + List (MRR) | MCQ,QA | MCQ-CoT,List-CoT | MCQ-Acc,List-MRR | QA-CoT | 18.57 | 52.96 | 7.94 | 45.37 | 5.56 | 39.36 | 15.22 | 53.94 | 11.82 | 47.91 |
| Qwen2.5 3B: RFT-MCQ | MCQ | MCQ-CoT | MCQ-Acc | QA-CoT | 8.92 | 35.36 | 5.38 | 38.81 | 3.31 | 33.17 | 9.24 | 38.45 | 6.82 | 36.45 |
| Qwen2.5 3B: RFT-QA | QA | QA-CoT | QA-Acc | QA-CoT | 1.46 | 25.22 | 0.87 | 33.39 | 0.10 | 23.11 | 0.95 | 28.67 | 0.84 | 27.60 |
| Qwen2.5 3B: RFT-List (Acc) | QA | List-CoT | List-Acc | QA-CoT | 11.52 | 40.96 | 5.73 | 40.32 | 2.78 | 35.81 | 8.97 | 40.08 | 7.25 | 39.29 |
| Qwen2.5 3B: RFT-List (MRR) | QA | List-CoT | List-MRR | QA-CoT | 11.11 | 40.71 | 6.38 | 41.19 | 3.21 | 36.53 | 11.01 | 42.39 | 7.93 | 40.20 |
| Qwen3 4B: RFT-MCQ | MCQ | MCQ-CoT | MCQ-Acc | QA-CoT | 17.11 | 51.09 | 8.53 | 45.05 | 5.18 | 39.55 | 13.18 | 48.51 | 11.00 | 46.05 |
| Qwen3 4B: RFT-QA | QA | QA-CoT | QA-Acc | QA-CoT | 12.90 | 48.99 | 7.48 | 44.91 | 3.69 | 35.04 | 10.19 | 50.95 | 8.57 | 44.97 |
| Qwen3 4B: RFT-List (Acc) | QA | List-CoT | List-Acc | QA-CoT | 18.57 | 53.28 | 9.04 | 46.15 | 5.70 | 40.70 | 13.45 | 53.67 | 11.69 | 48.45 |
| Qwen3 4B: RFT-List (MRR) | QA | List-CoT | List-MRR | QA | 18.41 | 53.45 | 9.08 | 45.50 | 6.09 | 40.46 | 14.13 | 54.76 | 11.93 | 48.54 |
| Qwen3 4B: RFT-List (MRR) | QA | List-CoT | List-MRR | QA-CoT | 19.71 | 54.18 | 9.86 | 45.37 | 6.42 | 42.09 | 14.40 | 52.17 | 12.60 | 48.45 |
| OpenThinker3 7B: RFT-MCQ | MCQ | MCQ-CoT | MCQ-Acc | QA-CoT | 6.49 | 43.71 | 4.68 | 37.66 | 2.25 | 36.91 | 7.34 | 45.11 | 5.19 | 40.85 |
| OpenThinker3 7B: RFT-QA | QA | QA-CoT | QA-Acc | QA-CoT | 6.33 | 41.85 | 4.50 | 42.20 | 1.87 | 32.69 | 5.98 | 42.39 | 4.67 | 39.78 |
| OpenThinker3 7B: RFT-List (Acc) | QA | List-CoT | List-Acc | QA-CoT | 7.54 | 45.58 | 5.05 | 38.76 | 2.11 | 38.69 | 8.29 | 42.66 | 5.75 | 41.42 |
| OpenThinker3 7B: RFT-List (MRR) | QA | List-CoT | List-MRR | QA-CoT | 7.14 | 44.12 | 5.23 | 39.13 | 2.83 | 36.34 | 7.07 | 43.48 | 5.57 | 40.77 |
| m1 7B 23K: RFT-MCQ | MCQ | MCQ-CoT | MCQ-Acc | QA-CoT | 12.65 | 48.18 | 6.42 | 38.58 | 3.34 | 33.60 | 11.14 | 46.33 | 8.49 | 41.67 |
| m1 7B 23K: RFT-QA | QA | QA-CoT | QA-Acc | QA-CoT | 10.87 | 47.85 | 5.37 | 45.78 | 3.12 | 36.91 | 8.15 | 51.63 | 6.88 | 45.54 |
| m1 7B 23K: RFT-List (Acc) | QA | List-CoT | List-Acc | QA-CoT | 13.06 | 50.28 | 5.94 | 45.21 | 3.79 | 35.38 | 11.28 | 49.18 | 8.96 | 43.92 |
| m1 7B 23K: RFT-List (MRR) | QA | List-CoT | List-MRR | QA-CoT | 14.68 | 52.15 | 6.93 | 40.87 | 4.46 | 36.91 | 13.32 | 51.49 | 9.85 | 45.36 |
| AlphaMed 7B: RFT-MCQ | MCQ | MCQ-CoT | MCQ-Acc | QA-CoT | 5.68 | 18.09 | 2.75 | 19.50 | 0.96 | 14.00 | 4.62 | 22.96 | 3.50 | 18.64 |
| AlphaMed 7B: RFT-QA | QA | QA-CoT | QA-Acc | QA-CoT | 6.24 | 38.61 | 3.26 | 40.55 | 1.49 | 31.59 | 4.76 | 42.66 | 3.94 | 38.35 |
| AlphaMed 7B: RFT-List (Acc) | QA | List-CoT | List-Acc | QA-CoT | 9.08 | 25.39 | 5.14 | 28.85 | 2.25 | 18.22 | 6.79 | 29.08 | 5.82 | 25.38 |
| AlphaMed 7B: RFT-List (MRR) | QA | List-CoT | List-MRR | QA-CoT | 6.47 | 34.13 | 3.37 | 31.99 | 1.87 | 19.99 | 8.32 | 37.23 | 7.16 | 30.38 |
| RFT-List (List-CoT, MRR, LP=0) | QA | List-CoT | List-MRR-0 | QA-CoT | 12.08 | 36.98 | 5.55 | 32.75 | 3.74 | 27.71 | 11.55 | 40.76 | 8.23 | 34.55 |
| RFT-List (List-CoT, MRR, LP=0.1) | QA | List-CoT | List-MRR-0.1 | QA-CoT | 18.10 | 51.26 | 8.53 | 44.91 | 5.66 | 38.54 | 15.22 | 54.21 | 11.85 | 47.23 |
| RFT-List (List-CoT, MRR, LP=0.3) | QA | List-CoT | List-MRR-0.3 | QA-CoT | 18.57 | 54.42 | 7.57 | 45.50 | 5.23 | 39.74 | 14.81 | 53.53 | 11.54 | 48.30 |
| RFT-List (List-CoT, MRR, LP=0.5) | QA | List-CoT | List-MRR-0.5 | QA-CoT | 17.60 | 50.93 | 8.03 | 45.60 | 5.18 | 39.55 | 14.13 | 48.91 | 11.24 | 46.25 |
| RFT-List (List-CoT, MRR, LP=0.7) | QA | List-CoT | List-MRR-0.7 | QA-CoT | 18.09 | 52.23 | 7.89 | 43.26 | 5.18 | 40.65 | 15.08 | 53.26 | 11.56 | 47.35 |
| RFT-List (List-CoT, MRR, LP=0.9) | QA | List-CoT | List-MRR-0.9 | QA-CoT | 17.53 | 49.47 | 7.20 | 44.31 | 4.99 | 40.27 | 13.45 | 50.27 | 10.34 | 46.08 |
| AlphaMed 7B: RFT-List (Acc, LP=0.3) | QA | List-CoT | List-Acc-LP-0.3 | QA-CoT | 15.82 | 45.17 | 7.02 | 36.83 | 3.69 | 29.19 | 12.09 | 43.21 | 9.66 | 38.60 |
| m1 7B 23K: RFT-List (Acc, LP=0.3) | QA | List-CoT | List-Acc-LP-0.3 | QA-CoT | 12.33 | 50.61 | 6.74 | 39.68 | 3.64 | 35.52 | 11.01 | 48.23 | 8.43 | 43.51 |
| OpenThinker3 7B: RFT-List (Acc, LP=0.3) | QA | List-CoT | List-Acc-LP-0.3 | QA-CoT | 7.14 | 44.69 | 5.00 | 39.45 | 2.97 | 37.63 | 8.02 | 45.38 | 5.78 | 41.79 |
| Qwen2.5 3B: RFT-List (Acc, LP=0.3) | QA | List-CoT | List-Acc-LP-0.3 | QA-CoT | 9.89 | 32.20 | 4.77 | 29.04 | 2.59 | 27.56 | 8.02 | 31.39 | 6.32 | 30.05 |

Table 32: Results table for RFT experiments evaluated on the benchmarks with a ranked-**list** answer format from Section 4.2.

| | Dataset | Prior Prompt | Reward Function | Prompting | MedQA | | | | MedMCQA | | | | MXQA | | | | MLUP-H | | | | Average | | | |
|---|---|---|---|---|---|---|---|---|---|---|---|---|---|---|---|---|---|---|---|---|---|---|---|---|
| | | | | | $Acc_{List}$ | $Acc_{List}^{LLM}$ | $MRR_{List}$ | $MRR_{List}^{LLM}$ | $Acc_{List}$ | $Acc_{List}^{LLM}$ | $MRR_{List}$ | $MRR_{List}^{LLM}$ | $Acc_{List}$ | $Acc_{List}^{LLM}$ | $MRR_{List}$ | $MRR_{List}^{LLM}$ | $Acc_{List}$ | $Acc_{List}^{LLM}$ | $MRR_{List}$ | $MRR_{List}^{LLM}$ | $Acc_{List}$ | $Acc_{List}^{LLM}$ | $MRR_{List}$ | $MRR_{List}^{LLM}$ |
| RFT-List (Baseline, List, Acc) | QA | List | List-Acc | List | 33.98 | 74.61 | 20.05 | 44.03 | 23.26 | 62.66 | 11.91 | 30.76 | 17.26 | 56.18 | 8.32 | 27.04 | 22.15 | 74.86 | 15.05 | 46.69 | 24.16 | 67.08 | 13.83 | 37.13 |
| RFT-List (Baseline, List, Acc) | QA | List | List-Acc | List-CoT | 26.12 | 65.21 | 19.80 | 49.22 | 17.20 | 52.80 | 11.18 | 33.20 | 12.66 | 46.79 | 7.51 | 28.15 | 18.61 | 69.02 | 15.57 | 52.06 | 18.65 | 58.46 | 13.52 | 40.66 |
| RFT-List (List-CoT, Acc) | QA | List-CoT | List-Acc | List-CoT | 31.87 | 66.34 | 24.12 | 49.83 | 19.40 | 47.29 | 13.11 | 30.77 | 13.95 | 45.97 | 8.91 | 28.68 | 23.23 | 66.85 | 18.54 | 51.75 | 22.11 | 56.61 | 16.17 | 40.26 |
| RFT-List (List, MRR) | QA | List | List-MRR | List | 30.98 | 69.91 | 22.22 | 48.04 | 20.55 | 56.93 | 12.80 | 34.36 | 14.81 | 52.59 | 8.74 | 29.30 | 21.88 | 72.55 | 17.34 | 53.24 | 22.06 | 63.00 | 15.27 | 41.24 |
| RFT-List (List, MRR) | QA | List | List-MRR | List-CoT | 29.36 | 66.50 | 22.56 | 50.65 | 17.29 | 48.30 | 11.80 | 31.78 | 12.85 | 46.26 | 8.42 | 30.31 | 20.05 | 72.55 | 17.37 | 55.47 | 20.04 | 58.40 | 15.04 | 42.05 |
| RFT-List (List-CoT, MRR) | QA | List-CoT | List-MRR | List-CoT | 30.33 | 70.88 | 24.14 | 54.84 | 18.30 | 52.54 | 12.25 | 36.69 | 13.47 | 49.42 | 9.02 | 31.75 | 21.74 | 74.59 | 17.89 | 58.23 | 20.96 | 61.60 | 15.83 | 44.89 |
| RFT-List (List-CoT, Judge-MRR) | QA | List-CoT | LLM-List-MRR | List-CoT | 22.30 | 67.32 | 18.51 | 56.07 | 12.61 | 53.62 | 9.66 | 39.62 | 9.73 | 49.42 | 6.88 | 35.64 | 14.81 | 73.23 | 13.60 | 63.41 | 14.86 | 60.90 | 12.16 | 48.68 |
| RFT-MCQ (Baseline, No Prompt) | MCQ | No Prompt | MCQ-Acc | List-CoT | 15.57 | 52.72 | 13.70 | 46.49 | 7.94 | 30.60 | 7.17 | 26.58 | 5.94 | 30.49 | 4.81 | 24.30 | 11.82 | 53.94 | 11.32 | 48.58 | 10.32 | 41.94 | 9.25 | 36.49 |
| RFT-MCQ (Baseline, MCQ) | MCQ | MCQ | MCQ-Acc | List | 14.11 | 45.50 | 12.23 | 38.35 | 7.71 | 28.21 | 7.12 | 24.26 | 6.23 | 28.95 | 4.93 | 22.12 | 12.09 | 52.45 | 11.36 | 46.18 | 10.04 | 38.78 | 8.91 | 32.73 |
| RFT-MCQ (Baseline, MCQ) | MCQ | MCQ | MCQ-Acc | List-CoT | 13.95 | 41.77 | 12.99 | 38.92 | 7.16 | 23.21 | 6.71 | 21.60 | 5.98 | 25.42 | 4.56 | 22.06 | 10.60 | 44.43 | 10.33 | 42.37 | 9.42 | 33.71 | 8.65 | 30.19 |
| RFT-MCQ (MCQ-CoT) | MCQ | MCQ-CoT | MCQ-Acc | List-CoT | 13.71 | 47.85 | 11.14 | 39.15 | 7.66 | 31.42 | 6.42 | 25.77 | 5.42 | 28.91 | 4.16 | 21.91 | 12.77 | 52.04 | 10.90 | 45.19 | 9.89 | 40.06 | 8.16 | 33.00 |
| RFT-QA (Baseline, No Prompt) | QA | No Prompt | QA-Acc | List-CoT | 15.41 | 49.80 | 12.87 | 42.31 | 8.39 | 34.36 | 7.44 | 28.23 | 5.80 | 28.62 | 4.22 | 20.94 | 9.38 | 54.89 | 8.73 | 47.52 | 9.75 | 41.92 | 8.32 | 34.75 |
| RFT-QA (Baseline, QA) | QA | QA | QA-Acc | List | 16.14 | 55.47 | 12.77 | 42.88 | 10.64 | 43.03 | 8.23 | 32.14 | 6.38 | 38.45 | 4.73 | 26.83 | 12.50 | 62.77 | 10.72 | 51.56 | 11.42 | 49.93 | 9.06 | 38.35 |
| RFT-QA (Baseline, QA) | QA | QA | QA-Acc | List-CoT | 14.03 | 56.37 | 11.98 | 47.07 | 7.66 | 36.79 | 6.54 | 29.31 | 5.56 | 35.62 | 4.00 | 26.06 | 7.88 | 60.05 | 7.27 | 51.72 | 8.78 | 47.21 | 7.45 | 38.54 |
| RFT-QA (QA-CoT) | QA | QA-CoT | QA-Acc | QA-CoT | 0.81 | 3.16 | 0.63 | 2.38 | 0.50 | 2.94 | 0.35 | 2.24 | 0.34 | 2.83 | 0.14 | 1.92 | 0.82 | 5.43 | 0.82 | 4.74 | 0.62 | 3.59 | 0.48 | 2.82 |
| RFT-MCQ (MCQ-CoT, 4 Epochs) | MCQ | MCQ-CoT | MCQ-Acc | List-CoT | 15.82 | 45.09 | 14.90 | 41.72 | 7.61 | 26.15 | 7.12 | 23.70 | 4.51 | 22.63 | 4.15 | 19.49 | 11.68 | 47.96 | 11.22 | 43.88 | 9.90 | 35.46 | 9.35 | 32.20 |
| RFT-MCQ (MCQ-CoT, No format) | MCQ | No Prompt | MCQ-Acc-NF | List-CoT | 15.33 | 46.23 | 14.06 | 41.85 | 7.43 | 28.26 | 6.85 | 24.66 | 5.32 | 26.46 | 4.56 | 21.35 | 11.96 | 50.68 | 11.29 | 46.00 | 10.01 | 37.91 | 9.19 | 33.47 |
| RFT-List (List-CoT, Judge-MRR-Simple) | QA | List-CoT | LLM-List-MRR-Simple | List-CoT | 7.06 | 30.74 | 5.76 | 26.90 | 4.27 | 34.71 | 3.32 | 23.74 | 2.78 | 23.63 | 1.85 | 16.92 | 4.89 | 41.03 | 4.53 | 37.21 | 4.75 | 31.84 | 3.86 | 26.19 |
| RFT-List (List-CoT, Judge-MRR-Gemini) | QA | List-CoT | LLM-List-MRR-Gemini | List-CoT | 19.71 | 65.13 | 16.71 | 55.63 | 11.51 | 52.57 | 9.09 | 39.23 | 8.29 | 47.79 | 6.12 | 34.58 | 12.77 | 71.88 | 11.87 | 62.57 | 13.07 | 59.34 | 10.95 | 48.00 |
| RFT-MCQ (Acc) + QA (Acc) | MCQ,QA | MCQ-CoT,QA-CoT | MCQ-Acc,QA-Acc | List-CoT | 9.41 | 48.99 | 8.04 | 41.56 | 5.92 | 33.07 | 5.18 | 27.50 | 3.16 | 28.19 | 2.48 | 21.28 | 7.07 | 54.48 | 6.52 | 46.60 | 6.39 | 41.18 | 5.56 | 34.24 |
| RFT-MCQ (Acc) + List (Acc) | MCQ,QA | MCQ-CoT,List-CoT | MCQ-Acc,List-Acc | List-CoT | 30.82 | 68.37 | 22.60 | 50.18 | 18.90 | 52.94 | 12.19 | 36.12 | 11.92 | 54.16 | 8.94 | 31.29 | 21.20 | 73.10 | 16.68 | 54.32 | 21.47 | 61.52 | 15.10 | 43.80 |
| RFT-MCQ (Acc) + List (MRR) | MCQ,QA | MCQ-CoT,List-CoT | MCQ-Acc,List-MRR | List-CoT | 27.58 | 66.42 | 22.03 | 53.14 | 16.01 | 47.91 | 11.85 | 33.86 | 12.27 | 45.73 | 8.38 | 31.36 | 20.11 | 69.27 | 17.29 | 56.85 | 18.99 | 57.34 | 14.89 | 43.80 |
| Qwen2.5 3B: RFT-MCQ | MCQ | MCQ-CoT | MCQ-Acc | List-CoT | 11.44 | 38.04 | 10.48 | 33.95 | 6.56 | 24.17 | 5.98 | 21.14 | 3.60 | 22.44 | 3.06 | 18.62 | 9.38 | 44.16 | 8.33 | 38.31 | 7.75 | 32.20 | 6.96 | 28.01 |
| Qwen2.5 3B: RFT-QA | QA | QA-CoT | QA-Acc | List-CoT | 14.44 | 52.02 | 10.82 | 40.00 | 9.37 | 37.16 | 6.31 | 26.88 | 5.36 | 35.00 | 3.60 | 22.44 | 9.39 | 55.48 | 6.92 | 44.78 | 7.34 | 44.78 | 5.36 | 30.48 |
| Qwen2.5 3B: RFT-List (Acc) | QA | List-CoT | List-Acc | List-CoT | 29.20 | 67.40 | 17.43 | 41.53 | 19.86 | 55.21 | 10.36 | 27.79 | 13.61 | 46.88 | 7.07 | 23.84 | 19.97 | 67.80 | 13.97 | 45.54 | 20.66 | 59.82 | 12.08 | 34.68 |
| Qwen2.5 3B: RFT-List (MRR) | QA | List-CoT | List-MRR | List-CoT | 28.95 | 66.10 | 17.75 | 41.56 | 19.17 | 52.89 | 10.36 | 28.15 | 13.71 | 46.88 | 7.07 | 24.28 | 20.24 | 67.80 | 14.93 | 45.64 | 20.52 | 58.42 | 12.53 | 34.91 |
| Qwen3 4B: RFT-MCQ | MCQ | MCQ-CoT | MCQ-Acc | List-CoT | 20.52 | 63.91 | 18.67 | 56.57 | 11.38 | 41.19 | 9.53 | 33.19 | 7.91 | 45.45 | 6.28 | 35.00 | 12.50 | 66.30 | 11.76 | 58.72 | 12.34 | 55.46 | 10.53 | 46.57 |
| Qwen3 4B: RFT-QA | QA | QA-CoT | QA-Acc | List-CoT | 18.65 | 66.67 | 16.05 | 58.09 | 12.61 | 53.62 | 9.95 | 34.38 | 7.48 | 46.76 | 5.35 | 35.09 | 12.50 | 66.30 | 11.76 | 58.72 | 12.34 | 55.46 | 10.53 | 46.57 |
| Qwen3 4B: RFT-List (Acc) | QA | List-CoT | List-Acc | List-CoT | 38.44 | 79.08 | 23.60 | 49.25 | 26.56 | 66.24 | 13.46 | 32.70 | 20.71 | 61.89 | 9.58 | 29.06 | 25.27 | 79.21 | 17.10 | 58.00 | 27.74 | 71.60 | 15.94 | 40.60 |
| Qwen3 4B: RFT-List (MRR) | QA | List-CoT | List-MRR | List-CoT | 31.95 | 69.51 | 24.94 | 53.10 | 19.63 | 50.96 | 13.19 | 32.93 | 14.86 | 50.67 | 9.68 | 32.44 | 22.69 | 70.38 | 18.40 | 55.16 | 22.28 | 60.38 | 16.55 | 43.41 |
| OpenThinker3 7B: RFT-MCQ | MCQ | MCQ-CoT | MCQ-Acc | List-CoT | 9.98 | 49.51 | 8.44 | 42.87 | 5.51 | 21.02 | 3.07 | 23.97 | 2.50 | 19.20 | 1.78 | 27.80 | 6.51 | 41.92 | 5.30 | 41.49 | 6.88 | 33.28 | 6.27 | 29.11 |
| OpenThinker3 7B: RFT-QA | QA | QA-CoT | QA-Acc | List-CoT | 9.90 | 46.23 | 5.20 | 39.29 | 7.36 | 52.57 | 9.63 | 26.62 | 4.67 | 46.77 | 5.52 | 21.71 | 20.24 | 66.30 | 13.37 | 41.51 | 20.05 | 56.98 | 11.03 | 31.93 |
| OpenThinker3 7B: RFT-List (Acc) | QA | List-CoT | List-Acc | List-CoT | 27.17 | 62.29 | 19.60 | 57.87 | 17.36 | 52.57 | 9.63 | 26.62 | 13.42 | 46.77 | 5.52 | 21.71 | 20.24 | 66.30 | 13.37 | 41.51 | 20.05 | 56.98 | 11.03 | 31.93 |
| OpenThinker3 7B: RFT-List (MRR) | QA | List-CoT | List-MRR | List-CoT | 22.79 | 63.18 | 14.68 | 40.33 | 18.07 | 53.76 | 10.00 | 28.14 | 11.27 | 47.60 | 5.61 | 23.81 | 20.79 | 73.23 | 14.72 | 47.84 | 18.23 | 59.44 | 11.25 | 35.03 |
| m1 7B 23K: RFT-MCQ | MCQ | MCQ-CoT | MCQ-Acc | List-CoT | 18.25 | 57.91 | 16.68 | 51.42 | 9.58 | 30.61 | 6.76 | 34.13 | 5.54 | 33.13 | 4.77 | 13.86 | 13.86 | 57.88 | 12.98 | 52.74 | 12.40 | 46.24 | 11.20 | 40.49 |
| m1 7B 23K: RFT-QA | QA | QA-CoT | QA-Acc | List-CoT | 14.68 | 57.10 | 13.08 | 51.24 | 9.13 | 34.22 | 8.22 | 29.74 | 5.47 | 32.89 | 4.55 | 27.07 | 8.83 | 57.20 | 8.22 | 52.13 | 9.53 | 45.35 | 8.52 | 40.04 |
| m1 7B 23K: RFT-List (Acc) | QA | List-CoT | List-Acc | List-CoT | 32.12 | 75.10 | 23.64 | 57.05 | 18.05 | 56.46 | 12.46 | 36.75 | 13.85 | 54.37 | 9.58 | 30.21 | 23.23 | 75.92 | 17.54 | 52.23 | 21.81 | 65.46 | 15.80 | 44.06 |
| m1 7B 23K: RFT-List (MRR) | QA | List-CoT | List-MRR | List-CoT | 26.85 | 69.69 | 22.23 | 53.82 | 15.50 | 42.88 | 13.50 | 35.50 | 9.58 | 43.07 | 7.53 | 32.08 | 18.34 | 69.57 | 16.45 | 57.17 | 17.57 | 56.30 | 14.93 | 44.64 |
| AlphaMed 7B: RFT-MCQ | MCQ | MCQ-CoT | MCQ-Acc | List-CoT | 2.11 | 3.89 | 2.11 | 3.85 | 1.40 | 3.35 | 1.40 | 3.07 | 0.38 | 1.20 | 0.36 | 1.15 | 2.17 | 5.84 | 2.17 | 5.71 | 1.52 | 3.57 | 1.51 | 3.44 |
| AlphaMed 7B: RFT-QA | QA | QA-CoT | QA-Acc | List-CoT | 16.71 | 46.23 | 15.67 | 43.25 | 7.20 | 27.34 | 6.95 | 25.18 | 5.23 | 26.89 | 4.69 | 23.61 | 11.41 | 51.09 | 11.21 | 47.75 | 10.14 | 37.89 | 9.63 | 34.95 |
| AlphaMed 7B: RFT-List (Acc) | QA | List-CoT | List-Acc | List-CoT | 27.44 | 63.99 | 23.47 | 54.00 | 15.37 | 45.55 | 13.50 | 40.83 | 11.06 | 31.92 | 11.46 | 27.13 | 19.97 | 69.16 | 17.14 | 58.95 | 18.47 | 55.22 | 15.05 | 43.91 |
| AlphaMed 7B: RFT-List (MRR) | QA | List-CoT | List-MRR | List-CoT | 26.85 | 69.69 | 22.23 | 53.82 | 15.50 | 42.88 | 13.50 | 35.50 | 9.58 | 43.07 | 7.53 | 32.08 | 18.34 | 69.57 | 16.45 | 57.17 | 17.57 | 56.30 | 14.93 | 44.64 |
| RFT-List (List-CoT, MRR, LP=0) | QA | List-CoT | List-MRR-0 | List-CoT | 29.12 | 65.61 | 23.71 | 52.71 | 17.25 | 45.14 | 12.82 | 32.25 | 12.80 | 42.76 | 8.90 | 29.28 | 20.38 | 66.30 | 17.70 | 52.13 | 19.89 | 54.95 | 15.78 | 41.59 |
| RFT-List (List-CoT, MRR, LP=0.1) | QA | List-CoT | List-MRR-0.1 | List-CoT | 20.52 | 44.28 | 19.87 | 42.00 | 9.91 | 27.43 | 9.53 | 25.74 | 7.13 | 24.02 | 7.37 | 22.22 | 17.39 | 48.37 | 17.12 | 46.88 | 13.74 | 36.02 | 13.40 | 33.40 |
| RFT-List (List-CoT, MRR, LP=0.3) | QA | List-CoT | List-MRR-0.3 | List-CoT | 18.90 | 36.50 | 18.40 | 35.35 | 9.40 | 22.89 | 9.03 | 22.13 | 6.38 | 16.76 | 6.26 | 15.58 | 15.90 | 38.86 | 15.59 | 37.77 | 12.64 | 28.64 | 12.34 | 27.71 |
| RFT-List (List-CoT, MRR, LP=0.5) | QA | List-CoT | List-MRR-0.5 | List-CoT | 16.63 | 33.41 | 16.63 | 33.37 | 8.85 | 21.71 | 8.85 | 21.71 | 6.47 | 15.58 | 6.47 | 15.58 | 15.49 | 33.38 | 15.49 | 33.38 | 11.86 | 26.02 | 11.86 | 26.01 |
| RFT-List (List-CoT, MRR, LP=0.7) | QA | List-CoT | List-MRR-0.7 | List-CoT | 17.27 | 34.14 | 17.27 | 34.10 | 8.53 | 22.57 | 8.53 | 22.57 | 8.67 | 22.92 | 8.67 | 22.90 | 14.27 | 36.14 | 14.27 | 36.14 | 12.19 | 27.07 | 11.83 | 27.34 |
| RFT-List (List-CoT, MRR, LP=0.9) | QA | List-CoT | List-MRR-0.9 | List-CoT | 18.80 | 34.93 | 18.00 | 34.59 | 8.67 | 22.50 | 8.67 | 22.22 | 6.38 | 17.89 | 6.26 | 16.74 | 14.27 | 36.14 | 14.27 | 36.14 | 12.19 | 27.07 | 11.83 | 27.34 |
| AlphaMed 7B: RFT-List (Acc, LP=0.3) | QA | List-CoT | List-Acc-LP-0.3 | List-CoT | 20.52 | 44.28 | 20.24 | 42.63 | 9.50 | 24.86 | 9.18 | 23.55 | 6.76 | 20.37 | 6.53 | 18.78 | 17.12 | 44.70 | 17.05 | 42.80 | 13.48 | 33.55 | 13.25 | 31.94 |
| m1 7B 23K: RFT-List (Acc, LP=0.3) | QA | List-CoT | List-Acc-LP-0.3 | List-CoT | 20.19 | 45.01 | 19.67 | 44.13 | 10.00 | 23.99 | 9.73 | 23.16 | 5.85 | 20.18 | 5.68 | 19.05 | 16.58 | 43.75 | 16.32 | 42.28 | 13.16 | 33.23 | 12.82 | 32.16 |
| OpenThinker3 7B: RFT-List (Acc, LP=0.3) | QA | List-CoT | List-Acc-LP-0.3 | List-CoT | 14.11 | 35.36 | 13.63 | 32.81 | 7.66 | 21.06 | 7.52 | 20.28 | 4.12 | 15.29 | 3.93 | 13.82 | 11.01 | 35.74 | 10.67 | 33.13 | 9.22 | 26.86 | 8.94 | 25.01 |
| Qwen2.5 3B: RFT-List (Acc, LP=0.3) | QA | List-CoT | List-Acc-LP-0.3 | List-CoT | 14.84 | 34.39 | 14.52 | 32.97 | 8.30 | 20.60 | 8.00 | 19.43 | 5.47 | 17.79 | 5.15 | 16.04 | 11.01 | 33.83 | 10.73 | 31.79 | 9.90 | 26.65 | 9.60 | 25.06 |

Table 33: Response length (mean ± standard deviation) for MCQ answer format for the experiments from Section 4.2.

| | Dataset | Prior Prompt | Reward Function | Prompting | MedQA | MedMCQA | MXQA | MLUP-H | Average |
|---|---|---|---|---|---|---|---|---|---|
| RFT-MCQ (Baseline, No Prompt) | MCQ | No Prompt | MCQ-Acc | MCQ-CoT | 529.06 ± 660.03 | 293.64 ± 281.92 | 573.46 ± 682.75 | 497.52 ± 559.83 | 473.42 |
| RFT-MCQ (Baseline, MCQ) | MCQ | MCQ | MCQ-Acc | MCQ | 176.62 ± 90.24 | 77.53 ± 64.76 | 141.82 ± 182.70 | 116.84 ± 107.32 | 128.20 |
| RFT-MCQ (Baseline, MCQ) | MCQ | MCQ | MCQ-Acc | MCQ-CoT | 231.71 ± 90.31 | 173.14 ± 128.79 | 222.21 ± 126.87 | 263.59 ± 143.15 | 222.66 |
| RFT-MCQ (MCQ-CoT) | MCQ | MCQ-CoT | MCQ-Acc | MCQ-CoT | 249.18 ± 67.22 | 136.57 ± 156.45 | 202.30 ± 91.73 | 199.08 ± 107.90 | 203.80 |
| RFT-QA (Baseline, No Prompt) | QA | No Prompt | QA-Acc | MCQ-CoT | 330.54 ± 670.02 | 171.21 ± 329.32 | 302.67 ± 403.41 | 229.17 ± 107.92 | 258.40 |
| RFT-QA (Baseline, QA) | QA | QA | QA-Acc | MCQ | 178.06 ± 332.86 | 66.85 ± 59.55 | 155.64 ± 247.44 | 117.39 ± 90.03 | 129.48 |
| RFT-QA (Baseline, QA) | QA | QA | QA-Acc | MCQ-CoT | 277.26 ± 398.19 | 140.54 ± 116.14 | 282.95 ± 296.65 | 212.52 ± 115.77 | 228.32 |
| RFT-QA (QA-CoT) | QA | QA-CoT | QA-Acc | MCQ-CoT | 323.16 ± 243.99 | 192.73 ± 163.01 | 382.19 ± 492.91 | 285.82 ± 305.12 | 295.97 |
| RFT-List (Baseline, List, Acc) | QA | List | List-Acc | MCQ | 124.63 ± 242.31 | 40.48 ± 51.09 | 108.21 ± 86.27 | 64.19 ± 73.56 | 84.38 |
| RFT-List (Baseline, List, Acc) | QA | List | List-Acc | MCQ-CoT | 220.93 ± 74.88 | 131.90 ± 54.69 | 224.68 ± 186.61 | 194.70 ± 96.21 | 193.05 |
| RFT-List (List-CoT, Acc) | QA | List-CoT | List-Acc | MCQ-CoT | 244.73 ± 326.16 | 132.18 ± 113.00 | 244.32 ± 185.84 | 208.52 ± 295.76 | 207.44 |
| RFT-List (List, MRR) | QA | List | List-MRR | MCQ | 21.51 ± 47.27 | 15.78 ± 28.55 | 45.85 ± 176.77 | 25.70 ± 46.81 | 27.21 |
| RFT-List (List, MRR) | QA | List | List-MRR | MCQ-CoT | 228.25 ± 65.15 | 159.50 ± 186.67 | 221.85 ± 96.12 | 238.78 ± 296.46 | 212.09 |
| RFT-List (List-CoT, MRR) | QA | List-CoT | List-MRR | MCQ-CoT | 187.05 ± 64.96 | 122.08 ± 44.46 | 208.78 ± 242.88 | 176.09 ± 83.81 | 173.50 |
| RFT-List (List-CoT, Judge-MRR-Gemini) | QA | List-CoT | LLM-List-MRR-Gemini | MCQ-CoT | 282.94 ± 98.54 | 150.54 ± 120.62 | 278.70 ± 174.24 | 243.20 ± 128.07 | 238.85 |
| RFT-MCQ (MCQ-CoT, 4 Epochs) | MCQ | MCQ-CoT | MCQ-Acc | MCQ-CoT | 325.86 ± 78.29 | 202.53 ± 66.22 | 435.50 ± 151.71 | 379.03 ± 130.02 | 335.73 |
| RFT-MCQ (MCQ-CoT, No format) | MCQ | No Prompt | MCQ-Acc-NF | MCQ-CoT | 206.18 ± 232.18 | 132.61 ± 152.00 | 190.21 ± 78.16 | 179.31 ± 89.46 | 177.08 |
| RFT-List (List-CoT, Judge-MRR) | QA | List-CoT | LLM-List-MRR | MCQ-CoT | 195.16 ± 395.85 | 114.92 ± 211.49 | 199.62 ± 491.15 | 160.93 ± 405.08 | 167.66 |
| RFT-List (List-CoT, Judge-MRR-Simple) | QA | List-CoT | LLM-List-MRR-Simple | MCQ-CoT | 212.04 ± 324.82 | 127.65 ± 53.78 | 236.39 ± 299.36 | 188.67 ± 97.02 | 191.19 |
| RFT-MCQ (Acc) + QA (Acc) | MCQ,QA | MCQ-CoT,QA-CoT | MCQ-Acc,QA-Acc | MCQ-CoT | 223.27 ± 55.45 | 149.48 ± 149.02 | 292.36 ± 74.87 | 263.07 ± 400.57 | 232.05 |
| RFT-MCQ (Acc) + List (Acc) | MCQ,QA | MCQ-CoT,List-CoT | MCQ-Acc,List-Acc | MCQ-CoT | 264.22 ± 234.10 | 191.86 ± 115.36 | 241.67 ± 187.56 | 284.41 ± 295.77 | 245.54 |
| RFT-MCQ (Acc) + List (MRR) | MCQ,QA | MCQ-CoT,List-CoT | MCQ-Acc,List-MRR | MCQ-CoT | 254.12 ± 234.90 | 149.82 ± 35.66 | 217.45 ± 88.60 | 215.51 ± 298.04 | 209.22 |
| Qwen2.5 3B: RFT-MCQ | MCQ | MCQ-CoT | MCQ-Acc | MCQ-CoT | 211.86 ± 230.01 | 153.12 ± 151.00 | 219.50 ± 179.85 | 203.22 ± 289.73 | 196.93 |
| Qwen2.5 3B: RFT-QA | QA | QA-CoT | QA-Acc | MCQ-CoT | 255.43 ± 232.29 | 173.03 ± 183.68 | 342.18 ± 400.94 | 285.33 ± 403.54 | 263.99 |
| Qwen2.5 3B: RFT-List (Acc) | QA | List-CoT | List-Acc | MCQ-CoT | 345.33 ± 109.04 | 270.55 ± 237.46 | 282.93 ± 266.59 | 359.82 ± 137.04 | 314.66 |
| Qwen2.5 3B: RFT-List (MRR) | QA | List-CoT | List-MRR | MCQ-CoT | 380.28 ± 455.47 | 266.14 ± 82.48 | 323.41 ± 264.62 | 365.62 ± 139.18 | 333.86 |
| Qwen3 4B: RFT-MCQ | MCQ | MCQ-CoT | MCQ-Acc | MCQ-CoT | 967.36 ± 727.82 | 508.20 ± 532.22 | 1133.19 ± 626.74 | 769.45 ± 562.13 | 844.55 |
| Qwen3 4B: RFT-QA | QA | QA-CoT | QA-Acc | MCQ-CoT | 999.76 ± 948.85 | 583.07 ± 749.84 | 1307.53 ± 948.42 | 890.22 ± 845.93 | 945.14 |
| Qwen3 4B: RFT-List (Acc) | QA | List-CoT | List-Acc | MCQ-CoT | 929.42 ± 816.84 | 493.04 ± 586.96 | 1109.69 ± 718.85 | 732.17 ± 625.36 | 816.08 |
| Qwen3 4B: RFT-List (MRR) | QA | List-CoT | List-MRR | MCQ-CoT | 1055.17 ± 919.84 | 545.32 ± 622.42 | 1235.89 ± 757.93 | 824.04 ± 683.32 | 915.11 |
| OpenThinker3 7B: RFT-MCQ | MCQ | MCQ-CoT | MCQ-Acc | MCQ-CoT | 1354.35 ± 662.76 | 977.25 ± 611.76 | 1577.16 ± 721.77 | 1344.25 ± 667.48 | 1313.24 |
| OpenThinker3 7B: RFT-QA | QA | QA-CoT | QA-Acc | MCQ-CoT | 1256.72 ± 535.40 | 966.07 ± 515.33 | 1507.42 ± 625.86 | 1349.09 ± 652.15 | 1269.83 |
| OpenThinker3 7B: RFT-List (Acc) | QA | List-CoT | List-Acc | MCQ-CoT | 1501.23 ± 869.45 | 1107.86 ± 738.65 | 1832.06 ± 929.06 | 1631.76 ± 1016.75 | 1518.23 |
| OpenThinker3 7B: RFT-List (MRR) | QA | List-CoT | List-MRR | MCQ-CoT | 1415.60 ± 815.55 | 1073.69 ± 707.64 | 1799.80 ± 928.06 | 1611.12 ± 1080.28 | 1475.05 |
| m1 7B 23K: RFT-MCQ | MCQ | MCQ-CoT | MCQ-Acc | MCQ-CoT | 1375.52 ± 715.10 | 980.54 ± 582.88 | 1826.03 ± 838.53 | 1397.08 ± 854.26 | 1394.79 |
| m1 7B 23K: RFT-QA | QA | QA-CoT | QA-Acc | MCQ-CoT | 1113.09 ± 618.71 | 848.73 ± 584.92 | 1489.26 ± 921.50 | 1278.22 ± 1049.86 | 1182.33 |
| m1 7B 23K: RFT-List (Acc) | QA | List-CoT | List-Acc | MCQ-CoT | 1424.52 ± 1174.36 | 1033.81 ± 1003.63 | 1867.50 ± 1173.63 | 1583.00 ± 1479.20 | 1477.21 |
| m1 7B 23K: RFT-List (MRR) | QA | List-CoT | List-MRR | MCQ-CoT | 1398.42 ± 837.17 | 1178.52 ± 959.30 | 1958.63 ± 1097.68 | 1635.26 ± 1257.96 | 1542.71 |
| AlphaMed 7B: RFT-MCQ | MCQ | MCQ-CoT | MCQ-Acc | MCQ-CoT | 391.99 ± 502.29 | 225.23 ± 157.44 | 408.06 ± 373.36 | 341.60 ± 110.20 | 341.72 |
| AlphaMed 7B: RFT-QA | QA | QA-CoT | QA-Acc | MCQ-CoT | 306.82 ± 114.31 | 172.14 ± 194.50 | 279.55 ± 126.54 | 263.30 ± 129.99 | 255.45 |
| AlphaMed 7B: RFT-List (Acc) | QA | List-CoT | List-Acc | MCQ-CoT | 281.13 ± 87.50 | 167.18 ± 130.20 | 294.71 ± 121.04 | 279.52 ± 115.69 | 255.63 |
| AlphaMed 7B: RFT-List (MRR) | QA | List-CoT | List-MRR | MCQ-CoT | 292.89 ± 85.25 | 184.36 ± 70.76 | 332.02 ± 200.01 | 297.19 ± 112.16 | 276.62 |
| RFT-List (List-CoT, MRR, LP=0) | QA | List-CoT | List-MRR-0 | MCQ-CoT | 190.76 ± 231.12 | 127.77 ± 182.62 | 193.30 ± 72.01 | 168.04 ± 77.98 | 169.97 |
| RFT-List (List-CoT, MRR, LP=0.1) | QA | List-CoT | List-MRR-0.1 | MCQ-CoT | 204.93 ± 235.12 | 126.46 ± 113.36 | 190.61 ± 182.44 | 181.12 ± 296.79 | 175.78 |
| RFT-List (List-CoT, MRR, LP=0.3) | QA | List-CoT | List-MRR-0.3 | MCQ-CoT | 187.04 ± 234.85 | 117.21 ± 185.19 | 190.08 ± 370.43 | 145.12 ± 297.39 | 159.86 |
| RFT-List (List-CoT, MRR, LP=0.5) | QA | List-CoT | List-MRR-0.5 | MCQ-CoT | 190.67 ± 76.47 | 119.39 ± 48.68 | 169.84 ± 81.90 | 145.40 ± 93.76 | 156.32 |
| RFT-List (List-CoT, MRR, LP=0.7) | QA | List-CoT | List-MRR-0.7 | MCQ-CoT | 187.11 ± 238.88 | 122.79 ± 155.43 | 197.52 ± 245.17 | 135.26 ± 103.58 | 160.67 |
| RFT-List (List-CoT, MRR, LP=0.9) | QA | List-CoT | List-MRR-0.9 | MCQ-CoT | 104.93 ± 70.75 | 23.91 ± 38.67 | 63.70 ± 65.73 | 46.31 ± 290.60 | 59.71 |
| AlphaMed 7B: RFT-List (Acc, LP=0.3) | QA | List-CoT | List-Acc-LP-0.3 | MCQ-CoT | 244.91 ± 75.56 | 148.10 ± 62.38 | 233.57 ± 105.36 | 222.71 ± 99.02 | 212.32 |
| m1 7B 23K: RFT-List (Acc, LP=0.3) | QA | List-CoT | List-Acc-LP-0.3 | MCQ-CoT | 1298.38 ± 662.69 | 953.24 ± 838.02 | 1824.68 ± 1118.28 | 1402.28 ± 1035.36 | 1369.64 |
| OpenThinker3 7B: RFT-List (Acc, LP=0.3) | QA | List-CoT | List-Acc-LP-0.3 | MCQ-CoT | 1328.06 ± 668.88 | 950.61 ± 564.96 | 1536.51 ± 689.73 | 1334.80 ± 758.55 | 1287.50 |
| Qwen2.5 3B: RFT-List (Acc, LP=0.3) | QA | List-CoT | List-Acc-LP-0.3 | MCQ-CoT | 222.24 ± 234.46 | 159.13 ± 120.15 | 251.06 ± 261.46 | 285.32 ± 133.61 | 229.44 |

Table 34: Response length (mean ± standard deviation) for QA answer format for the experiments from Section 4.2.

| | Dataset | Prior Prompt | Reward Function | Prompting | MedQA | MedMCQA | MXQA | MLUP-H | Average |
|---|---|---|---|---|---|---|---|---|---|
| RFT-QA (Baseline, No Prompt) | QA | No Prompt | QA-Acc | QA-CoT | 432.44 ± 1002.46 | 397.70 ± 1136.40 | 507.90 ± 1198.04 | 314.13 ± 721.69 | 413.04 |
| RFT-QA (Baseline, QA) | QA | QA | QA-Acc | QA | 197.96 ± 333.22 | 109.11 ± 57.36 | 190.82 ± 359.00 | 130.58 ± 74.56 | 157.12 |
| RFT-QA (Baseline, QA) | QA | QA | QA-Acc | QA-CoT | 315.43 ± 645.61 | 184.61 ± 385.28 | 281.30 ± 436.33 | 206.47 ± 308.15 | 246.95 |
| RFT-QA (QA-CoT) | QA | QA-CoT | QA-Acc | QA-CoT | 333.82 ± 404.39 | 229.49 ± 258.57 | 338.98 ± 212.60 | 258.20 ± 110.95 | 290.12 |
| RFT-MCQ (Baseline, No Prompt) | MCQ | No Prompt | MCQ-Acc | QA-CoT | 352.93 ± 405.16 | 297.23 ± 665.80 | 361.84 ± 359.53 | 273.12 ± 129.88 | 321.28 |
| RFT-MCQ (Baseline, MCQ) | MCQ | MCQ | MCQ-Acc | QA | 140.94 ± 80.24 | 82.22 ± 64.89 | 149.78 ± 91.19 | 109.25 ± 78.78 | 120.55 |
| RFT-MCQ (Baseline, MCQ) | MCQ | MCQ | MCQ-Acc | QA-CoT | 216.32 ± 87.21 | 169.16 ± 84.79 | 231.81 ± 94.38 | 193.66 ± 90.98 | 202.74 |
| RFT-MCQ (MCQ-CoT) | MCQ | MCQ-CoT | MCQ-Acc | QA-CoT | 183.95 ± 63.42 | 133.19 ± 60.97 | 194.40 ± 68.97 | 155.69 ± 66.20 | 166.81 |
| RFT-List (Baseline, List, Acc) | QA | List | List-Acc | QA | 164.01 ± 253.15 | 111.97 ± 202.70 | 165.53 ± 207.60 | 132.05 ± 109.84 | 143.39 |
| RFT-List (Baseline, List, Acc) | QA | List | List-Acc | QA-CoT | 207.89 ± 401.54 | 139.15 ± 182.24 | 197.26 ± 76.08 | 156.66 ± 70.12 | 175.24 |
| RFT-List (List-CoT, Acc) | QA | List-CoT | List-Acc | QA-CoT | 238.33 ± 606.57 | 150.06 ± 348.68 | 226.65 ± 469.20 | 164.94 ± 304.21 | 195.00 |
| RFT-List (List, MRR) | QA | List | List-MRR | QA | 109.74 ± 96.56 | 72.32 ± 87.51 | 116.82 ± 99.87 | 101.04 ± 313.73 | 99.98 |
| RFT-List (List, MRR) | QA | List | List-MRR | QA-CoT | 195.21 ± 238.29 | 144.15 ± 66.33 | 208.19 ± 311.32 | 159.03 ± 67.04 | 176.64 |
| RFT-List (List-CoT, MRR) | QA | List-CoT | List-MRR | QA-CoT | 172.84 ± 328.54 | 145.95 ± 388.67 | 165.41 ± 57.77 | 139.83 ± 52.68 | 156.01 |
| RFT-List (List-CoT, Judge-MRR) | QA | List-CoT | LLM-List-MRR | QA-CoT | 242.11 ± 854.74 | 144.45 ± 460.66 | 230.31 ± 746.17 | 151.78 ± 424.00 | 192.16 |
| RFT-MCQ (MCQ-CoT, 4 Epochs) | MCQ | MCQ-CoT | MCQ-Acc | QA-CoT | 229.24 ± 91.13 | 168.53 ± 92.16 | 243.08 ± 265.24 | 199.44 ± 99.84 | 210.07 |
| RFT-MCQ (MCQ-CoT, No format) | MCQ | No Prompt | MCQ-Acc-NF | QA-CoT | 169.14 ± 54.71 | 134.18 ± 49.48 | 179.33 ± 61.03 | 149.91 ± 61.48 | 158.14 |
| RFT-List (List-CoT, Judge-MRR-Gemini) | QA | List-CoT | LLM-List-MRR-Gemini | QA-CoT | 230.34 ± 340.88 | 154.86 ± 61.19 | 232.24 ± 87.67 | 182.32 ± 77.84 | 199.94 |
| RFT-List (List-CoT, Judge-MRR-Simple) | QA | List-CoT | LLM-List-MRR-Simple | QA-CoT | 174.30 ± 236.07 | 132.00 ± 249.00 | 193.59 ± 356.67 | 150.93 ± 301.78 | 162.70 |
| RFT-MCQ (Acc) + QA (Acc) | MCQ,QA | MCQ-CoT,QA-CoT | MCQ-Acc,QA-Acc | QA-CoT | 234.12 ± 327.94 | 173.61 ± 52.23 | 230.53 ± 87.67 | 194.29 ± 67.50 | 208.24 |
| RFT-MCQ (Acc) + List (Acc) | MCQ,QA | MCQ-CoT,List-CoT | MCQ-Acc,List-Acc | QA-CoT | 209.97 ± 135.21 | 181.50 ± 184.99 | 212.16 ± 66.68 | 197.59 ± 302.26 | 200.31 |
| RFT-MCQ (Acc) + List (MRR) | MCQ,QA | MCQ-CoT,List-CoT | MCQ-Acc,List-MRR | QA-CoT | 182.22 ± 236.95 | 142.83 ± 250.70 | 195.99 ± 310.87 | 159.73 ± 303.55 | 170.19 |
| Qwen2.5 3B: RFT-MCQ | MCQ | MCQ-CoT | MCQ-Acc | QA-CoT | 201.90 ± 232.77 | 162.63 ± 42.00 | 213.34 ± 307.08 | 178.58 ± 50.93 | 189.11 |
| Qwen2.5 3B: RFT-QA | QA | QA-CoT | QA-Acc | QA-CoT | 321.63 ± 235.17 | 263.17 ± 347.28 | 332.50 ± 307.87 | 284.53 ± 301.63 | 300.46 |
| Qwen2.5 3B: RFT-List (Acc) | QA | List-CoT | List-Acc | QA-CoT | 230.47 ± 243.03 | 200.84 ± 254.94 | 233.08 ± 201.22 | 199.69 ± 80.62 | 216.02 |
| Qwen2.5 3B: RFT-List (MRR) | QA | List-CoT | List-MRR | QA-CoT | 249.39 ± 90.68 | 210.89 ± 78.76 | 255.26 ± 195.31 | 243.26 ± 423.55 | 239.70 |
| Qwen3 4B: RFT-MCQ | MCQ | MCQ-CoT | MCQ-Acc | QA-CoT | 610.41 ± 468.46 | 501.89 ± 703.79 | 656.59 ± 436.12 | 446.52 ± 426.97 | 553.85 |
| Qwen3 4B: RFT-QA | QA | QA-CoT | QA-Acc | QA-CoT | 722.40 ± 755.68 | 621.48 ± 838.28 | 804.42 ± 778.89 | 543.68 ± 571.10 | 673.00 |
| Qwen3 4B: RFT-List (Acc) | QA | List-CoT | List-Acc | QA-CoT | 540.34 ± 339.57 | 494.94 ± 873.36 | 591.03 ± 528.84 | 413.53 ± 351.13 | 509.96 |
| Qwen3 4B: RFT-List (MRR) | QA | List-CoT | List-MRR | QA-CoT | 719.24 ± 645.08 | 631.51 ± 997.37 | 813.48 ± 789.06 | 551.11 ± 684.62 | 678.84 |
| OpenThinker3 7B: RFT-MCQ | MCQ | MCQ-CoT | MCQ-Acc | QA-CoT | 1474.04 ± 1217.77 | 1261.28 ± 1148.17 | 1512.71 ± 1070.72 | 1254.21 ± 953.77 | 1375.56 |
| OpenThinker3 7B: RFT-QA | QA | QA-CoT | QA-Acc | QA-CoT | 1113.08 ± 459.48 | 1008.73 ± 586.59 | 1176.12 ± 599.89 | 1052.97 ± 598.95 | 1087.72 |
| OpenThinker3 7B: RFT-List (Acc) | QA | List-CoT | List-Acc | QA-CoT | 1254.37 ± 784.65 | 1141.69 ± 901.81 | 1319.16 ± 750.65 | 1165.66 ± 778.19 | 1220.22 |
| OpenThinker3 7B: RFT-List (MRR) | QA | List-CoT | List-MRR | QA-CoT | 1190.46 ± 673.34 | 1166.79 ± 930.09 | 1289.06 ± 795.35 | 1132.11 ± 865.45 | 1194.61 |
| m1 7B 23K: RFT-MCQ | MCQ | MCQ-CoT | MCQ-Acc | QA-CoT | 1131.54 ± 741.99 | 982.54 ± 585.13 | 1283.87 ± 799.90 | 1020.40 ± 683.03 | 1104.59 |
| m1 7B 23K: RFT-QA | QA | QA-CoT | QA-Acc | QA-CoT | 814.45 ± 421.60 | 726.60 ± 459.46 | 940.77 ± 551.10 | 717.45 ± 376.24 | 799.82 |
| m1 7B 23K: RFT-List (Acc) | QA | List-CoT | List-Acc | QA-CoT | 1102.75 ± 1090.86 | 1070.58 ± 1160.48 | 1343.82 ± 1340.30 | 1073.31 ± 1250.26 | 1147.61 |
| m1 7B 23K: RFT-List (MRR) | QA | List-CoT | List-MRR | QA-CoT | 1223.82 ± 1179.41 | 1174.57 ± 1172.67 | 1472.91 ± 1344.32 | 1122.70 ± 1117.05 | 1248.50 |
| AlphaMed 7B: RFT-MCQ | MCQ | MCQ-CoT | MCQ-Acc | QA-CoT | 383.35 ± 930.58 | 261.71 ± 595.90 | 335.53 ± 698.01 | 254.68 ± 424.40 | 308.82 |
| AlphaMed 7B: RFT-QA | QA | QA-CoT | QA-Acc | QA-CoT | 296.74 ± 564.64 | 193.48 ± 257.98 | 285.55 ± 421.96 | 207.53 ± 100.18 | 245.82 |
| AlphaMed 7B: RFT-List (Acc) | QA | List-CoT | List-Acc | QA-CoT | 231.53 ± 402.91 | 178.72 ± 493.23 | 224.28 ± 313.48 | 170.86 ± 82.34 | 201.35 |
| AlphaMed 7B: RFT-List (MRR) | QA | List-CoT | List-MRR | QA-CoT | 231.50 ± 403.26 | 158.69 ± 253.98 | 234.58 ± 399.18 | 167.18 ± 80.51 | 197.99 |
| RFT-List (List-CoT, MRR, LP=0) | QA | List-CoT | List-MRR-0 | QA-CoT | 161.31 ± 57.18 | 151.97 ± 387.99 | 193.24 ± 433.85 | 156.14 ± 301.09 | 165.66 |
| RFT-List (List-CoT, MRR, LP=0.1) | QA | List-CoT | List-MRR-0.1 | QA-CoT | 180.52 ± 400.85 | 158.07 ± 519.68 | 192.54 ± 468.38 | 147.16 ± 301.81 | 169.57 |
| RFT-List (List-CoT, MRR, LP=0.3) | QA | List-CoT | List-MRR-0.3 | QA-CoT | 207.23 ± 648.51 | 124.97 ± 303.54 | 191.27 ± 469.10 | 128.72 ± 63.07 | 163.05 |
| RFT-List (List-CoT, MRR, LP=0.5) | QA | List-CoT | List-MRR-0.5 | QA-CoT | 156.96 ± 236.35 | 111.05 ± 43.68 | 156.53 ± 60.98 | 128.68 ± 57.80 | 138.31 |
| RFT-List (List-CoT, MRR, LP=0.7) | QA | List-CoT | List-MRR-0.7 | QA-CoT | 184.95 ± 401.18 | 130.23 ± 303.87 | 198.13 ± 435.43 | 149.82 ± 303.94 | 165.78 |
| RFT-List (List-CoT, MRR, LP=0.9) | QA | List-CoT | List-MRR-0.9 | QA-CoT | 71.79 ± 86.99 | 52.64 ± 65.27 | 68.50 ± 89.23 | 59.79 ± 74.88 | 63.18 |
| AlphaMed 7B: RFT-List (Acc, LP=0.3) | QA | List-CoT | List-Acc-LP-0.3 | QA-CoT | 183.91 ± 69.14 | 146.01 ± 182.71 | 196.74 ± 311.27 | 165.74 ± 303.43 | 173.10 |
| m1 7B 23K: RFT-List (Acc, LP=0.3) | QA | List-CoT | List-Acc-LP-0.3 | QA-CoT | 926.48 ± 528.48 | 791.97 ± 488.93 | 1038.29 ± 663.29 | 802.68 ± 467.72 | 889.85 |
| OpenThinker3 7B: RFT-List (Acc, LP=0.3) | QA | List-CoT | List-Acc-LP-0.3 | QA-CoT | 1121.20 ± 556.09 | 986.88 ± 532.46 | 1164.90 ± 593.27 | 1033.99 ± 657.10 | 1076.74 |
| Qwen2.5 3B: RFT-List (Acc, LP=0.3) | QA | List-CoT | List-Acc-LP-0.3 | QA-CoT | 188.58 ± 55.69 | 161.44 ± 248.54 | 193.32 ± 60.30 | 167.12 ± 55.51 | 177.61 |

Table 35: Response length (mean ± standard deviation) for a ranked-list answer format for the experiments from Section 4.2.

| | Dataset | Prior Prompt | Reward Function | Prompting | MedQA | MedMCQA | MXQA | MLUP-H | Average |
|---|---|---|---|---|---|---|---|---|---|
| RFT-List (Baseline, List, Acc) | QA | List | List-Acc | List | 7977.79 ± 1277.05 | 7717.38 ± 1891.88 | 7958.86 ± 1326.88 | 7800.39 ± 1707.57 | 7863.60 |
| RFT-List (Baseline, List, Acc) | QA | List | List-Acc | List-CoT | 3379.40 ± 3956.56 | 2825.40 ± 3826.98 | 3380.38 ± 3959.44 | 3149.64 ± 3919.37 | 3183.70 |
| RFT-List (List-CoT, Acc) | QA | List-CoT | List-Acc | List-CoT | 183.77 ± 233.69 | 147.13 ± 246.70 | 181.84 ± 182.46 | 147.06 ± 45.10 | 164.95 |
| RFT-List (List, MRR) | QA | List | List-MRR | List | 80.58 ± 232.64 | 75.62 ± 303.56 | 73.10 ± 31.02 | 85.18 ± 368.82 | 78.62 |
| RFT-List (List, MRR) | QA | List | List-MRR | List-CoT | 123.95 ± 35.36 | 101.17 ± 176.37 | 130.06 ± 252.26 | 115.37 ± 300.03 | 117.64 |
| RFT-List (List-CoT, MRR) | QA | List-CoT | List-MRR | List-CoT | 409.30 ± 1380.15 | 265.23 ± 1059.86 | 342.08 ± 1194.10 | 255.61 ± 980.80 | 318.05 |
| RFT-List (List-CoT, Judge-MRR) | QA | List-CoT | LLM-List-MRR | List-CoT | 150.56 ± 327.42 | 108.58 ± 177.57 | 143.49 ± 49.30 | 126.61 ± 300.57 | 132.31 |
| RFT-MCQ (Baseline, No Prompt) | MCQ | No Prompt | MCQ-Acc | List-CoT | 162.24 ± 65.98 | 123.02 ± 348.54 | 165.23 ± 67.44 | 135.25 ± 302.00 | 146.44 |
| RFT-MCQ (Baseline, MCQ) | MCQ | MCQ | MCQ-Acc | List | 26.75 ± 15.78 | 17.96 ± 11.82 | 28.86 ± 19.72 | 22.13 ± 14.45 | 23.92 |
| RFT-MCQ (Baseline, MCQ) | MCQ | MCQ | MCQ-Acc | List-CoT | 122.76 ± 53.89 | 80.84 ± 40.83 | 130.25 ± 64.71 | 96.81 ± 52.09 | 107.67 |
| RFT-MCQ (MCQ-CoT) | MCQ | MCQ-CoT | MCQ-Acc | List-CoT | 34.24 ± 35.95 | 29.84 ± 247.83 | 37.91 ± 180.96 | 27.93 ± 24.40 | 32.48 |
| RFT-QA (Baseline, No Prompt) | QA | No Prompt | QA-Acc | List-CoT | 180.77 ± 332.36 | 129.07 ± 250.78 | 169.92 ± 261.72 | 140.56 ± 304.27 | 155.08 |
| RFT-QA (Baseline, QA) | QA | QA | QA-Acc | List | 48.89 ± 23.26 | 31.86 ± 15.97 | 49.08 ± 24.28 | 38.58 ± 21.30 | 42.10 |
| RFT-QA (Baseline, QA) | QA | QA | QA-Acc | List-CoT | 160.03 ± 236.06 | 104.87 ± 41.05 | 149.65 ± 57.38 | 117.44 ± 53.06 | 133.00 |
| RFT-QA (QA-CoT) | QA | QA-CoT | QA-Acc | List-CoT | 61.37 ± 239.85 | 29.37 ± 27.61 | 52.42 ± 51.27 | 40.43 ± 44.25 | 45.90 |
| RFT-MCQ (MCQ-CoT, 4 Epochs) | MCQ | MCQ-CoT | MCQ-Acc | List-CoT | 201.07 ± 87.92 | 134.61 ± 187.36 | 207.87 ± 197.89 | 164.63 ± 90.19 | 177.04 |
| RFT-MCQ (MCQ-CoT, No format) | MCQ | No Prompt | MCQ-Acc-NF | List-CoT | 138.93 ± 53.27 | 101.75 ± 41.95 | 144.04 ± 57.74 | 127.44 ± 302.30 | 128.04 |
| RFT-List (List-CoT, Judge-MRR-Simple) | QA | List-CoT | LLM-List-MRR-Simple | List-CoT | 156.06 ± 43.61 | 114.27 ± 37.69 | 159.25 ± 46.14 | 126.77 ± 42.46 | 139.09 |
| RFT-List (List-CoT, Judge-MRR-Gemini) | QA | List-CoT | LLM-List-MRR-Gemini | List-CoT | 160.31 ± 287.29 | 113.02 ± 35.09 | 157.10 ± 42.85 | 132.22 ± 43.51 | 140.66 |
| RFT-MCQ (Acc) + QA (Acc) | MCQ,QA | MCQ-CoT,QA-CoT | MCQ-Acc,QA-Acc | List-CoT | 137.23 ± 242.17 | 103.24 ± 252.99 | 121.84 ± 194.14 | 101.34 ± 70.48 | 115.91 |
| RFT-MCQ (Acc) + List (Acc) | MCQ,QA | MCQ-CoT,List-CoT | MCQ-Acc,List-Acc | List-CoT | 206.07 ± 512.63 | 154.45 ± 423.94 | 224.03 ± 632.92 | 186.76 ± 593.83 | 192.83 |
| RFT-MCQ (Acc) + List (MRR) | MCQ,QA | MCQ-CoT,List-CoT | MCQ-Acc,List-MRR | List-CoT | 143.37 ± 38.17 | 121.26 ± 176.64 | 144.46 ± 38.48 | 123.71 ± 37.68 | 133.20 |
| Qwen2.5 3B: RFT-MCQ | MCQ | MCQ-CoT | MCQ-Acc | List-CoT | 192.78 ± 63.89 | 158.83 ± 181.50 | 198.30 ± 62.74 | 165.53 ± 67.83 | 178.86 |
| Qwen2.5 3B: RFT-QA | QA | QA-CoT | QA-Acc | List-CoT | 313.74 ± 396.93 | 336.72 ± 898.91 | 355.40 ± 672.24 | 301.23 ± 588.92 | 326.77 |
| Qwen2.5 3B: RFT-List (Acc) | QA | List-CoT | List-Acc | List-CoT | 7981.94 ± 1267.20 | 7839.86 ± 1628.74 | 7903.21 ± 1474.86 | 7797.95 ± 1716.17 | 7880.74 |
| Qwen2.5 3B: RFT-List (MRR) | QA | List-CoT | List-MRR | List-CoT | 7466.06 ± 2271.20 | 6866.39 ± 2958.28 | 7472.22 ± 2267.48 | 6922.35 ± 2905.46 | 7181.76 |
| Qwen3 4B: RFT-MCQ | MCQ | MCQ-CoT | MCQ-Acc | List-CoT | 485.37 ± 521.74 | 380.66 ± 649.82 | 545.02 ± 584.60 | 399.98 ± 637.79 | 452.76 |
| Qwen3 4B: RFT-QA | QA | QA-CoT | QA-Acc | List-CoT | 564.63 ± 481.94 | 470.46 ± 698.12 | 618.16 ± 522.63 | 440.12 ± 440.77 | 523.34 |
| Qwen3 4B: RFT-List (Acc) | QA | List-CoT | List-Acc | List-CoT | 8091.63 ± 874.48 | 7734.96 ± 1831.24 | 8046.00 ± 1057.95 | 7841.95 ± 1616.59 | 7928.64 |
| Qwen3 4B: RFT-List (MRR) | QA | List-CoT | List-MRR | List-CoT | 627.55 ± 563.57 | 562.06 ± 805.94 | 732.16 ± 857.50 | 481.87 ± 413.29 | 600.91 |
| OpenThinker3 7B: RFT-MCQ | MCQ | MCQ-CoT | MCQ-Acc | List-CoT | 1106.96 ± 1054.86 | 930.15 ± 932.33 | 1181.00 ± 1017.38 | 947.84 ± 805.67 | 1041.49 |
| OpenThinker3 7B: RFT-QA | QA | QA-CoT | QA-Acc | List-CoT | 815.42 ± 376.80 | 729.65 ± 484.47 | 860.92 ± 453.05 | 756.26 ± 552.82 | 790.56 |
| OpenThinker3 7B: RFT-List (Acc) | QA | List-CoT | List-Acc | List-CoT | 8127.28 ± 671.54 | 8032.82 ± 1041.90 | 8131.62 ± 630.39 | 8052.70 ± 956.61 | 8086.10 |
| OpenThinker3 7B: RFT-List (MRR) | QA | List-CoT | List-MRR | List-CoT | 8106.50 ± 771.38 | 7972.78 ± 1247.39 | 8105.47 ± 767.94 | 8006.78 ± 1145.71 | 8047.88 |
| m1 7B 23K: RFT-MCQ | MCQ | MCQ-CoT | MCQ-Acc | List-CoT | 1125.62 ± 917.94 | 996.85 ± 958.58 | 1250.13 ± 876.86 | 988.88 ± 849.25 | 1090.37 |
| m1 7B 23K: RFT-QA | QA | QA-CoT | QA-Acc | List-CoT | 835.24 ± 1121.29 | 687.59 ± 880.35 | 866.42 ± 1052.75 | 603.96 ± 688.68 | 748.30 |
| m1 7B 23K: RFT-List (Acc) | QA | List-CoT | List-Acc | List-CoT | 8130.05 ± 626.72 | 8152.35 ± 464.26 | 8173.18 ± 300.29 | 8175.90 ± 308.29 | 8157.87 |
| m1 7B 23K: RFT-List (MRR) | QA | List-CoT | List-MRR | List-CoT | 7638.32 ± 1963.90 | 7569.11 ± 2086.31 | 7819.54 ± 1627.80 | 7685.50 ± 1898.64 | 7678.12 |
| AlphaMed 7B: RFT-MCQ | MCQ | MCQ-CoT | MCQ-Acc | List-CoT | 332.31 ± 778.98 | 215.31 ± 459.06 | 331.19 ± 766.59 | 232.98 ± 424.80 | 277.95 |
| AlphaMed 7B: RFT-QA | QA | QA-CoT | QA-Acc | List-CoT | 267.25 ± 406.98 | 170.93 ± 254.13 | 252.97 ± 201.00 | 185.38 ± 85.02 | 219.13 |
| AlphaMed 7B: RFT-List (Acc) | QA | List-CoT | List-Acc | List-CoT | 2695.47 ± 3635.98 | 4582.20 ± 3956.01 | 2571.44 ± 3580.99 | 3823.51 ± 3952.14 | 3418.16 |
| AlphaMed 7B: RFT-List (MRR) | QA | List-CoT | List-MRR | List-CoT | 264.34 ± 106.99 | 157.38 ± 74.03 | 261.13 ± 101.89 | 188.60 ± 94.62 | 217.86 |
| RFT-List (List-CoT, MRR, LP=0) | QA | List-CoT | List-MRR-0 | List-CoT | 150.30 ± 325.93 | 119.60 ± 175.40 | 137.81 ± 31.18 | 119.40 ± 30.37 | 131.78 |
| RFT-List (List-CoT, MRR, LP=0.1) | QA | List-CoT | List-MRR-0.1 | List-CoT | 131.31 ± 41.85 | 114.34 ± 302.35 | 146.14 ± 308.43 | 112.45 ± 41.90 | 126.06 |
| RFT-List (List-CoT, MRR, LP=0.3) | QA | List-CoT | List-MRR-0.3 | List-CoT | 11.34 ± 5.52 | 10.89 ± 4.47 | 11.24 ± 6.61 | 10.38 ± 3.60 | 10.96 |
| RFT-List (List-CoT, MRR, LP=0.5) | QA | List-CoT | List-MRR-0.5 | List-CoT | 10.27 ± 2.49 | 9.74 ± 2.25 | 10.20 ± 2.81 | 9.58 ± 2.38 | 9.95 |
| RFT-List (List-CoT, MRR, LP=0.7) | QA | List-CoT | List-MRR-0.7 | List-CoT | 10.63 ± 2.88 | 10.46 ± 3.67 | 10.80 ± 3.59 | 10.13 ± 3.10 | 10.51 |
| RFT-List (List-CoT, MRR, LP=0.9) | QA | List-CoT | List-MRR-0.9 | List-CoT | 10.67 ± 2.82 | 10.27 ± 3.01 | 10.65 ± 3.29 | 10.13 ± 3.11 | 10.43 |
| AlphaMed 7B: RFT-List (Acc, LP=0.3) | QA | List-CoT | List-Acc-LP-0.3 | List-CoT | 178.32 ± 329.44 | 122.20 ± 49.75 | 167.40 ± 64.77 | 134.74 ± 58.33 | 150.66 |
| m1 7B 23K: RFT-List (Acc, LP=0.3) | QA | List-CoT | List-Acc-LP-0.3 | List-CoT | 777.42 ± 424.12 | 649.14 ± 457.05 | 882.74 ± 562.16 | 656.73 ± 356.51 | 741.51 |
| OpenThinker3 7B: RFT-List (Acc, LP=0.3) | QA | List-CoT | List-Acc-LP-0.3 | List-CoT | 752.86 ± 521.78 | 660.12 ± 418.33 | 785.64 ± 467.11 | 681.26 ± 508.39 | 719.97 |
| Qwen2.5 3B: RFT-List (Acc, LP=0.3) | QA | List-CoT | List-Acc-LP-0.3 | List-CoT | 139.21 ± 35.86 | 117.15 ± 175.88 | 145.37 ± 179.86 | 119.65 ± 35.07 | 130.34 |

Table 36: Metrics related to the ranked list answer format from the generated evaluation responses across benchmarks from Section 4.2.

| | Dataset | Prior Prompt | Reward Function | Prompting | MedQA | | | MedMCQA | | | MXQA | | | MLUP-H | | |
|---|---|---|---|---|---|---|---|---|---|---|---|---|---|---|---|---|
| | | | | | LL | VLL | CP | LL | VLL | CP | LL | VLL | CP | LL | VLL | CP |
| RFT-List (Baseline, List, Acc) | QA | List | List-Acc | List | 591.39 | 591.39 | 4.59 | 658.82 | 658.82 | 6.39 | 567.17 | 567.17 | 6.14 | 643.31 | 643.31 | 5.15 |
| RFT-List (Baseline, List, Acc) | QA | List | List-Acc | List-CoT | 243.62 | 244.02 | 2.38 | 229.27 | 229.38 | 3.64 | 234.58 | 234.92 | 4.24 | 265.06 | 265.06 | 2.47 |
| RFT-List (List-CoT, Acc) | QA | List-CoT | List-Acc | List-CoT | 5.98 | 5.98 | 1.86 | 5.78 | 5.79 | 2.31 | 5.97 | 5.97 | 2.37 | 6.03 | 6.03 | 1.73 |
| RFT-List (List, MRR) | QA | List | List-MRR | List | 9.55 | 9.55 | 2.36 | 9.60 | 9.60 | 2.75 | 9.39 | 9.39 | 3.08 | 9.49 | 9.49 | 2.17 |
| RFT-List (List, MRR) | QA | List | List-MRR | List-CoT | 6.70 | 6.71 | 1.86 | 6.31 | 6.31 | 2.34 | 6.97 | 6.98 | 2.36 | 8.02 | 8.03 | 1.87 |
| RFT-List (List-CoT, MRR) | QA | List-CoT | List-MRR | List-CoT | 21.90 | 21.90 | 1.89 | 15.13 | 15.15 | 2.31 | 17.93 | 17.94 | 2.41 | 12.90 | 12.90 | 1.82 |
| RFT-List (List-CoT, Judge-MRR) | QA | List-CoT | LLM-List-MRR | List-CoT | 4.60 | 4.61 | 1.51 | 4.29 | 4.29 | 1.82 | 4.56 | 4.57 | 1.84 | 4.36 | 4.37 | 1.39 |
| RFT-MCQ (Baseline, No Prompt) | MCQ | No Prompt | MCQ-Acc | List-CoT | 2.46 | 2.46 | 1.29 | 1.89 | 1.89 | 1.34 | 2.42 | 2.43 | 1.49 | 2.14 | 2.14 | 1.23 |
| RFT-MCQ (Baseline, MCQ) | MCQ | MCQ | MCQ-Acc | List | 2.34 | 2.35 | 1.40 | 1.83 | 1.83 | 1.45 | 2.32 | 2.33 | 1.61 | 2.08 | 2.08 | 1.28 |
| RFT-MCQ (Baseline, MCQ) | MCQ | MCQ | MCQ-Acc | List-CoT | 1.52 | 1.57 | 1.17 | 1.60 | 1.64 | 1.30 | 1.61 | 1.62 | 1.11 | 1.62 | 1.62 | 1.11 |
| RFT-MCQ (MCQ-CoT) | MCQ | MCQ-CoT | MCQ-Acc | List-CoT | 2.46 | 2.46 | 1.43 | 2.07 | 2.07 | 1.45 | 2.44 | 2.44 | 1.59 | 2.19 | 2.19 | 1.32 |
| RFT-QA (Baseline, No Prompt) | QA | No Prompt | QA-Acc | List-CoT | 3.22 | 3.41 | 1.41 | 2.39 | 2.41 | 1.49 | 2.96 | 3.30 | 1.75 | 2.63 | 2.75 | 1.34 |
| RFT-QA (Baseline, QA) | QA | QA | QA-Acc | List | 3.86 | 3.86 | 1.67 | 3.13 | 3.13 | 1.71 | 3.65 | 3.65 | 1.87 | 3.29 | 3.29 | 1.45 |
| RFT-QA (Baseline, QA) | QA | QA | QA-Acc | List-CoT | 3.39 | 3.39 | 1.43 | 2.65 | 2.65 | 1.53 | 3.27 | 3.27 | 1.74 | 2.88 | 2.88 | 1.36 |
| RFT-QA (QA-CoT) | QA | QA-CoT | QA-Acc | List-CoT | 0.32 | 3.71 | 1.85 | 0.27 | 2.52 | 1.55 | 0.29 | 3.12 | 1.95 | 0.29 | 2.71 | 1.35 |
| RFT-MCQ (MCQ-CoT, 4 Epochs) | MCQ | MCQ-CoT | MCQ-Acc | List-CoT | 1.71 | 1.71 | 1.18 | 1.66 | 1.66 | 1.23 | 1.81 | 1.81 | 1.34 | 1.78 | 1.78 | 1.21 |
| RFT-MCQ (MCQ-CoT, No format) | MCQ | No Prompt | MCQ-Acc-NF | List-CoT | 1.99 | 2.00 | 1.24 | 1.78 | 1.80 | 1.31 | 2.11 | 2.11 | 1.50 | 1.96 | 1.96 | 1.21 |
| RFT-List (List-CoT, Judge-MRR-Simple) | QA | List-CoT | LLM-List-MRR-Simple | List-CoT | 3.71 | 3.71 | 1.34 | 3.36 | 3.36 | 1.71 | 3.57 | 3.57 | 1.75 | 3.49 | 3.49 | 1.25 |
| RFT-List (List-CoT, Judge-MRR-Gemini) | QA | List-CoT | LLM-List-MRR-Gemini | List-CoT | 4.38 | 4.38 | 1.40 | 4.08 | 4.08 | 1.76 | 4.32 | 4.32 | 1.81 | 4.24 | 4.24 | 1.36 |
| RFT-MCQ (Acc) + QA (Acc) | MCQ,QA | MCQ-CoT,QA-CoT | MCQ-Acc,QA-Acc | List-CoT | 3.18 | 3.19 | 1.41 | 2.34 | 2.34 | 1.46 | 3.01 | 3.01 | 1.66 | 2.58 | 2.58 | 1.37 |
| RFT-MCQ (Acc) + List (Acc) | MCQ,QA | MCQ-CoT,List-CoT | MCQ-Acc,List-Acc | List-CoT | 9.47 | 9.47 | 2.15 | 9.45 | 9.45 | 2.67 | 11.45 | 11.46 | 2.89 | 11.01 | 11.01 | 2.13 |
| RFT-MCQ (Acc) + List (MRR) | MCQ,QA | MCQ-CoT,List-CoT | MCQ-Acc,List-MRR | List-CoT | 4.99 | 4.99 | 1.60 | 4.79 | 4.79 | 1.94 | 4.99 | 4.99 | 2.01 | 4.90 | 4.90 | 1.57 |
| Qwen2.5 3B: RFT-MCQ | MCQ | MCQ-CoT | MCQ-Acc | List-CoT | 2.08 | 2.08 | 1.26 | 1.69 | 1.69 | 1.36 | 2.10 | 2.10 | 1.44 | 1.90 | 1.90 | 1.34 |
| Qwen2.5 3B: RFT-QA | QA | QA-CoT | QA-Acc | List-CoT | 4.83 | 4.84 | 1.71 | 3.42 | 3.47 | 1.77 | 4.46 | 4.49 | 2.02 | 3.94 | 3.96 | 1.62 |
| Qwen2.5 3B: RFT-List (Acc) | QA | List-CoT | List-Acc | List-CoT | 795.83 | 796.48 | 3.96 | 814.50 | 815.25 | 5.45 | 777.20 | 777.20 | 5.05 | 843.40 | 843.40 | 3.43 |
| Qwen2.5 3B: RFT-List (MRR) | QA | List-CoT | List-MRR | List-CoT | 755.56 | 755.56 | 3.51 | 720.06 | 721.05 | 5.02 | 739.30 | 739.65 | 4.56 | 720.55 | 723.50 | 3.08 |
| Qwen3 4B: RFT-MCQ | MCQ | MCQ-CoT | MCQ-Acc | List-CoT | 3.75 | 3.76 | 1.30 | 2.92 | 3.03 | 1.52 | 3.70 | 3.71 | 1.62 | 3.22 | 3.23 | 1.33 |
| Qwen3 4B: RFT-QA | QA | QA-CoT | QA-Acc | List-CoT | 4.07 | 4.07 | 1.36 | 3.02 | 3.03 | 1.55 | 4.05 | 4.06 | 1.69 | 3.39 | 3.40 | 1.33 |
| Qwen3 4B: RFT-List (Acc) | QA | List-CoT | List-Acc | List-CoT | 808.06 | 808.06 | 7.09 | 786.94 | 790.93 | 8.35 | 787.74 | 787.74 | 10.58 | 818.02 | 818.02 | 8.56 |
| Qwen3 4B: RFT-List (MRR) | QA | List-CoT | List-MRR | List-CoT | 7.66 | 7.66 | 1.82 | 11.40 | 11.40 | 2.44 | 13.27 | 13.27 | 2.42 | 7.41 | 7.42 | 1.81 |
| OpenThinker3 7B: RFT-MCQ | MCQ | MCQ-CoT | MCQ-Acc | List-CoT | 2.92 | 2.97 | 1.38 | 2.13 | 2.15 | 1.25 | 2.98 | 3.02 | 1.58 | 2.30 | 2.31 | 1.29 |
| OpenThinker3 7B: RFT-QA | QA | QA-CoT | QA-Acc | List-CoT | 3.42 | 3.42 | 1.42 | 2.18 | 2.18 | 1.29 | 3.50 | 3.50 | 1.61 | 2.73 | 2.73 | 1.31 |
| OpenThinker3 7B: RFT-List (Acc) | QA | List-CoT | List-Acc | List-CoT | 820.24 | 821.58 | 5.22 | 829.59 | 831.88 | 6.45 | 816.43 | 817.99 | 7.20 | 849.27 | 852.75 | 4.95 |
| OpenThinker3 7B: RFT-List (MRR) | QA | List-CoT | List-MRR | List-CoT | 804.32 | 806.28 | 4.94 | 805.84 | 809.92 | 7.47 | 794.34 | 797.40 | 6.67 | 841.73 | 844.03 | 4.42 |
| m1 7B 23K: RFT-MCQ | MCQ | MCQ-CoT | MCQ-Acc | List-CoT | 2.76 | 2.79 | 1.28 | 2.23 | 2.23 | 1.36 | 2.63 | 2.63 | 1.37 | 2.36 | 2.37 | 1.21 |
| m1 7B 23K: RFT-QA | QA | QA-CoT | QA-Acc | List-CoT | 3.56 | 3.56 | 1.26 | 2.14 | 2.14 | 1.34 | 4.65 | 4.65 | 1.48 | 2.46 | 2.46 | 1.22 |
| m1 7B 23K: RFT-List (Acc) | QA | List-CoT | List-Acc | List-CoT | 757.42 | 758.04 | 6.90 | 808.49 | 808.86 | 7.02 | 738.56 | 740.69 | 8.68 | 785.32 | 785.32 | 7.01 |
| m1 7B 23K: RFT-List (MRR) | QA | List-CoT | List-MRR | List-CoT | 683.00 | 683.00 | 4.60 | 729.51 | 730.85 | 5.77 | 706.56 | 708.26 | 6.87 | 751.77 | 754.85 | 4.96 |
| AlphaMed 7B: RFT-MCQ | MCQ | MCQ-CoT | MCQ-Acc | List-CoT | 0.48 | 1.28 | 1.02 | 0.70 | 1.50 | 1.18 | 0.60 | 1.38 | 1.08 | 0.64 | 1.62 | 1.05 |
| AlphaMed 7B: RFT-QA | QA | QA-CoT | QA-Acc | List-CoT | 1.58 | 1.59 | 1.15 | 1.44 | 1.48 | 1.19 | 1.65 | 1.66 | 1.28 | 1.52 | 1.53 | 1.14 |
| AlphaMed 7B: RFT-List (Acc) | QA | List-CoT | List-Acc | List-CoT | 4.61 | 4.62 | 1.54 | 4.77 | 4.77 | 2.04 | 5.14 | 5.15 | 1.92 | 4.77 | 4.77 | 1.52 |
| AlphaMed 7B: RFT-List (MRR) | QA | List-CoT | List-MRR | List-CoT | 5.92 | 5.92 | 1.47 | 4.89 | 4.89 | 1.95 | 5.82 | 5.82 | 1.81 | 5.28 | 5.28 | 1.42 |
| RFT-List (List-CoT, MRR, LP=0) | QA | List-CoT | List-MRR-0 | List-CoT | 5.26 | 5.26 | 1.61 | 4.78 | 4.78 | 1.90 | 4.41 | 4.41 | 1.98 | 4.40 | 4.40 | 1.65 |
| RFT-List (List-CoT, MRR, LP=0.1) | QA | List-CoT | List-MRR-0.1 | List-CoT | 1.43 | 1.43 | 1.11 | 1.52 | 1.52 | 1.16 | 1.60 | 1.61 | 1.27 | 1.52 | 1.52 | 1.16 |
| RFT-List (List-CoT, MRR, LP=0.3) | QA | List-CoT | List-MRR-0.3 | List-CoT | 1.14 | 1.14 | 1.00 | 1.18 | 1.18 | 1.08 | 1.15 | 1.15 | 1.10 | 1.16 | 1.16 | 1.00 |
| RFT-List (List-CoT, MRR, LP=0.5) | QA | List-CoT | List-MRR-0.5 | List-CoT | 1.00 | 1.00 | 1.00 | 1.01 | 1.01 | 1.00 | 1.01 | 1.01 | 1.00 | 1.00 | 1.01 | 1.00 |
| RFT-List (List-CoT, MRR, LP=0.7) | QA | List-CoT | List-MRR-0.7 | List-CoT | 1.02 | 1.02 | 1.00 | 1.07 | 1.07 | 1.00 | 1.05 | 1.05 | 1.01 | 1.06 | 1.06 | 1.00 |
| RFT-List (List-CoT, MRR, LP=0.9) | QA | List-CoT | List-MRR-0.9 | List-CoT | 1.01 | 1.01 | 1.00 | 1.04 | 1.04 | 1.00 | 1.02 | 1.02 | 1.01 | 1.03 | 1.03 | 1.00 |
| AlphaMed 7B: RFT-List (Acc, LP=0.3) | QA | List-CoT | List-Acc-LP-0.3 | List-CoT | 1.24 | 1.24 | 1.08 | 1.34 | 1.34 | 1.12 | 1.32 | 1.32 | 1.18 | 1.35 | 1.35 | 1.09 |
| m1 7B 23K: RFT-List (Acc, LP=0.3) | QA | List-CoT | List-Acc-LP-0.3 | List-CoT | 1.14 | 1.14 | 1.04 | 1.28 | 1.28 | 1.09 | 1.23 | 1.24 | 1.14 | 1.24 | 1.24 | 1.08 |
| OpenThinker3 7B: RFT-List (Acc, LP=0.3) | QA | List-CoT | List-Acc-LP-0.3 | List-CoT | 1.37 | 1.37 | 1.16 | 1.38 | 1.38 | 1.20 | 1.42 | 1.43 | 1.20 | 1.46 | 1.46 | 1.18 |
| Qwen2.5 3B: RFT-List (Acc, LP=0.3) | QA | List-CoT | List-Acc-LP-0.3 | List-CoT | 1.31 | 1.31 | 1.09 | 1.29 | 1.31 | 1.12 | 1.40 | 1.41 | 1.20 | 1.36 | 1.36 | 1.12 |

The attending tells the resident that the patient will do fine, and there is no need to report this minor complication that will not harm the patient, as he does not want to make the patient worry unnecessarily. He tells the resident to leave this complication out of the operative report. Which of the following is the correct next action for the resident to take?" The available choices are omitted for brevity. The ground-truth answer is "B) Tell the attending that he cannot fail to disclose this mistake."

Although not all responses for the MCQ task exhibit this behavior, it occurs frequently enough to cause the model to miss points due to formatting errors. This illustrates how SFT with synthetic reasoning traces can lead the model to overfit to the specific training format, reducing its ability to follow other instructions. In contrast, the response generated by m1 for the same question, shown in Figure 29, follows the formatting instruction correctly.

---

## Thinking

Okay, so the resident accidentally cuts a flexor tendon during the carpal tunnel repair. That's definitely a mistake, but it's not causing any harm to the patient. The attending says it's fine and tells the resident not to mention it in the report. Hmm, that's a bit tricky. On one hand, the attending is trying to protect the patient from unnecessary worry, which is kind of nice. But on the other hand, isn't it important to be completely transparent with the patient about what happened during their surgery? I mean, the patient has a right to know everything that went on, even if it's a small complication.

Now, let's think about the ethical side of this. The resident is in a tough spot. The attending is suggesting they leave it out of the report, but that feels like it could be misleading. The resident has to weigh the importance of honesty with the patient against the attending's advice. It's a balancing act.

Oh, and there's the whole issue of the operative report. It's a legal document, and it's supposed to be comprehensive. If the resident doesn't report the mistake, they might be in trouble later on. It's not just about the patient's right to know; it's also about maintaining the integrity of the medical records.

So, what should the resident do? Well, the best course of action seems to be to disclose the error to the patient and include it in the operative report. That way, the patient is fully informed, and the report is accurate. It's a tough call, but it's the right thing to do.

Yeah, I'm pretty sure that's the right move. The resident should go with option A: disclose the error to the patient and put it in the operative report. That way, they're being both ethical and responsible.

## Final Response

A) Disclose the error to the patient and put it in the operative report

The resident should disclose the error to the patient and include it in the operative report. This action ensures that the patient is fully informed about what happened during their surgery, respecting their right to know all aspects of their medical care. Additionally, it maintains the integrity of the medical records, which is crucial for both ethical and legal reasons.

---

Figure 28: Generated response from HuatuoGPT-o1 ignore the final answer instruction.

### O.1  SFT MODELS

Figures 30 to 32 show examples of generated responses to the question: 'What viral disease is the only one to have been eradicated worldwide through vaccination efforts¿ The ground-truth answer is "smallpox." The responses are produced by SFT-MCQ, SFT-QA, and SFT-List under MCQ, QA, and List evaluations, respectively. Figure 33 shows a shortened response generated by SFT-MCQ during the List evaluation, containing repeated items within the generated list.

A.
The correct next action for the resident is to **A) Disclose the error to the patient and put it in the operative report**.

**Explanation:**
Cutting a flexor tendon during carpal tunnel repair is a significant complication, even if it is repaired without further issues. Ethically and legally, the resident has a duty to transparently communicate this error to the patient. Withholding such information could breach trust and potentially lead to legal consequences. Additionally, the operative report is a critical document for patient care and legal records; omitting the complication would be inaccurate and could harm future care if the patient is unaware. The resident should report the complication honestly, ensuring transparency while providing necessary support to the patient.

**Why Other Options Are Incorrect:**
– **B)** Telling the attending they cannot omit the mistake is unprofessional and could escalate the situation without addressing the core issue.
– **C)** Reporting the physician to the ethics committee is premature and could damage the learning environment or lead to unnecessary conflict without verifying facts.
– **D)** Refusing to dictate the report is ineffective, as the resident is still responsible for accurate documentation.

\boxed{A}

Figure 29: Generated response from m1 that follows the formatting instruction.

## O.2 RFT MODELS

Figures 34 to 38 show examples of generated responses from the RFT models for the same question: In the scenario where a surgical attending instructs a resident not to disclose an intraoperative complication to the patient or include it in the operative report, what is the most appropriate immediate action the resident should take?" The ground-truth answer is Tell the attending that he cannot fail to disclose this mistake." The figures correspond to responses from RFT-MCQ, RFT-QA, RFT-List-Acc, RFT-List-MRR, and RFT-List-Judge-MRR evaluated under their respective answer formats. Figure 39 shows an additional example from the RFT-QA model, which exhibits list-like behavior in its final answer. Figures 40 and 41 show examples of generated responses from RFT-List-Judge-MRR under MCQ and QA evaluations.

3024
3025
3026
3027
3028
3029
3030
3031
3032
3033
3034
3035
3036
3037
3038
3039
3040
3041
3042
3043
3044
3045
3046
3047
3048
3049
3050
3051
3052
3053
3054
3055
3056
3057
3058
3059
3060
3061
3062
3063
3064
3065
3066
3067
3068
3069
3070
3071
3072
3073
3074
3075
3076
3077

<think>
Okay, let's tackle this question. The user is asking about what viruses have encouraged us to change our world, with options from A to H. I need to recall historical and current medical achievements related to viruses.

First, I remember that viruses have been a major cause of diseases, so efforts to combat them have led to significant advancements. Let me go through each option one by one.

Option A: Developed a universal vaccine for all types of flu. I think there's no universal flu vaccine yet. Flu vaccines are usually updated annually because the virus changes, so a universal one isn't available. So A is probably not correct.

Option B: Eradicated polio. Polio was eradicated globally through vaccination efforts, right? The Global Polio Eradication Initiative led by the WHO and others has been successful, and polio is now almost eradicated. So B might be a candidate.

Option C: Created a vaccine for the common cold. The common cold is caused by many different viruses, and there's no vaccine for it. There are some experimental ones, but none are widely available or effective. So C is likely incorrect.

Option D: Found a cure for all types of cancer. Cancer is a complex disease, and while there are treatments, there's no universal cure. Vaccines exist for some cancers (like HPV for cervical cancer), but not for all types. So D is probably not right.

Option E: Eradicated smallpox. Smallpox was eradicated in 1980, thanks to the smallpox vaccine. That's a huge achievement. So E could be correct.

Option F: Completely stopped the spread of all infectious diseases. That's impossible because new diseases emerge, and some are always spreading. So F is definitely wrong.

Option G: Developed immunity to all types of viruses. Immunity is specific to each virus, and we can't develop immunity to all viruses. Vaccines target specific pathogens, so G is incorrect.

Option H: Eradicated HIV/AIDS. HIV/AIDS is still a major global health issue, and while there's progress, it's not eradicated. There's no cure, and it's not eradicated yet. So H is not correct.

Now, between B and E. Polio eradication is ongoing, but it's been declared eradicated in many regions. Smallpox was eradicated globally in 1980. The question says "eradicated," which means completely eliminated from the planet. Smallpox is the only human disease eradicated by vaccination. Polio is close but not fully eradicated yet. The WHO has declared it eradicated in the wild, but there are still cases in a few countries. However, smallpox was eradicated in 1980, which is a significant milestone.

Wait, the question says "eradicated polio." If polio is still not fully eradicated, then B might not be correct. But the question might be considering the fact that polio is on the verge of eradication. However, the standard answer for such questions usually points to smallpox as the one that was eradicated. Let me double−check.

Smallpox was eradicated in 1980, and polio is still being eradicated. The Global Polio Eradication Initiative is working towards that goal, but it's not yet complete. So the correct answer should be E, eradicated smallpox.

Wait, but the question says "eradicated polio" as option B. If polio isn't fully eradicated, then B is incorrect. But the question might be using "eradicated" loosely, but in medical terms, eradication means global elimination. Smallpox is the only one that's been eradicated. So E is correct.

Let me confirm: Smallpox was eradicated in 1980, polio is still not eradicated. So the answer should be E.

But let's make sure about the other options. Option B says "eradicated polio," but if it's not fully eradicated, then B is wrong. The question might be tricking us. However, in standard multiple−choice questions, smallpox eradication is a key point. Also, the other options are clearly wrong. For example, there's no universal flu vaccine, no cure for all cancer, no immunity to all viruses, etc.

So the correct answer is E. Eradicated smallpox.
</think>

The question asks what viruses have encouraged us to change our world, with the options provided. Let's evaluate each choice based on historical and current medical achievements:

− **Option A (Developed a universal vaccine for all types of flu)**: Incorrect. Influenza vaccines are updated annually due to viral mutations, and a universal vaccine for all flu strains does not exist.
− **Option B (Eradicated polio)**: Incorrect. While polio eradication efforts have been successful in many regions, polio is **not yet fully eradicated** globally. The Global Polio Eradication Initiative continues to work toward this goal, but it is not yet complete.
− **Option C (Created a vaccine for the common cold)**: Incorrect. The common cold is caused by multiple viruses (e.g., rhinoviruses), and no effective vaccine exists for it.
− **Option D (Found a cure for all types of cancer)**: Incorrect. Cancer is a complex disease with no universal cure. While targeted treatments exist for specific cancers, a cure for all types does not exist.
− **Option E (Eradicated smallpox)**: **Correct**. Smallpox was the **only human disease eradicated by vaccination**. The World Health Organization (WHO) declared smallpox eradicated in 1980, following a global vaccination campaign. This is a landmark achievement in public health.
− **Option F (Completely stopped the spread of all infectious diseases)**: Incorrect. New infectious diseases emerge (e.g., SARS−CoV−2), and diseases like HIV/AIDS persist, so this is impossible.
− **Option G (Developed immunity to all types of viruses)**: Incorrect. Immunity is specific to each virus; vaccines target particular pathogens, not all viruses.
− **Option H (Eradicated HIV/AIDS)**: Incorrect. HIV/AIDS remains a major global health challenge with no cure, though antiretroviral therapy has improved outcomes.

The key historical milestone is the eradication of smallpox, which is a direct result of viral disease control efforts. While polio eradication is ongoing, smallpox is the only disease eradicated by vaccination.

$\boxed{E}$

Figure 30: Generated response from SFT-MCQ on MCQ evaluation.

<think>
Okay, the user is asking about a viral disease that's been eradicated worldwide through vaccination efforts. Let me recall what I know about global health achievements.

Hmm, I remember that smallpox was the only disease eradicated through vaccination. The smallpox vaccine was developed by Edward Jenner in the 18th century, and after decades of global vaccination campaigns, the World Health Organization declared smallpox eradicated in 1980.

Wait, let me double−−check to make sure I'm not confusing it with other diseases. Polio is close−− there's been significant progress with the vaccine, but it's not fully eradicated yet. Measles and rubella are controlled but not eradicated. Rabies can be prevented with vaccines, but it's not eradicated globally.

The key point here is "eradicated worldwide," meaning no cases anywhere on Earth. Smallpox fits that description perfectly. The WHO certified it as eradicated in 1980, and since then, no natural cases have been reported.

I should confirm if there are any other candidates. Bacterial diseases like cholera or tuberculosis have been controlled with vaccines, but they're not eradicated. So smallpox is definitely the answer.

The user seems to want a straightforward, factual answer−−probably for educational purposes. They specified the reasoning process in tags, so I'll make sure to follow that format exactly. No need for extra fluff; just the facts.

Final check: Yes, smallpox is the only viral disease eradicated through vaccination. The answer is clear−cut.
</think>

The question asks for the viral disease eradicated worldwide through vaccination efforts. To determine this, I recall historical and epidemiological facts about global disease eradication.

− Smallpox (caused by *Variola major* virus) was the only viral disease eradicated globally via vaccination. The smallpox vaccine, developed by Edward Jenner in 1796, was pivotal in the eradication campaign.
− The World Health Organization (WHO) launched a global eradication initiative in 1967, and after extensive vaccination efforts, the last naturally occurring case was reported in Somalia in 1977. The WHO declared smallpox eradicated in 1980.
− Other viral diseases like polio, measles, and rubella are controlled but not yet eradicated (polio remains endemic in a few countries, and measles has seen recent outbreaks despite vaccination). Bacterial diseases (e.g., cholera, tuberculosis) are not viral and thus irrelevant here.

No other viral disease has achieved complete global eradication through vaccination. Therefore, smallpox is the correct answer.

$\boxed{\text{smallpox}}$

Figure 31: Generated response from SFT-QA on QA evaluation.

<think>
Okay, the user is asking about a viral disease that has been eradicated worldwide through vaccination efforts. Let me recall what I know about disease eradication.

First, I remember that smallpox was the first and only disease eradicated globally. The World Health Organization (WHO) declared smallpox eradicated in 1980 after a massive vaccination campaign.

I should double−check if there are any other diseases that might have been eradicated. Polio is close to eradication but hasn't been officially declared eradicated yet. Measles and rubella are controlled but not eradicated. Rabies can be controlled with vaccination but isn't eradicated globally.

So, smallpox is the correct answer. The user might be a student or someone interested in public health, so accuracy is important. They probably need this for a test or to understand historical achievements in medicine.

I should confirm that smallpox is indeed the only one. Yes, WHO states that smallpox is the only human disease eradicated through vaccination. The answer should be straightforward.
</think>

# Final Answer
1. Smallpox

Figure 32: Generated response from SFT-List on List evaluation.

# Final Answer
1. Tryptophan
2. Histidine
3. Methionine
4. Lysine
5. Leucine
6. Isoleucine
7. Valine
8. Threonine
9. Phenylalanine
10. Arginine
11. Alanine
12. Glycine
13. Proline
14. Serine
15. Cysteine
16. Tyrosine
17. Aspartic acid
18. Glutamic acid
19. Asparagine
20. Glutamine
21. Methionine
22. Cysteine
23. Tyrosine
// ...
// Omitted for breity
// ...
1045. Cysteine
1046. Tyrosine
1047. Phenylalan

Figure 33: Generated response from SFT-MCQ on List evaluation with repeating content.

<think>
The resident is in a situation where there is a minor complication that occurred during the surgery, specifically the inadvertent cut to a flexor tendon. The attending physician is suggesting that this complication be omitted from the operative report and not disclosed to the patient. However, it is important to adhere to ethical standards and patient safety protocols. Minor complications should still be documented and disclosed to ensure transparency and proper patient care.

The resident has a duty to report the complication to the patient and include it in the operative report, even if the attending physician advises against it. This ensures that the patient is informed and that the medical record is accurate.

Options:
A) Disclose the error to the patient and put it in the operative report – This is the correct action as it adheres to ethical standards and patient rights.
B) Tell the attending that he cannot fail to disclose this mistake – This is also correct but does not address the reporting to the patient.
C) Report the physician to the ethics committee – This is an extreme measure and not necessary in this situation.
D) Refuse to dictate the operative report – This is not the best approach as it does not address the patient's right to know and the need for accurate documentation.

The correct action is to disclose the error to the patient and include it in the operative report.
</think>
\boxed{A}

Figure 34: Generated response from RFT-MCQ on MCQ evaluation.

<think>
The scenario presents a ethical dilemma involving patient safety, transparency, and legal compliance. The resident is instructed not to disclose an intraoperative complication to the patient or include it in the operative report. This instruction is unethical and potentially illegal, as patients have a right to know about complications that occur during surgery, and failure to report complications in the operative record can lead to malpractice lawsuits and harm to patient care in the future.

The resident should not follow this instruction and should take immediate action to ensure patient safety and compliance with medical ethics and legal standards. The resident should:
1. Document the complication in the patient's medical record, ensuring it is included in the operative report.
2. Inform the patient about the complication as soon as it is safe to do so, respecting the patient's right to know and the principle of informed consent.
3. Report the incident to a supervisor or the hospital's risk management department to address the ethical and legal concerns.
4. Consider seeking advice from a legal or ethical consultant within the hospital to ensure compliance with all relevant standards and laws.
</think>

The most appropriate immediate action the resident should take is to \boxed{document the complication in the operative report and patient's medical record, inform the patient about the complication as soon as it is safe to do so, report the incident to a supervisor or the hospital's risk management department, and seek advice from a legal or ethical consultant if necessary}.

Figure 35: Generated response from RFT-QA on QA evaluation.

<think>In this scenario, the resident is faced with a directive that conflicts with ethical standards and legal requirements for patient safety and transparency. The resident should not follow the attending's instruction to withhold information from the patient or omit it from the operative report. The appropriate actions include reporting the situation to a supervisor, the department head, or the institution's ethics committee. The resident should also document the incident in their own notes and consider seeking advice from a more senior colleague or legal counsel.</think>

# Final Answer
1. Report the incident to a senior surgeon
2. Document the incident in the resident's notes
3. Seek advice from a more senior colleague
4. Report the incident to the department head
5. Report the incident to the institution's ethics committee
6. Follow the attending's instruction

Figure 36: Generated response from RFT-List-Acc on List evaluation.

<think>The resident is faced with a ethical dilemma involving patient safety and transparency. The most appropriate action would be to report the complication to a supervisor or the medical ethics committee to ensure patient safety and maintain proper documentation. The resident should not follow the attending's instruction to withhold information from the patient or omit it from the report.</think>

# Final Answer
1. Report the complication to a senior surgeon
2. Report the complication to the medical ethics committee
3. Follow the attending's instructions
4. Disclose the complication to the patient directly
5. Document the complication in the patient's chart without reporting it

Figure 37: Generated response from RFT-List-MRR on List evaluation.

<think>Considering the ethical and legal responsibilities of a medical professional, the resident should prioritize patient safety and transparency. The options should include reporting the issue to a supervisor, the medical director, or a hospital ethics committee if the attending's instructions are to withhold information from the patient and not document it. The most appropriate action is to report the attending's instructions to a higher authority or ethics committee to ensure patient safety and compliance with medical ethics.</think>

# Final Answer
1. Report the attending's instructions to a hospital ethics committee
2. Report the attending's instructions to the medical director
3. Follow the attending's instructions without reporting them
4. Discuss the situation with a senior colleague or another attending physician
5. Document the complication in the operative report and inform the patient without informing the attending

Figure 38: Generated response from RFT-List-Judge-MRR on List evaluation.

<think>
Copper IUDs are generally considered safe and effective for most women. However, there are certain medical conditions that make their use contraindicated. These include:
1. Pelvic inflammatory disease (PID) or a history of PID, as the IUD can increase the risk of infection.
2. Active genital or pelvic infection at the time of insertion.
3. Uterine or cervical cancer.
4. Known or suspected pregnancy.
5. Severe liver disease.
6. Allergy to copper or other IUD components.
7. Certain sexually transmitted infections (STIs) or risk factors for STIs.
8. Previous uterine surgery that has altered the uterine cavity.
9. Adhesions from previous pelvic surgery.
10. Severe endometriosis or adenomyosis, as the IUD can exacerbate symptoms.
11. Copper IUD is contraindicated in women with Wilson's disease, as copper toxicity can be a concern.
12. Thrombophlebitis or thrombosis, as the IUD can increase the risk of thromboembolic events.
13. Active genital herpes or other sexually transmitted infections that could be exacerbated by the IUD.
Given the question, we are specifically looking for past medical history that would make copper IUD placement contraindicated. This would include a history of pelvic inflammatory disease, previous pelvic or uterine surgery, endometriosis, adenomyosis, or Wilson's disease.
</think>

The past medical history that would make copper IUD placement contraindicated in a woman requesting this form of contraception includes \boxed{pelvic inflammatory disease, previous pelvic or uterine surgery, endometriosis, adenomyosis, Wilson's disease, or a history of severe liver disease, active genital or pelvic infection, or known or suspected pregnancy at the time of consultation}.

Figure 39: Generated response from RFT-QA for the question: "What past medical history would make copper intrauterine device (IUD) placement contraindicated in a woman requesting this form of contraception?" The ground-truth answer is "Active or recurrent pelvic inflammatory disease (PID)."

<think> The resident has a duty to ensure that the patient is fully informed about the procedures and any complications that occur during surgery. Additionally, accurate and complete documentation is crucial for medical records and patient care. The attending's suggestion to omit the complication from the operative report and not inform the patient is unethical and could lead to potential harm if the patient is unaware of the issue. The resident has a professional and ethical obligation to report the complication and include it in the operative report.</think>

The correct next action for the resident to take is to disclose the error to the patient and put it in the operative report.

\boxed{A}

Figure 40: Generated response from RFT-List-Judge-MRR on MCQ evaluation.

<think>Considering the ethical and legal implications, the resident should first question the attending to understand the reasoning behind the instruction. However, if the attending insists on not disclosing the complication to the patient and not including it in the operative report, the resident should report this to a more senior physician or the hospital's ethics committee to ensure patient safety and compliance with medical standards. Directly following the attending's instructions without question could potentially compromise patient care and legal standards.</think>

The most appropriate immediate action the resident should take is to report the situation to a more senior physician or the hospital's ethics committee.

\boxed{Report the situation to a more senior physician or the hospital's ethics committee}

Figure 41: Generated response from RFT-List-Judge-MRR on QA evaluation.

