# OpenReview forum: "Single Answer is Not Enough: On Generating Ranked Lists with Medical Reasoning Models"
_ICLR.cc/2026/Conference — ICLR 2026 Conference Withdrawn Submission_

### Official Review · Reviewer_2LD3 · 2025-10-22

**Soundness:** 2
**Presentation:** 3
**Contribution:** 2
**Rating:** 4
**Confidence:** 5

**Summary:**

This paper addresses the alignment problem between the multiple choice question format and real world medical complexity where a single answer may not be enough to accurately represent the best next steps and uncertainty due to incomplete information.

The authors propose to transform existing MCQ evaluations into a ranked list and open QA format to compensate the limitations of the MCQ format. Using these conversions they perform multiple performance comparison and proceed to use different techniques including prompt engineering, SFT and RFT to study the impact of different methods to adapt existing models to new formats.

**Strengths:**

Moving from MCQ to more representative methods is well motivated. In addition, the authors provide a comparison of proprietary models, general open models and medical finetuned models. The three methods used to train models provide an interesting comparison of these techniques to adapt existing models and show how RFT generalizes better to new formats.

Overall the paper is well written and easy to follow with a clear graphical abstract that gives a good overview of the contribution and the structure of the paper.

**Weaknesses:**

While the motivation to move beyond standard accuracy on single choice MCQ is detailed and sound, the authors do not discuss prior work that introduced different testing methodologies (e.g. AI Hospital, MetaMedQA, MAI-DxO, HealthBench) [1-4]. The authors do not sufficiently position their work compared to other approaches to justify the use of ranked lists and possibly explain how ranked lists could be integrated in non MCQ formats.

The reliance on existing benchmarks such as MedQA and MedMCQA imports the known limitations of board style vignettes which mostly test pattern recognition with shortcuts [4, 5]. In addition, the authors do not discuss the alignment between board style vignettes and real-world clinical work, for instance, the absence of longitudinal and noisy data severely limits their relevance to real world data. As the authors explain in the limitations, the source datasets are single answer which limits the use of ranked lists where multiple options can be correct and their relative priority informs us on the understanding of appropriate next steps or diagnosis. Finally, these QA samples are not representative of medicine as a whole and priorities may shift depending on the context, for instance, in a high resource setting for a patient with a suspected STEMI we direct the patient to a cath lab immediately but in lower resource settings we would consider thrombolysis as a first step. A ranked list would fail to accurately represent two correct answers depending on context, these limitations are not discussed.

The ablation studies use relatively small models due to compute availability which limits generalizability of the results. A single experiment with a large model would help in demonstrating the scalability of this method.

[1] AI Hospital: Benchmarking Large Language Models in a Multi-agent Medical Interaction Simulator (Fan et al., COLING 2025)

[2] Large Language Models lack essential metacognition for reliable medical reasoning (Griot et al., Nature Communications 2025)

[3] Sequential Diagnosis with Language Models (Nori et al. Preprint 2025)

[4] HealthBench: Evaluating Large Language Models Towards Improved Human Health (Arora et al. Preprint 2025)

[5] Pattern Recognition or Medical Knowledge? The Problem with Multiple-Choice Questions in Medicine (Griot et al., ACL 2025)

[6] Language Models are Surprisingly Fragile to Drug Names in Biomedical Benchmarks (Gallifant et al., Findings 2024)

**Questions:**

Suggestions:

1) Improving the related work introduction to better position this paper in the boarder realm of medical evaluations for LLMs would strengthen the motivation for this particular approach.

2) A small (n = 50-100) benchmark designed by experts with multiple ranked options as a test evaluation would strengthen the findings and support the proposed methods.

3) Discussing existing limitations of the source datasets and how the authors compensate for these limitations to support their conclusions would improve the validity of the experiments.

4) Discussing the alignment problem between automated testing and real-world applications is necessary to position the work as the goal of application research is to translate to real-world benefits. For instance, training models to use a ranked list format is not motivated in relation to the applicability of these lists in real-world settings. When in clinical workflows would this be useful?

5) Scaling the ablation to a larger model would add credibility to the approach to specialize models to new formats.

---

> ### Author Response · Authors · 2025-11-21
>
> Thank you for your careful review and constructive feedback. We appreciate your recognition of the motivation behind moving beyond MCQ-style evaluation, as well as your positive comments on our comparison across model types and training strategies. We are also glad that you found the paper clearly written and the graphical abstract helpful in framing our contributions.
>
> > While the motivation to move beyond standard accuracy on single choice MCQ is detailed and sound, the authors do not discuss prior work that introduced different testing methodologies (e.g. AI Hospital, MetaMedQA, MAI-DxO, HealthBench) [1-4]. The authors do not sufficiently position their work compared to other approaches to justify the use of ranked lists and possibly explain how ranked lists could be integrated in non MCQ formats. Improving the related work introduction to better position this paper in the boarder realm of medical evaluations for LLMs would strengthen the motivation for this particular approach.
>
> - Thank you for your suggestions. We have added a paragraph in the Introduction and Appendix A.4 Related Work to better position our paper and to clarify how our work differs from research focused specifically on differential diagnosis. We have also added future directions in the Limitations section, including broader evaluation settings such as multi-turn interactions. Below, we provide a comparison between our work and the references you suggested:
> - **[1] AI Hospital: Benchmarking Large Language Models in a Multi-agent Medical Interaction Simulator (Fan et al., COLING 2025)**
> - This paper proposes a new scenario for evaluating LLMs by testing whether they can simulate a doctor in a multi-agent, multi-turn environment. In contrast, our work focuses on medical reasoning models rather than medical LLMs and evaluates them in single-turn, task-specific benchmark settings.
> - **[2] Large Language Models Lack Essential Metacognition for Reliable Medical Reasoning (Griot et al., Nature Communications 2025)**
> - This paper introduces MetaMedQA, a benchmark designed to assess metacognitive behaviors in medical contexts. In contrast, one of our contributions is a new evaluation scheme centered on answer formats—specified through instructions in the prompt—across a broad set of MCQ and open-ended benchmarks for MRMs.
> - **[3] Sequential Diagnosis with Language Models (Nori et al., Preprint 2025)**
> - This work explores iterative medical reasoning, where a model requests additional details and updates its diagnostic conclusions. It focuses on multi-turn interactions, whereas our study is scoped to single-turn, accuracy-focused benchmarks for MRMs.
> - **[4] HealthBench: Evaluating Large Language Models Toward Improved Human Health (Arora et al., Preprint 2025)**
> - This paper presents a multi-turn evaluation framework using LLM judges with rubric-based scoring. In contrast, our focus is on single-turn MRM evaluation. Although both works employ LLM judges, our evaluation relies on a reference-based approach that compares generated answers to ground truth, whereas HealthBench uses rubric-based criteria.
> - We hope these clarifications highlight the distinctions between our work and prior research.

---

> ### Author Response · Authors · 2025-11-21
>
> > The reliance on existing benchmarks such as MedQA and MedMCQA imports the known limitations of board style vignettes which mostly test pattern recognition with shortcuts [4, 5]. In addition, the authors do not discuss the alignment between board style vignettes and real-world clinical work, for instance, the absence of longitudinal and noisy data severely limits their relevance to real world data. As the authors explain in the limitations, the source datasets are single answer which limits the use of ranked lists where multiple options can be correct and their relative priority informs us on the understanding of appropriate next steps or diagnosis. Finally, these QA samples are not representative of medicine as a whole and priorities may shift depending on the context, for instance, in a high resource setting for a patient with a suspected STEMI we direct the patient to a cath lab immediately but in lower resource settings we would consider thrombolysis as a first step. A ranked list would fail to accurately represent two correct answers depending on context, these limitations are not discussed. A small (n = 50-100) benchmark designed by experts with multiple ranked options as a test evaluation would strengthen the findings and support the proposed methods. Discussing existing limitations of the source datasets and how the authors compensate for these limitations to support their conclusions would improve the validity of the experiments.
> - We agree that models are prone to memorization of MCQ benchmarks [5, 6]. This is one reason we evaluate models across a broader range of answer formats—especially ranked lists, which models are less likely to have encountered during training.
> - As you noted, we discuss this limitation in our Limitations section, and we also highlight throughout the paper that current benchmarks typically contain only a single valid answer. This motivated the case study in Section 5, where we explore what would be required to construct a benchmark with multiple valid answers and what kinds of insights such a benchmark could provide. By simulating multiple valid answers, we illustrate the potential benefits such an evaluation setting could bring. While this simulation is not a full representation of the complexity of real-world medicine—just as no single benchmark can be—we believe it offers a glimpse into important aspects we care about, such as how robust an MRM is across answer formats. Our hope is that this work raises awareness of the need for more representative benchmarks that better reflect real clinical requirements, including those with multiple valid answers.
> - We added a discussion on the implications of our settings and outlined future directions for context-aware ranked-list generation, such as adjusting treatments based on additional conditions, in the newly added Section 4.4, Discussions.
> - However, developing such benchmarks is nontrivial even at a small scale. Constructing high-quality datasets that require professional medical expertise is costly and involves multiple stages of design, quality control, and validation. This process is beyond the scope of our present study.
>
> > The ablation studies use relatively small models due to compute availability which limits generalizability of the results. A single experiment with a large model would help in demonstrating the scalability of this method. Scaling the ablation to a larger model would add credibility to the approach to specialize models to new formats.
>
> - We note that we examined the generalization of our setup using five different initial models of varying sizes—3B, 4B, and 7B—in Section 4.3.2. However, we recognize that the insights drawn from our setup may not fully generalize to larger-scale models due to limitations in compute resources and the scope of this initial systematic study. We leave this for future work. We have revised the corresponding section in the Limitations to explicitly clarify this point: “Given our limited compute budget, these results are informative in our setting but may not fully generalize to larger-scale RFT training, e.g., larger models, longer training, and larger datasets; however, we expect several findings to hold.”

---

> ### Author Response · Authors · 2025-11-21
>
> > Discussing the alignment problem between automated testing and real-world applications is necessary to position the work as the goal of application research is to translate to real-world benefits. For instance, training models to use a ranked list format is not motivated in relation to the applicability of these lists in real-world settings. When in clinical workflows would this be useful?
>
> - We agree that the automated testing used in our study is necessarily limited in scope, and that real-world clinical settings are far more complex and nuanced. To address this, we added Section 4.4 Discussions, where we further examine the implications of our work, discuss how ranked-list reasoning may translate to real-world workflows, and highlight relevant safety considerations and mitigation strategies.
> - We include differential diagnosis as one example where a ranked list may be useful. Other scenarios where ranked-list reasoning can provide meaningful clinical value include: differential diagnosis, listing plausible etiologies in cases of incomplete or ambiguous presentations, and triage decision support.
>
> We hope these answers clarify your concerns. If anything remains unclear or you have additional questions, we would be happy to discuss them further.

---

> > ### Comment · Reviewer_2LD3 · 2025-11-21
> >
> > Dear Authors,
> >
> > Thank you for your detailed response.
> >
> > I appreciate the improved related work section, which better frames the contribution.
> >
> > However, the limitations of relevance, quality issues due to the source material (including potential data contamination), and the absence of expert participation remain fundamental issues that persist even when acknowledged. As this is a practical study applied to the clinical domain, I am reviewing it as a clinician and do not see the value for the clinical world in this current form.
> >
> > For this new task format, original data and questions designed at least in part by experts are required. While the limits of MCQs are well documented and new methods are required, the justification for this specific task lacks real-world evidence.

---

### Official Review · Reviewer_XJqA · 2025-10-27

**Soundness:** 2
**Presentation:** 2
**Contribution:** 1
**Rating:** 2
**Confidence:** 4

**Summary:**

The authors highlight concerns about the tendency of medical reasoning models to produce a single final answer even in open-ended settings. They explore prompting and fine-tuning methods in order to produce ranked lists of outputs, including reinforcement fine-tuning and supervised fine-tuning across several models.

**Strengths:**

There are a few strengths of this paper. The issue of overly narrow benchmarks failing to consider the full breadth of clinical responses in evaluation is a valid one (and is in line with broader concerns in the field about the limitations of MCQs). Better and more nuanced assessment of the quality of differential diagnoses and clinical reasoning is important.

The authors apply appropriate technical rigor to the implementation of methods of supervised and reinforcement fine-tuning. They also engage with an interesting array of medical-specific models, even if frontier models are absent.

**Weaknesses:**

Speaking frankly, it is not entirely clear to me that the authors substantiate the concern underlying this paper. Both in my personal experience with these models as well as in the literature I am familiar with (e.g. https://arxiv.org/abs/2412.10849 and others), it is relatively trivial to instruct any of the state-of-the-art language models to produce a clearly ranked differential diagnosis list (e.g., multiple outputs). While it is true that overconfidence in models can be a problem (e.g. https://ai.nejm.org/doi/pdf/10.1056/AIdbp2500120), it remains much more complex than simply a limitation in the length of outputs.

Perhaps these problems are more substantial for specific medical reasoning models like some of those models that the authors use (I have rarely seen these evaluated or used), but this does not appear to be an issue for the main frontier models. Unfortunately, the authors only include extensive evaluation of these various undersized models, and extrapolate conclusions more generally to LLMs as a class based on them. This is a severe limitation of this work, and although it can be cost-prohibitive, this work really should be performed at the frontier.

Further, it is not clear that the ranked list generation methods that the authors employ actually map cleanly onto any of the aspects of list ranking that are valued by clinicians in reality. For example, a robust differential diagnosis should also include "rare but dangerous" considerations, or "rare but treatable" considerations in order that they are fully considered by the clinicians at hand, even if these would not technically maximize score based on rank list structure (see discussions of this in e.g. https://pmc.ncbi.nlm.nih.gov/articles/PMC3270234/ and https://www.nature.com/articles/s41586-025-08869-4/). Ultimately, the assessment of ranked lists is an important and nuanced question, that depends on the specifics of the clinical problem at hand. Basic probabilistic metrics, or knowledge-based metrics like their expanded medQA are insufficient.

The use of LLM-as-judge is inappropriate without at least some degree of human validation by physicians. Medical answers and diagnoses are very complex and nuanced, and it cannot simply be assumed that the semantic equivalence matching is high-quality. While it is reasonable to use a validated LLM-as-judge as an extension, you must use core evaluation methods for any specific implementation. The entire section evaluating different judge models is incoherent in this context. You cannot state that "to change the judge model significantly impacts performance", because the underlying performance of the systems are the same. What has changed is the performance onf the judge model, which the authors have not appropriately validated.

**Questions:**

1. Are these concerns about the brevity of model answers substantiated in frontier models?
2. How do these concerns about answer length manifest across the range of various clinical tasks, beyond multiple choice? Does this apply to differential diagnosis alone? What of clinical decisionmaking in other realms?

---

> ### Author Response · Authors · 2025-11-21
>
> Thank you for your thoughtful evaluation of our work and for highlighting both its strengths and areas for improvement. We appreciate your recognition of the importance of moving beyond narrowly defined benchmarks in clinical evaluation, as well as your acknowledgement of the technical rigor in our implementation of supervised and reinforcement fine-tuning across medical-specific models.
>
> > Speaking frankly, it is not entirely clear to me that the authors substantiate the concern underlying this paper. Both in my personal experience with these models as well as in the literature I am familiar with (e.g. https://arxiv.org/abs/2412.10849 and others), it is relatively trivial to instruct any of the state-of-the-art language models to produce a clearly ranked differential diagnosis list (e.g., multiple outputs). While it is true that overconfidence in models can be a problem (e.g. https://ai.nejm.org/doi/pdf/10.1056/AIdbp2500120), it remains much more complex than simply a limitation in the length of outputs. Perhaps these problems are more substantial for specific medical reasoning models like some of those models that the authors use (I have rarely seen these evaluated or used), but this does not appear to be an issue for the main frontier models. Unfortunately, the authors only include extensive evaluation of these various undersized models, and extrapolate conclusions more generally to LLMs as a class based on them. This is a severe limitation of this work, and although it can be cost-prohibitive, this work really should be performed at the frontier. Are these concerns about the brevity of model answers substantiated in frontier models?
>
> - While frontier models can indeed be instructed to produce various output formats, and studying training recipes at that scale would be valuable given the lack of existing research, doing so would require an enormous amount of compute that we do not have available. Moreover, the objective would differ from that of our study. Therefore, this falls outside the scope of the present work. Our work focuses on examining the robustness of answer formats in medical reasoning models (MRMs), with particular attention to ranked-list outputs and approaches for enabling MRMs to produce such formats reliably.
> - While the effectiveness of prompting has been well studied in medical LLMs, much less is known about how effectively MRMs—or reasoning models more broadly—follow instructions. As we note in the paper, several recent studies (Li et al., 2025a; Fu et al., 2025; Jang et al., 2025) have reported degraded instruction-following ability in such models. This motivated us to investigate this effect directly. Our broader goal is to understand robustness to answer format—particularly the ranked-list format—and to study how MRMs can be enabled to reliably produce such outputs through prompting and training.
> - We focus on 7B-size models because our analysis in Section 3 compares MRMs developed across multiple studies. Although these MRMs were trained with different methodologies, they share the same backbone—Qwen2.5-7B-Instruct—which reduces confounding factors and allows clearer comparison of their behaviors. In our prompting experiments, we also evaluate larger and frontier models. However, due to compute constraints, subsequent training experiments focus on 7B models or smaller.
> - Finally, we clarify that our work does not claim any limitations related to output length, nor do we express concerns about response brevity.

---

> ### Author Response · Authors · 2025-11-21
>
> > Further, it is not clear that the ranked list generation methods that the authors employ actually map cleanly onto any of the aspects of list ranking that are valued by clinicians in reality. For example, a robust differential diagnosis should also include "rare but dangerous" considerations, or "rare but treatable" considerations in order that they are fully considered by the clinicians at hand, even if these would not technically maximize score based on rank list structure (see discussions of this in e.g. https://pmc.ncbi.nlm.nih.gov/articles/PMC3270234/ and https://www.nature.com/articles/s41586-025-08869-4/). Ultimately, the assessment of ranked lists is an important and nuanced question, that depends on the specifics of the clinical problem at hand. Basic probabilistic metrics, or knowledge-based metrics like their expanded medQA are insufficient.
>
> - Thank you for raising this point. We have added a discussion of item-level considerations in the newly included Section 4.4 Discussions, where we address this limitation and outline potential mitigation strategies—for example, incorporating more robust reward functions that explicitly emphasize this aspect. Because our work represents the first systematic study of ranked-list medical reasoning models, we intentionally kept the reward function simple to avoid introducing confounding factors into our analysis. Although this issue is important, fully addressing it falls outside the scope of the present study.
> - As noted throughout the paper, the lack of available benchmarks with multiple valid answers constrains our experimental setting. To partially address this, we include a simulated multiple–valid-answer benchmark in Section 5 to explore a case study where ranked-list reasoning may better capture clinical ambiguity. We also clarify that our choice of metrics is driven by the structure of existing benchmarks, which contain only a single correct answer. For this reason, we use accuracy and MRR—standard rank-aware metrics for settings with a single ground-truth label—as our primary evaluation measures.
>
> > The use of LLM-as-judge is inappropriate without at least some degree of human validation by physicians. Medical answers and diagnoses are very complex and nuanced, and it cannot simply be assumed that the semantic equivalence matching is high-quality. While it is reasonable to use a validated LLM-as-judge as an extension, you must use core evaluation methods for any specific implementation. The entire section evaluating different judge models is incoherent in this context. You cannot state that "to change the judge model significantly impacts performance", because the underlying performance of the systems are the same. What has changed is the performance onf the judge model, which the authors have not appropriately validated.
>
> - We would like to clarify that judge models are used in two distinct ways in our study: (1) for evaluation and (2) within RFT. In both settings, the judge serves as a comparator between the ground-truth answer and the model-generated answer, determining whether the two are equivalent in a binary manner.
> - The use of language-model judges for both evaluation and training is well established in the literature—for example, in evaluation frameworks such as HealthBench and in training pipelines such as DeepSeek R1. Comparing generated text against ground truth is a common and well-supported approach, particularly in reference-based evaluation settings, where LLM-based judges have been shown to provide higher accuracy (see https://arxiv.org/abs/2412.05579).
> - In our study, each question—both during training and evaluation—has a single valid answer, reflecting the limitations of currently available benchmarks. While MCQ settings allow correctness to be checked via option selection, open-ended formats are more challenging. For these open-ended tasks, we report exact match scores (following Med-U1) alongside LLM-judge metrics. The latter captures semantic equivalence that may not appear in exact string matches, such as synonymy or phrasing differences.
> - LLM judges also offer a practical advantage: they are considerably more cost-effective than human evaluators and allow evaluation at scale, which is essential during RFT where hundreds of thousands of generated responses must be assessed efficiently.
> - Regarding the performance improvement observed when changing the judge model in our RFT ablation study, we agree that the improvement arises from using a stronger judge model. This is precisely the intent of the ablation—to isolate the effect of reward-model strength on downstream performance. All other factors were held constant, allowing us to attribute the improvements directly to the judge model. Importantly, for evaluation on the test benchmarks, we use a single, fixed judge model throughout the paper to ensure consistency.

---

> > ### Comment · Reviewer_XJqA · 2025-11-25
> > **Reviewer Response:**
> >
> > 1. Frontier-model generalizability
> >
> > I accept that the authors lack compute to perform full RFT/SFT experiments at frontier scale. But my criticism is conceptual, not computational:
> >
> > You use observations from a family of instruction-fragile 7B Qwen2.5 models to motivate a strongly stated general claim: that “current MRMs” or “LLMs trained on MCQs” exhibit a single-answer bias requiring architectural or training interventions. The paper’s narrative implicitly generalizes this behavior to the entire class of medical LLMs, despite the fact that (as I noted and as the authors now confirm) frontier models trivially generate robust ranked differentials with prompting alone.
> >
> > If the paper’s thesis is “certain 7B MRMs are brittle”, that is a narrower, supportable claim - but it is not the way the abstract, introduction, or framing currently read. Without frontier-model controls, you cannot generalize from these specific models to MRMs broadly. I strongly recommend revising the claims to make their scope explicit.
> >
> > 2. Reward Functions
> > I appreciate the authors' addition of further clarification in this regard. While obviously any of these systems is going to be an abstraction of clinical reasoning, it just needs to be clearly identified as such.
> >
> > 3. Regarding LLM-As-Judge:
> >
> > I want to clarify why HealthBench does not support the manuscript’s current use of LLM-as-judge. HealthBench is not an example of simply deploying a judge model without verification. Clinicians first define rubric criteria; clinicians then grade responses against those criteria; and only after that does HealthBench perform a meta-evaluation comparing the LLM grader directly to clinician judgments. When the LLM’s grading falls within the range of human–human disagreement, then it is adopted as the default scorer.
> >
> > As HealthBench itself states:
> > “HealthBench scores are only meaningful if the models we use for model-based grading perform well—ideally as well as human experts. We therefore perform meta-evaluation, evaluating whether the evaluation corresponds to physician judgment with high fidelity.” (Section 8)
> >
> > This is exactly the sort of baseline validation that is missing here. LLM-as-judge can be effective—indeed, I am enthusiastic about it when properly anchored—but in clinical domains it must be tied to a human-grounded standard, otherwise judge variability cannot be meaningfully interpreted.
> >
> > Using an unvalidated judge inside an RFT loop is fine in principle (DeepSeek and others do this), but what ultimately matters is whether the final outputs have demonstrated clinical validity when assessed against an established, human-grounded metric. That external anchor is absent at present.
> >
> > Even unvalidated LLM-as-judge within the RFT loop itself is fine (a la Deepseek), so long as there is some clearly established clinical validity to the final grading. The proof is in the pudding as to whether whatever RFT process is used ultimately improves the final, validated metric.
> >
> > 4. Regarding judge replacement
> > Thank you for the clarification. My earlier comment arose from conflating the evaluation-time judge with the RFT reward-time judge. Exploring different judges within RFT is indeed reasonable and can be a useful sensitivity analysis. My concern applies only to the lack of clinician-validated evaluation at the end of the pipeline.

---

> ### Author Response · Authors · 2025-11-21
>
> > How do these concerns about answer length manifest across the range of various clinical tasks, beyond multiple choice? Does this apply to differential diagnosis alone? What of clinical decisionmaking in other realms?
>
> - We clarify that answer length is not the focus of this study, and therefore consideration of length-related metrics across a broader range of clinical tasks is beyond our scope. We also note that our benchmarks include both MCQ and open-ended versions, which we use across three answer-format settings: MCQ, QA, and List. Our primary focus is the ranked-list format. Differential diagnosis is one example of a clinical scenario where ranked lists are applicable, but the same format is also relevant in other settings such as generating prioritized next-step recommendations or listing plausible etiologies given incomplete information. Broader clinical decision-making is outside the scope of this paper; our focus is specifically on how MRMs handle answer formats, particularly ranked lists.
>
> We hope these answers clarify your questions. If anything remains unclear or you have additional questions, we would be happy to discuss them further.

---

### Official Review · Reviewer_MGaR · 2025-10-31

**Soundness:** 2
**Presentation:** 2
**Contribution:** 1
**Rating:** 2
**Confidence:** 4

**Summary:**

This paper argues that clinical decision-making inherently involves considering multiple possible answers, yet current medical reasoning models (MRMs) are trained to produce only one. To address this limitation, the authors propose enabling MRMs to generate ranked lists of plausible answers through prompting, supervised fine-tuning (SFT), and reinforcement fine-tuning (RFT) with novel ranking-based reward functions. Experiments across medical QA benchmarks show that RFT models generalize more robustly across answer formats (MCQ, QA, list) than SFT models, and that ranking-oriented rewards (e.g., MRR, LLM-judge) improve list quality. A case study on a modified MedQA dataset with multiple valid answers further demonstrates that MRMs often recognize correct alternatives overlooked by single-answer benchmarks.

**Strengths:**

1. The evaluation is comprehensive, covering a wide range of models and datasets, which convincingly demonstrates the effectiveness and generality of the proposed methods.

2. The case study on MedQA with multiple valid answers is particularly interesting and provides valuable insights into the limitations of single-answer benchmarks and the potential of ranked-list reasoning in clinical contexts.

**Weaknesses:**

1. Overstated novelty claim (“first systematic study”). Several recent works have already prompted LLMs to generate ranked differential diagnoses and evaluated them using top-k or position-aware metrics across diverse medical datasets and model families [1, 2, 3]. The paper should more clearly delineate its unique contribution, maybe emphasizing the reinforcement fine-tuning (RFT) reward design or curriculum learning over answer formats.

2. Limited analysis of SFT vs RFT generalization. The claim that “SFT-MCQ generalizes across formats while RFT generalizes both across formats and unseen examples” is intriguing but under-explained. The concept of answer format is not formally defined, and the paper lacks controlled ablations or causal evidence clarifying why RFT achieves superior transfer.

[1] Zhou, Yuxuan, et al. "Reliable and Diverse Evaluation of LLM Medical Knowledge Mastery." The Thirteenth International Conference on Learning Representations.

[2] Lin, Tianwei, et al. "HealthGPT: A Medical Large Vision-Language Model for Unifying Comprehension and Generation via Heterogeneous Knowledge Adaptation." Forty-second International Conference on Machine Learning.

[3] Lim, Seungseop, et al. "H-DDx: A Hierarchical Evaluation Framework for Differential Diagnosis." The Second Workshop on GenAI for Health: Potential, Trust, and Policy Compliance.

**Questions:**

1. On novelty and positioning: How does this work differ concretely from prior ranked-differential evaluations

2. On generalization mechanics: Could you provide controlled ablations isolating why RFT transfers across formats—such as (a) keeping compute and data identical for SFT vs RFT, and (b) varying prompt templates to rule out prompt or distribution confounds?

---

> ### Author Response · Authors · 2025-11-21
>
> Thank you for taking the time to review our work and for providing thoughtful feedback. We appreciate your recognition of the breadth of our evaluation and the value of the case study demonstrating the potential of ranked-list reasoning in clinical settings.
>
> > Overstated novelty claim (“first systematic study”). Several recent works have already prompted LLMs to generate ranked differential diagnoses and evaluated them using top-k or position-aware metrics across diverse medical datasets and model families [1, 2, 3]. The paper should more clearly delineate its unique contribution, maybe emphasizing the reinforcement fine-tuning (RFT) reward design or curriculum learning over answer formats. On novelty and positioning: How does this work differ concretely from prior ranked-differential evaluations
>
> - We would like to clarify that the “first systematic study” referenced in our paper specifically refers to the systematic study of how well an MRM trained on one answer format performs when instructed to respond in another answer format, which we define as robustness. We added Appendix A.4 Related Work to better position our paper. This focus is distinct from the mentioned papers and differs from them in the following ways:
> - **[1] Zhou, Yuxuan, et al. "Reliable and Diverse Evaluation of LLM Medical Knowledge Mastery." The Thirteenth International Conference on Learning Representations.**
> - This paper discusses how to generate a new, grounded evaluation set when a medical knowledge base is already available for assessing medical LLMs. In contrast, we do not propose new dataset; instead, we focus on robustness to answer format, with particular emphasis on the ranked-list format. Additionally, our work centers on medical reasoning models rather than general medical LLMs.
> - **[2] Lin, Tianwei, et al. "HealthGPT: A Medical Large Vision-Language Model for Unifying Comprehension and Generation via Heterogeneous Knowledge Adaptation." Forty-second International Conference on Machine Learning.**
> - This paper focuses on introducing various tasks involving visual modalities, such as X-ray comprehension, ultrasound interpretation, and image reconstruction. In contrast, our work centers on medical reasoning models that operate in a text-to-text paradigm. Visual-based tasks therefore fall outside the scope of our study.
> - **[3] Lim, Seungseop, et al. "H-DDx: A Hierarchical Evaluation Framework for Differential Diagnosis." The Second Workshop on GenAI for Health: Potential, Trust, and Policy Compliance.**
> - This paper introduces a hierarchical evaluation framework for differential diagnosis, using a retrieval and reranking pipeline to map free-text diagnoses to ICD-10 codes and applying a hierarchical metric. While our paper mentions differential diagnosis as one potential application of ranked lists, we do not focus exclusively on that task. Instead, our emphasis is on the answer format itself—specifically, the ranked-list format. We also do not propose a new metric; rather, we adopt standard non-hierarchical rank-aware metrics such as MRR. Although we introduce an evaluation scheme, our goals are broader: we study robustness to answer format and investigate how to enable MRMs to reliably follow instructions for producing ranked-list outputs.

---

> ### Author Response · Authors · 2025-11-21
>
> > Limited analysis of SFT vs RFT generalization. The claim that “SFT-MCQ generalizes across formats while RFT generalizes both across formats and unseen examples” is intriguing but under-explained. The concept of answer format is not formally defined, and the paper lacks controlled ablations or causal evidence clarifying why RFT achieves superior transfer. On generalization mechanics: Could you provide controlled ablations isolating why RFT transfers across formats—such as (a) keeping compute and data identical for SFT vs RFT, and (b) varying prompt templates to rule out prompt or distribution confounds?
>
> - We added further explanation of the generalization capabilities of RFT models on unseen examples and formats in Section 4.2.2. In particular, we elaborate on why RFT models retain stronger generalization, drawing on insights from prior work (https://arxiv.org/abs/2506.02126) discussing how RFT achieves superior transfer compared to SFT models.
> - We also expanded our explanation of answer formats in Section 2.1. Specifically, we define an answer format as “the structure in which the model must present the final answer in its generated response” and provide additional detail for each format included in our study.
> - Additionally, we note that the training datasets for both SFT and RFT were controlled to draw from the same data sources. We clarify that our study associates each prompt template with a corresponding answer format, along with a distinction between zero-shot and CoT prompting. This results in a total of six prompt templates across three answer formats with both zero-shot and CoT variants. We conduct a comprehensive analysis of how each prompt template affects model behavior in Section 3, and we include an ablation study on the impact of prompt template choice, particularly the inclusion or omission of CoT, for RFT in Section 4.3.1.
>
> We hope these answers clarify your questions. If anything remains unclear or you have additional questions, we would be happy to discuss them further.

---

### Official Review · Reviewer_qJ9Y · 2025-10-31

**Soundness:** 3
**Presentation:** 2
**Contribution:** 2
**Rating:** 4
**Confidence:** 3

**Summary:**

The paper conducted a systematic study of steering Medical Reasoning Models (MRM) via prompting and fine-tuning for generating ranked-list outputs.

The authors introduced an evaluation framework covering three answer formats and shown that many models know the correct answers but often fail to select the benchmark-preferred one.

They then conducted a comprehensive study of SFT and RFT under specific answer formats, finding that SFT with MCQ generalizes well across formats, while RFT models generalize both to unseen examples and across answer formats.

The paper also provided practical insights into RFT design choices, including reward design and curriculum-style sequencing of RFT to improve answer robustness.

**Strengths:**

This is a comprehensive systematic investigation of methods to enable MRMs to generate answers as ranked lists, a useful alternative to single answers in medical QA.

The ranked-list format aligns with clinicians’ real-world decision processes (e.g., weighing multiple differentials or treatments), which can reduce risk from over-reliance on a single LLM output.

The paper also explored reward designs for RFT on ranked lists and conducted ablations to identify optimal design choices such as the training curriculum, which is are actionable for the community and likely to generalize to other medical reasoning settings.

**Weaknesses:**

While the ranked-list format is a promising alternative to single-answer outputs, if a few incorrect answers are included and passed to clinicians, wouldn’t that still mislead or contribute to diagnostic failures? Does your training discourage listing incorrect options (e.g., via loss shaping or negative rewards)? A discussion of error propagation and interpretive risks from ranked outputs would clarify the safety implications.

As noted in the limitations, the ranked lists do not reflect probability magnitudes. Do you have strategies, via prompting or training, to elicit calibrated probabilities (e.g., model-reported confidence scores) or to learn them?

How does a simple baseline that ranks options by model likelihoods over the option tokens (e.g., a clinician pre-select top 5 likely diagnosis options) compare to explicit ranked-list training in terms of answer accuracy? Adding this baseline would quantify the added value of ranked-list training over simply outputting top choices, especially given your claim that models often already “know” the correct answer.

Did you include qualitative examples in the appendix illustrating SFT vs. RFT behaviors and their output formats? Including such examples would help readers interpret results sections like “RFT-QA Exhibits List-like Behavior” and “Reward Function Effects With RFT-List.” (If present, please point to them; if not, consider adding.)

Minor: please fix small typos in the introduction. Sections 3 and 4 both describe experimental setups; reducing subsection titles and headings could improve readability.

**Questions:**

Just for clarification, in 'Reward Function Effects With RFT-List', when you say 'RFT performs robustly on non-list formats and
generalizes better to unseen answer formats, what exactly do you mean: do you mean that RFT also obeys other formats you defined in training well other than ranked-list, or do you mean RFT sticks with ranked-list outputs even that there is instruction on other answer format?

---

> ### Author Response · Authors · 2025-11-21
>
> Thank you to reviewer qJ9Y for the thoughtful and constructive assessment of our work. We appreciate the recognition of the study’s breadth, the relevance of ranked-list outputs for clinical decision-making, and the value of the ablations and design insights. We addressed your questions in the following section.
>
> > While the ranked-list format is a promising alternative to single-answer outputs, if a few incorrect answers are included and passed to clinicians, wouldn’t that still mislead or contribute to diagnostic failures? Does your training discourage listing incorrect options (e.g., via loss shaping or negative rewards)? A discussion of error propagation and interpretive risks from ranked outputs would clarify the safety implications.
>
> - Thank you for raising this point. We have added further discussion on the safety aspects, implications, and potential mitigation approaches in the newly added Section 4.4 Discussions. In particular, we now emphasize that while our experiments are conducted in benchmark settings—where performance metrics such as accuracy dominate. An RFT model optimized solely for benchmark accuracy may overlook important constraints, such as avoiding potentially harmful suggestions or ensuring adequate coverage of rare but clinically significant conditions. We highlight that future work could develop more realistic benchmarks or integrate reward terms that explicitly incorporate safety, coverage, and clinical relevance.
>
> > As noted in the limitations, the ranked lists do not reflect probability magnitudes. Do you have strategies, via prompting or training, to elicit calibrated probabilities (e.g., model-reported confidence scores) or to learn them?
>
> - We added additional discussion on this topic in the newly added Section 4.4 Discussions. In particular, we highlight that there are concurrent studies on incentivizing models to produce calibrated confidence in RFT setups, such as https://arxiv.org/abs/2507.16806.
>
> > How does a simple baseline that ranks options by model likelihoods over the option tokens (e.g., a clinician pre-select top 5 likely diagnosis options) compare to explicit ranked-list training in terms of answer accuracy? Adding this baseline would quantify the added value of ranked-list training over simply outputting top choices, especially given your claim that models often already “know” the correct answer.
>
> - As we noted throughout the paper, benchmarks with multiple valid options are currently unavailable. Constructing such a benchmark is by no means an easy or inexpensive task. Given the scale required, it falls outside the scope of our work, and we believe it merits its own dedicated study. Instead, in Section 5, we simulate this scenario in a case study to demonstrate the potential.
> - Regarding baselines, because our study focuses on answer-format robustness, we consider baselines from a methodological perspective—ranging from prompting-based approaches (no training) to training-based approaches. We highlight that simply prompting MRMs is often ineffective (see Section 3; “However, AlphaMed, an MRM trained with RFT for MCQ, fails to generalize to other answer formats; we revisit this finding again in Section 4.2.“), and we further investigate how different training strategies influence this aspect of model behavior. In summary, we show the benefits of training over simple prompting, which serves as our core baseline.
> - We argue that the model often already knows the answers, based on the increased accuracy observed in Section 3 when we instruct the model to output all possible answers in a list-evaluation setting, along with the fact that the mean correct position indicates the correct answer is frequently not ranked first. We acknowledge, however, that additional factors may also contribute to these results. Given the scope of our study, we leave confirmation of this observation to future work. We have revised our wording to more precisely reflect this uncertainty and added the note: “Although additional studies are required to confirm these causal effects.”

---

> ### Author Response · Authors · 2025-11-21
>
> > Did you include qualitative examples in the appendix illustrating SFT vs. RFT behaviors and their output formats? Including such examples would help readers interpret results sections like “RFT-QA Exhibits List-like Behavior” and “Reward Function Effects With RFT-List.” (If present, please point to them; if not, consider adding.)
>
> - Thank you for your suggestions. We have added more than ten qualitative examples to our paper in Appendix O to illustrate many of the key points.
>
> > Minor: please fix small typos in the introduction. Sections 3 and 4 both describe experimental setups; reducing subsection titles and headings could improve readability.
>
> - Thank you. We have fixed the typos in the Introduction. We intentionally placed the experimental setups for SFT and RFT near the corresponding results and discussions to keep the relevant information co-located.
>
> > Just for clarification, in 'Reward Function Effects With RFT-List', when you say 'RFT performs robustly on non-list formats and generalizes better to unseen answer formats, what exactly do you mean: do you mean that RFT also obeys other formats you defined in training well other than ranked-list, or do you mean RFT sticks with ranked-list outputs even that there is instruction on other answer format?
>
> - We mean that “RFT also follows other formats defined during training well, not only the ranked-list format”. We have also added a definition of robustness early in the Introduction: “how well a model trained in one format can follow instructions to produce another”.
>
> We hope these answers address your questions. If anything remains unclear or you have additional questions, we would be happy to discuss them further.

---

### Author Response · Authors · 2025-11-21

We thank all reviewers for their careful evaluation of our work and for the thoughtful comments and suggestions. We appreciate the recognition of our motivation to move beyond narrow MCQ-style evaluation, the relevance of studying ranked-list reasoning in clinical contexts, the breadth of our empirical analysis, and the technical rigor of our comparisons across prompting, supervised fine-tuning, and reinforcement fine-tuning. We are also grateful for the positive feedback on the clarity and structure of the paper.

In response to the reviewers’ suggestions, we have revised and expanded the manuscript. These updates include a reworked abstract,introduction, and related work to better articulate the problem setting and contributions addressing reviewers Reviewer XJqA, MGaR, and 2LD3; additional definitions and clarifications on robustness and answer formats addressing Reviewers qJ9Y and MGaR; more than ten qualitative examples integrated throughout the paper to illustrate key behaviors in response to Reviewer qJ9Y; expanded discussion on the broader implications of this work for real-world medical applications in response to Reviewers qJ9Y, XJqA, and 2LD3; and additional analysis clarifying the generalization properties and differences between RFT and SFT models addressing Reviewers MGaR and 2LD3. We believe these revisions strengthen the presentation and better situate our contributions within the larger landscape of medical reasoning research.

We appreciate the reviewers’ insights and hope the updated manuscript addresses their concerns and clarifies the value of our study.

---

### Note · Authors · 2025-12-01

I have read and agree with the venue's withdrawal policy on behalf of myself and my co-authors.